# Small leucine-rich proteoglycans inhibit CNS regeneration by modifying the structural and mechanical properties of the lesion environment

Julia Kolb [1,2,3], Vasiliki Tsata [4,5,14], Nora John [1,2,3,14], Kyoohyun Kim [1,2], Conrad Möckel[1,2,6], Gonzalo Rosso[1,2], Veronika Kurbel [7], Asha Parmar [1,2,6], Gargi Sharma[1,8], Kristina Karandasheva [7], Shada Abuhattum [1,2], Olga Lyraki[1,2,3], Timon Beck [1,2], Paul Müller [1,2], Raimund Schlüßler [9], Renato Frischknecht [3], Anja Wehner[10], Nicole Krombholz[10], Barbara Steigenberger[10], Dimitris Beis [4,11], Aya Takeoka[12,13], Ingmar Blümcke [7], Stephanie Möllmert[1,2], Kanwarpal Singh[1,2,6], Jochen Guck [1,2,6], Katja Kobow [7] & Daniel Wehner [1,2] ✉

Extracellular matrix (ECM) deposition after central nervous system (CNS) injury leads to inhibitory scarring in humans and other mammals, whereas it facilitates axon regeneration in the zebrafish. However, the molecular basis of these different fates is not understood. Here, we identify small leucine-rich proteoglycans (SLRPs) as a contributing factor to regeneration failure in mammals. We demonstrate that the SLRPs chondroadherin, fibromodulin, lumican, and prolargin are enriched in rodent and human but not zebrafish CNS lesions. Targeting SLRPs to the zebrafish injury ECM inhibits axon regeneration and functional recovery. Mechanistically, we find that SLRPs confer mechano-structural properties to the lesion environment that are adverse to axon growth. Our study reveals SLRPs as inhibitory ECM factors that impair axon regeneration by modifying tissue mechanics and structure, and identifies their enrichment as a feature of human brain and spinal cord lesions. These findings imply that SLRPs may be targets for therapeutic strategies to promote CNS regeneration.

The ability to regenerate long-distance axonal connections after central nervous system (CNS) injury differs significantly among vertebrates. Why certain species, such as zebrafish, possess a high regenerative capacity but not others is poorly understood[1]. Fibrous scar formation is considered to be a major factor in limiting axon regeneration in the adult mammalian CNS[2]. Several scar components, such as myelin-associated factors, basal lamina components, and high molecular weight chondroitin sulfate proteoglycans (CSPGs), have been identified as inhibitors of axonal regrowth through mechanisms including growth cone collapse, repellence or entrapment, and prevention of inflammation resolution[3–5]. However, removing these extracellular matrix (ECM) factors results in only modest regeneration, suggesting that critical molecules and mechanisms contributing to regeneration failure remain to be discovered[6–8]. The CNS scar may inhibit axon growth not only through its biochemical composition but also by changes in local mechanical properties of the

microenvironment, such as elasticity and viscosity[9–12]. However, in vivo evidence for a causal relationship between scar tissue mechanics and regenerative success is lacking. Moreover, molecular factors that influence the mechanical properties of CNS scars have not been identified.

Unlike humans and other mammals, zebrafish establish an axon growth-conducive ECM after CNS injury, leading to recovery of swimming function both at larval and adult stages[13–17]. It remains obscure as to why injury-associated ECM deposits inhibit axon regeneration in the mammalian CNS but not in zebrafish. Here, we identify small leucine-rich proteoglycans (SLRPs) as ECM factors which drive CNS healing toward inhibitory scarring. We demonstrate that the SLRPs chondroadherin, fibromodulin, lumican, and prolargin are enriched in human and rodent but not zebrafish CNS lesions. Increasing the abundance of SLRPs in the zebrafish injury ECM inhibits axon regeneration and functional recovery. Mechanistically, we find that SLRPs confer mechano-structural properties to the lesion environment that render it adverse to axon growth. Our data identify SLRPs as inhibitory ECM factors that impair axon regeneration by altering tissue mechanics and structure, and reveal their enrichment as a feature of human CNS lesions. Targeting SLRPs therefore presents itself as a potential therapeutic strategy to promote axon growth across CNS lesions.

## Results

### Matrisome dynamics of zebrafish spinal cord regeneration

In order to identify factors that confer axon growth-limiting properties to CNS scars by altering tissue mechanics, we first set out to map the changes in ECM composition in a regeneration context. To that aim, we applied label-free mass spectrometry (MS)-based quantitative proteomics to a larval zebrafish spinal cord injury (SCI) model. In this injury paradigm, the spinal cord is completely transected, which allows axon regeneration across an ECM-rich, non-neural lesion environment and recovery of swimming function within two days post-lesion (dpl; Fig. 1a, b)[14,18]. Proteomic profiling of trunk tissue containing the lesion site at 1 dpl and 2 dpl as well as corresponding age-matched unlesioned control tissue identified 4782 unique proteins after quality control (Supplementary Fig. 1a, b). Differential abundance analysis (FDR < 0.1, | FC | ≥ 1.3, $s_0 = 0.1$) revealed 910 proteins whose abundance was altered in 1 dpl compared to unlesioned control samples (573 up- and 337 down-regulated; Supplementary Fig. 1c, e). The abundance of 656 proteins differed between 2 dpl and unlesioned control samples (443 up- and 213 down-regulated; Supplementary Fig. 1d, f). Reactome analysis of enriched proteins resulted in several ECM-associated terms being overrepresented at 1 dpl and 2 dpl (Supplementary Fig. 1g, h). We thus examined the matrisome, which can be subdivided into core matrisome (glycoproteins, collagens, proteoglycans), matrisome-associated (ECM affiliated proteins, ECM regulators, secreted factors), and putative matrisome proteins (Fig. 1c, d)[19]. Among the matrisome proteins which exhibited a high abundance at 1 dpl were several proteins that have previously been reported to show increased expression after SCI in zebrafish, including Aspn, Cthrc1a, Col5a1, Col6a2, Col12a1a, Col12a1b, Fn1a, Fn1b, Tnc, and Thbs2b[13–15,20]. In addition, we identified immune system-related factors, such as cathepsins, serpins, galectins, and interleukins, which is consistent with the critical role of injury-activated macrophages at this stage of regeneration[21,22]. At 2 dpl, the number of matrisome proteins exhibiting an altered abundance decreased by 28%, compared to 1 dpl. Comparing differentially regulated proteins across time points showed that 30 such matrisome proteins were common to 1 dpl and 2 dpl (asterisks in Fig. 1c, d), including proteins previously implicated in axon growth promotion or guidance: Col12a1a, Col12a1b, Fn1a, Fn1b, Tnc, Cthrc1a, and Thbs1a[13,14,20,23–29]. 34 proteins with altered abundance were unique to 1 dpl, and 16 proteins to 2 dpl. Among the uniquely differentially regulated matrisome proteins at 1 dpl, we identified nine immune system-related and coagulation factors (Ctsa, Ctsla, Ctsc, F2,

Il16, Lgals9l3, Plg, Serpinf2b, Serpine2), consistent with the initial blood clotting reaction and transient proinflammatory phase after SCI in zebrafish[21,30]. To further validate our MS results, we examined whether changes in mRNA levels accompanied the alterations in protein abundance. We performed in situ hybridization (ISH) to analyze the expression of genes coding for 30 differentially regulated matrisome proteins which exhibited high abundance at 1 dpl. Transcripts of all analyzed genes were locally upregulated in the lesion site compared to both adjacent unlesioned trunk tissue and unlesioned age-matched controls (Fig. 1e; Supplementary Fig. 1i). Furthermore, fluorescence ISH on tissue sections for 10 selected genes confirmed transcript induction in the spinal cord stump, its immediate vicinity, or the highly disorganized lesion core (Fig. 1f). This reinforces the findings of our proteomics profiling. Collectively, these data identify the dynamics of the matrisome landscape after SCI in a vertebrate species, which exhibits a high regenerative capacity for the CNS.

### Distinct matrisome signatures after SCI in rodents and zebrafish

To identify interspecies differences in the ECM composition that could account for the ability of severed axons to regrow after CNS injury, we compared the zebrafish proteomics dataset with that of adult Sprague-Dawley rats at seven days post-contusion SCI[31]. We first screened for matrisome proteins that were enriched in the zebrafish lesion site (FDR < 0.1, FC ≥ 1.3) but down-regulated or not significantly regulated after SCI in rat (FDR < 0.1, FC ≤ −1.3 | n.s.). This identified four proteins (Supplementary Fig. 2a). By contrast, 61 matrisome proteins were down-regulated or not significantly regulated after SCI in zebrafish (FDR < 0.1, FC ≤ −1.3 | n.s.) but enriched in the rat lesion site (FDR < 0.1, FC ≥ 1.3) (Fig. 2a). This suggests that the regeneration-permissive properties of the zebrafish injury ECM can be attributed to the absence of axon growth-limiting components rather than the presence of species-specific growth-promoting factors. Indeed, among the differentially regulated matrisome proteins that showed low abundance in the zebrafish spinal lesion site were the neurite growth-inhibitory factors neurocan and 16 components of the basal lamina (including type IV collagens, laminins, nidogens, heparan sulfate proteoglycans, Fbln1, Sparc)[4,32,33]. Quantitative RT-PCR (qRT-PCR) confirmed that the expression of ncanb and basal lamina components was not upregulated in the zebrafish spinal lesion site at 1 dpl (Supplementary Fig. 2b). Similarly, anti-Col IV immunoreactivity was not locally increased at 1 dpl (Supplementary Fig. 2c). Thus, the CSPG neurocan and basal lamina networks are not principal constituents of the zebrafish injury ECM. These data demonstrate the value of cross-species comparative approaches in identifying inhibitory components of the mammalian injury ECM.

The comparative dataset further revealed seven members of the highly conserved small leucine-rich proteoglycan (SLRP) family to be enriched in the rat spinal lesion site but down-regulated or not significantly regulated after SCI in zebrafish, namely chondroadherin (Chad), lumican (Lum), osteoglycin (Ogna), decorin (Dcn), fibromodulin (Fmoda, Fmodb), and prolargin (Prelp) (Fig. 2a). Assessment of additional proteomics profiles showed comparable enrichment of these SLRPs in rats at seven days and eight weeks post-contusion SCI (Supplementary Fig. 2d)[31,34,35]. Interestingly, asporin (Aspn) was the only SLRP family member which exhibited an increased abundance after SCI in both rat and zebrafish (Supplementary Fig. 2d). To ascertain whether the low protein abundance of SLRPs in the zebrafish spinal lesion site is also reflected at the transcriptional level, we performed qRT-PCR. This revealed that with the exception of aspn, expression of all 21 SLRPs present in the zebrafish genome was not increased at 1 dpl as compared to unlesioned controls (Fig. 2b; Supplementary Fig. 2e). Average fold changes of aspn, chad, dcn, fmoda, fmodb, lum, ogna, and prelp, as determined by qRT-PCR, correlated with the MS data (Pearson correlation, $R^2 = 0.9443$), thus further validating our proteomics profile (Supplementary Fig. 2f). Additionally,

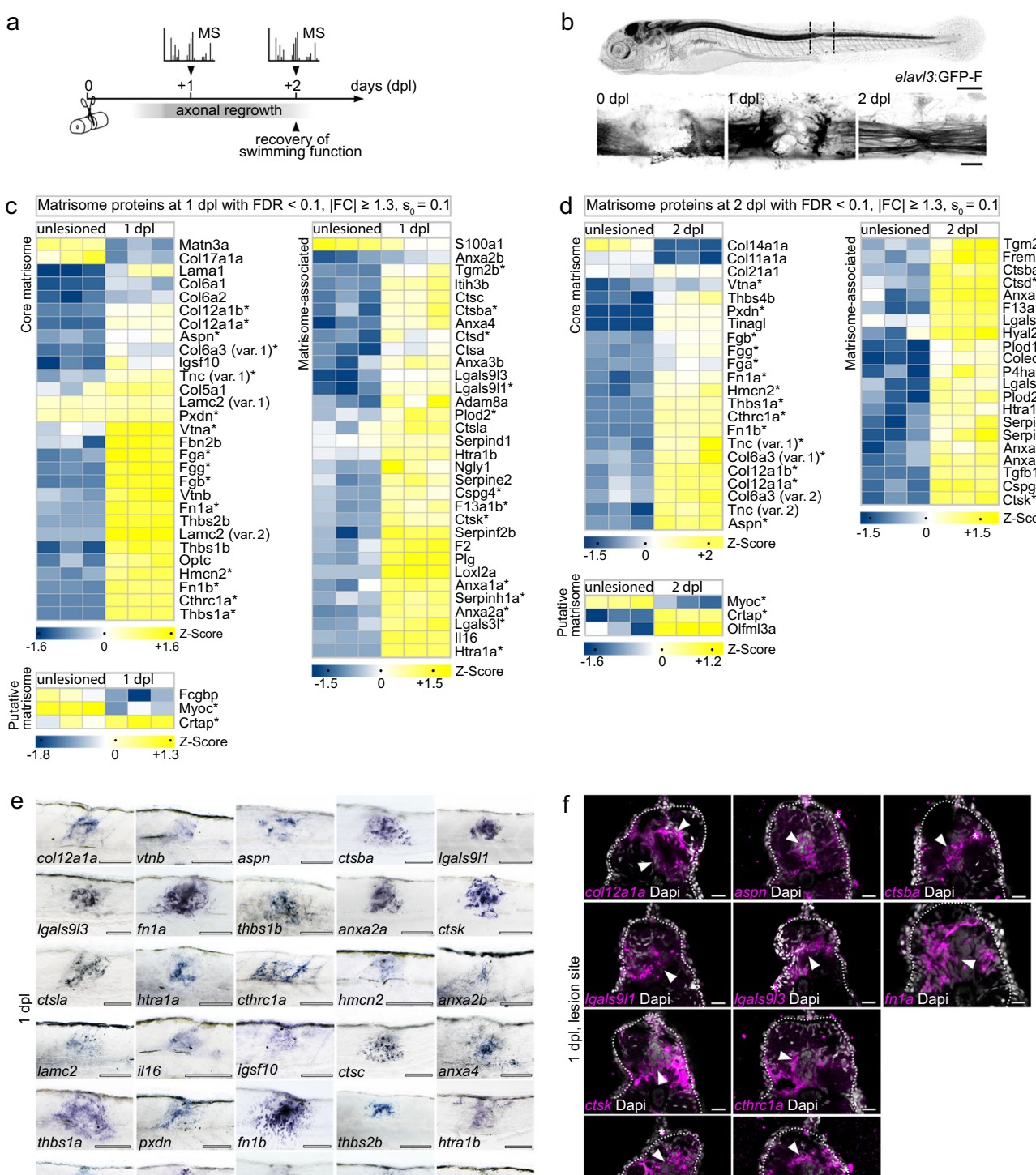

**Fig. 1 | Mass spectrometry-based quantitative proteomics reveals changes in ECM composition during zebrafish spinal cord regeneration. a** Timeline of axonal regrowth and functional recovery after SCI in larval zebrafish. Timepoints of tissue collection for mass spectrometry (MS) analysis are indicated. **b** Time course of axonal regrowth after spinal cord transection. Shown is the same animal at different timepoints after SCI. Dashed lines indicate the dissected trunk region for MS analysis. Images shown are maximum intensity projections of the spinal lesion site (lateral view; rostral is left). **c**–**d** Heatmaps of matrisome proteins exhibiting differential abundance between lesioned (1 dpl, **c**; 2 dpl, **d**) and unlesioned age-matched groups ($n = 3$ independent biological replicates for each experimental group). Each column represents one biological replicate and each row one protein.

Permutation-based FDR calculation, two-tailed Student's $t$-test. Asterisks indicate matrisome proteins that are common to both timepoints. **e** Expression of the indicated genes, coding for differentially regulated matrisome proteins, is upregulated in the lesion site, as determined by in situ hybridization (ISH; lateral view; rostral is left). $n \geq 9$ animals for each gene. **f** Expression of indicated genes, coding for differentially regulated matrisome proteins, is detected in the spinal cord stump, its immediate vicinity, or the highly disorganized lesion core (arrowhead), as determined by fluorescence ISH on transversal tissue sections (dorsal is up). Asterisks indicate staining artifacts. $n \geq 5$ animals for each gene. **a**–**f** Scale bars: 250 μm (**b**, top), 100 μm (**e**), 25 μm (**b**, bottom), 20 μm (**f**). dpl days post-lesion, FC fold change, FDR false discovery rate, var variant.

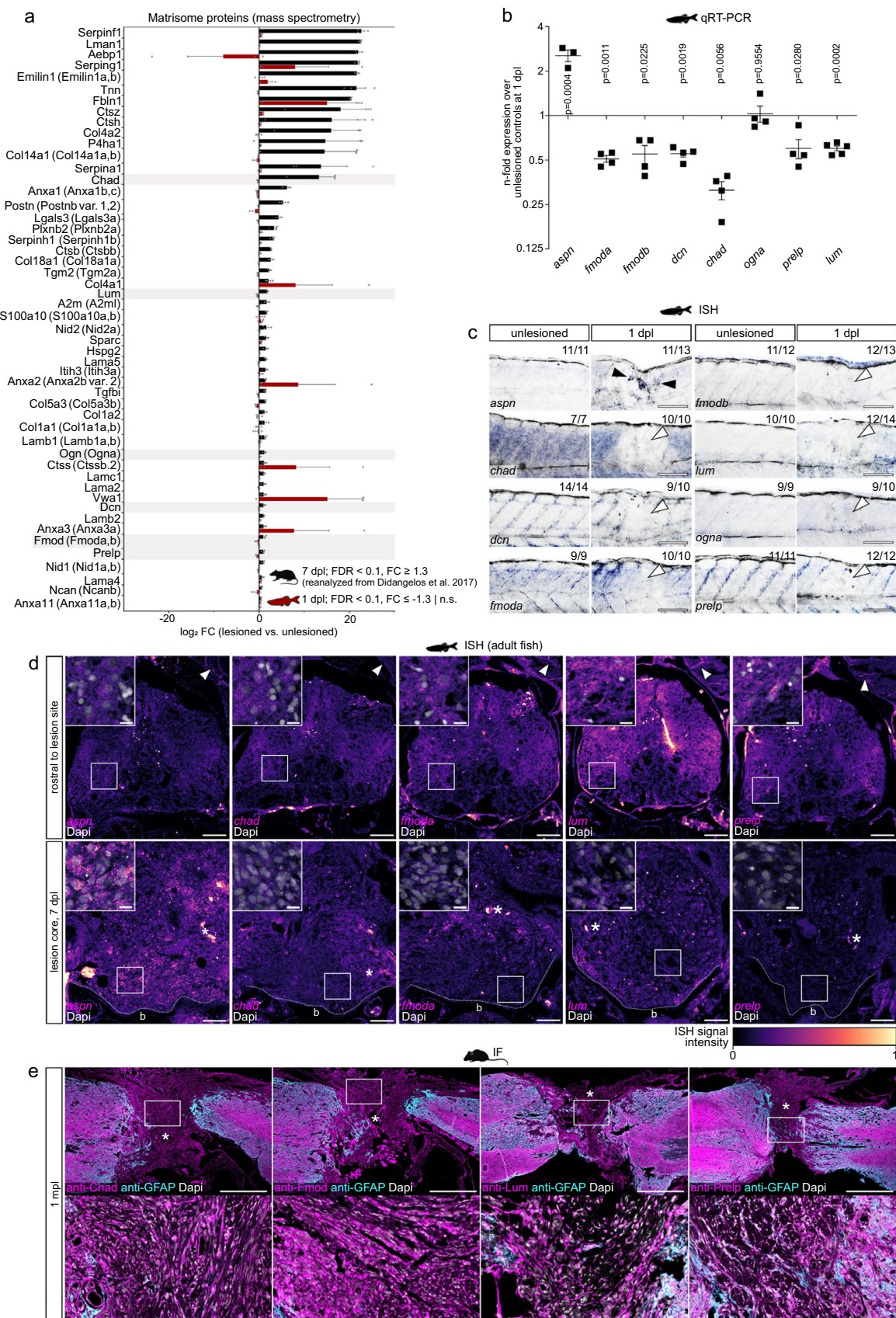

ISH showed upregulation of *aspn* transcripts in the lesion site at 1 dpl while *chad*, *dcn*, *fmoda*, *fmodb*, *lum*, *ogna*, and *prelp* expression was not locally increased (Figs. 1f and 2c; Supplementary Fig. 2g). Altogether, these data reveal an opposing abundance of Chad, Lum, Ogn,

Dcn, Fmod, and Prelp proteins in the spinal lesion site of rat and zebrafish, respectively.

To obtain further evidence that the capacity for axon regeneration scales with SLRP protein abundance, we analyzed their expression

**Fig. 2 | SLRPs are differentially enriched in CNS lesions of rodents and zebrafish.** **a** Matrisome proteins that exhibit a high abundance after SCI in rat (black) but a low abundance in the zebrafish (red) spinal lesion site are shown. The SLRPs Chad, Lum, Ogn (Ogna), Dcn, Fmod (Fmoda, Fmodb), and Prelp are highlighted. Data are means ± SEM. **b** Fold change expression of indicated genes in the zebrafish spinal lesion site over unlesioned controls, as determined by qRT-PCR. Expression of *aspn* but not *fmoda*, *fmodb*, *dcn*, *chad*, *ogna*, *prelp*, and *lum* is upregulated. Fold change values are presented in log scale. Each data point represents one independent experiment. Data are means ± SEM. Paired two-tailed Student's *t*-test. **c** Expression of *aspn* (black arrowheads) but not *chad*, *dcn*, *fmoda*, *fmodb*, *lum*, *ogna*, and *prelp* (white arrowheads) is upregulated in the zebrafish spinal lesion site, as determined by in situ hybridization (ISH). The number of animals displaying the phenotype and the total number of animals is given. Images show lesion site or unlesioned trunk (lateral view; rostral is left). **d** Expression of *aspn* but not *chad*, *fmoda*, *lum*, and *prelp* is increased in the spinal lesion core of adult zebrafish, as compared to baseline levels in rostral control segments of the same animal. Arrowheads indicate fluorescence ISH signal in myotendinous junctions. Asterisks indicate signal in blood vessels. Shown are transversal sections (dorsal is up). Dapi channel is only shown in insets. Dashed lines indicate vertebra. The same results were obtained in three indepedent experiments over *n* = 6 animals. **e** Chad, Fmod, Lum, and Prelp proteins are abundant in the fibrous scar after spinal cord transection in adult mice, as detected by immunofluorescence (IF). Images shown are sagittal sections. Asterisks indicate lesion core. Dapi channel is only shown in insets. *n* = 3 animals for each protein. **a**–**e** Scale bars: 500 μm (**e**), 100 μm (**c**), 50 μm (**d**, insets in **e**), 10 μm (insets in **d**). b bone, dpl days post-lesion, FC fold change, FDR false discovery rate, mpl months post-lesion, n.s not significant. The rat and mouse icons in **a** and **e** were created using BioRender. Source data are provided as a Source Data file.

after SCI in adult zebrafish and adult mice, which possess high and low regenerative capacity for the CNS, respectively. We decided to concentrate on Fmod, Lum, and Prelp due to their structural similarities as class II SLRPs. Additionally, we focused on Chad which showed the highest abundance among SLRPs in the rat spinal lesion site (Fig. 2a). Consistent with our findings in larval animals, fluorescence ISH on tissue sections of adult zebrafish revealed upregulation of *aspn* expression in the highly disorganized lesion core at 7 dpl, whereas expression of *chad*, *fmoda*, *lum*, and *prelp* was not increased over baseline levels (Fig. 2d). In contrast, we found prominent anti-Chad, anti-Fmod, anti-Lum, and anti-Prelp immunoreactivity in spinal cord stumps and the lesion core of adult mice at one month after spinal cord transection (Fig. 2e). The abundance of Chad, Fmod, Lum, and Prelp proteins in the injury ECM therefore reciprocally correlates with the CNS regenerative capacity.

## SLRPs are abundant in human CNS lesions

We next sought to determine whether the SLRPs CHAD, FMOD, LUM, and PRELP are abundant in the injured human CNS. In surgically removed brain tissue samples from six patients with traumatic brain injury (TBI) or brain surgery, we detected prominent anti-CHAD, anti-FMOD, anti-LUM, and anti-PRELP immunoreactivity in areas of scarring caused by contusion, local hemorrhage, or previous surgery (Fig. 3; Supplementary Fig. 3a, b; Supplementary Table 1). By contrast, negligible immunoreactivity was observed in a reference group of six human brain autopsy and biopsy controls without signs of fibrous scarring (Supplementary Fig. 3c, d; Supplementary Table 1). We next analyzed post-mortem spinal cord tissue from six patients with traumatic SCI. SCI occurred in the cervical region through compression or contusion, and patients survived between 9 and 111 days after injury (subacute to intermediate phase; Supplementary Table 2). We found a localized increase in immunoreactivity in the lesion epicenter, as compared to rostral or caudal control segments of the same patient, for anti-LUM in five out of six cases, for anti-PRELP and anti-FMOD in four out of six cases, and anti-CHAD in two out of six cases (Fig. 4; Supplementary Table 3). The enrichment of SLRPs is therefore a feature of human CNS lesions.

## SLRPs are inhibitory to CNS axon regeneration

Since CHAD, FMOD, LUM, and PRELP proteins are highly abundant in the injury ECM of poorly regenerating rodents and humans but are absent in zebrafish with a high regenerative capacity for the CNS, we hypothesized that SLRPs inhibit regeneration. We therefore investigated the effect of increasing SLRP protein levels in the zebrafish injury ECM on axonal regrowth and functional recovery after SCI. We generated doxycycline (DOX)-inducible Tet-responder zebrafish lines to target expression of *chad-mcherry*, *fmoda-mcherry*, *lum-mcherry*, or *prelp-mcherry* fusions specifically to *pdgfrb*⁺ myoseptal and perivascular cells when used in combination with a *pdgfrb* promotor-driven Tet-activator line (henceforth abbreviated as *pdgfrb*:SLRP) (Fig. 5a).

Additionally, we created a Tet-responder line for the selective induction of *aspn-mcherry* to control for potential overexpression artifacts of SLRPs. *pdgfrb*⁺ cells are rapidly recruited in response to SCI to secrete an injury-specific ECM in the lesion site (Fig. 5b)[13]. Moreover, qRT-PCR on GFP⁺ and GFP⁻ cells isolated from uninjured *pdgfrb*:GFP transgenic animals by FACS revealed several-fold higher expression levels of endogenous *aspn* ($\overline{FC}$ = 22.3), *chad* ($\overline{FC}$ = 3.8), *fmoda* ($\overline{FC}$ = 4.8), *lum* ($\overline{FC}$ = 6.6), and *prelp* ($\overline{FC}$ = 10.5) transcripts in *pdgfrb*⁺ cells as compared to *pdgfrb*⁻ cells (Supplementary Fig. 4a). This indicates that *pdgfrb*⁺ cells are a main source of SLRP proteins under physiological conditions. Following SCI, *fmoda*, *lum*, and *prelp* expression is downregulated whereas *aspn* is upregulated in *pdgfrb*⁺ cells in zebrafish, which we have reported previously[13]. Together, this makes *pdgfrb*⁺ fibroblast-like cells an ideal target for manipulating the ECM in the zebrafish spinal lesion site.

Induction of *aspn-mcherry*, *chad-mcherry*, *fmoda-mcherry*, *lum-mcherry*, or *prelp-mcherry* expression in unlesioned *pdgfrb*:SLRP transgenic animals resulted in fluorescent labeling of mainly the myosepta and vasculature, thus resembling the *pdgfrb*⁺ cell niche (Supplementary Fig. 4b, c)[13]. 3D reconstruction of high-resolution images of *pdgfrb*:SLRP;*pdgfrb*:GFP transgenics revealed accumulation of mCherry signal in the pericellular environment of *pdgfrb*⁺ cells (Supplementary Fig. 4d). Furthermore, we observed variations in the mCherry fluorescence pattern among different SLRP proteins, including distinct mesh and fiber-like structures in the case of Aspn, Fmoda, and Prelp (Supplementary Fig. 4b, d). Together, this supports the secretion and binding of the SLRP-mCherry fusion proteins to fibrillar ECM[36]. Importantly, using the TUNEL assay showed that TetON system-driven expression of *chad*, *fmoda*, *lum*, or *prelp* did not lead to an increase in the number of apoptotic *pdgfrb*⁺ cells over baseline levels (Supplementary Fig. 4e). Of note, our protocol was sufficiently sensitive to detect *pdgfrb*⁺/TUNEL⁺ cells in whole-mount preparations following nitroreductase-mediated targeted ablation in unlesioned *pdgfrb*:GFP;*pdgfrb*:NTR-mCherry transgenic animals (Supplementary Fig. 4f). At 1 dpl, in the highly disorganized lesion core of *pdgfrb*:SLRP;*pdgfrb*:GFP transgenic animals, *pdgfrb*⁺ fibroblasts selectively expressed the *slrp-mcherry* fusions (Fig. 5c, d). Furthermore, mCherry fluorescence signal accumulated in the lesion core, indicating enrichment of the secreted proteins (Fig. 5e). *pdgfrb*:SLRP transgenics thereby enable the experimental increase of SLRP protein levels in the injury ECM after SCI.

To assess whether axon regeneration is affected in *pdgfrb*:SLRP transgenics, we determined the thickness of the axonal bridge that reconnects the severed spinal cord ends in live *elavl3*:GFP-F transgenic animals at 2 dpl, a measure that correlates with the recovery of swimming function (Fig. 5f)[21]. We found that the average axonal bridge thickness was reduced by 41-52% when Chad, Fmoda, Lum, or Prelp was targeted to the injury ECM, as compared to their controls (Fig. 5g; Supplementary Fig. 4g). Moreover, quantification of swimming distance at 2 dpl showed that *pdgfrb*:SLRP transgenics exhibited worse functional recovery than their controls (30-42% reduced swim

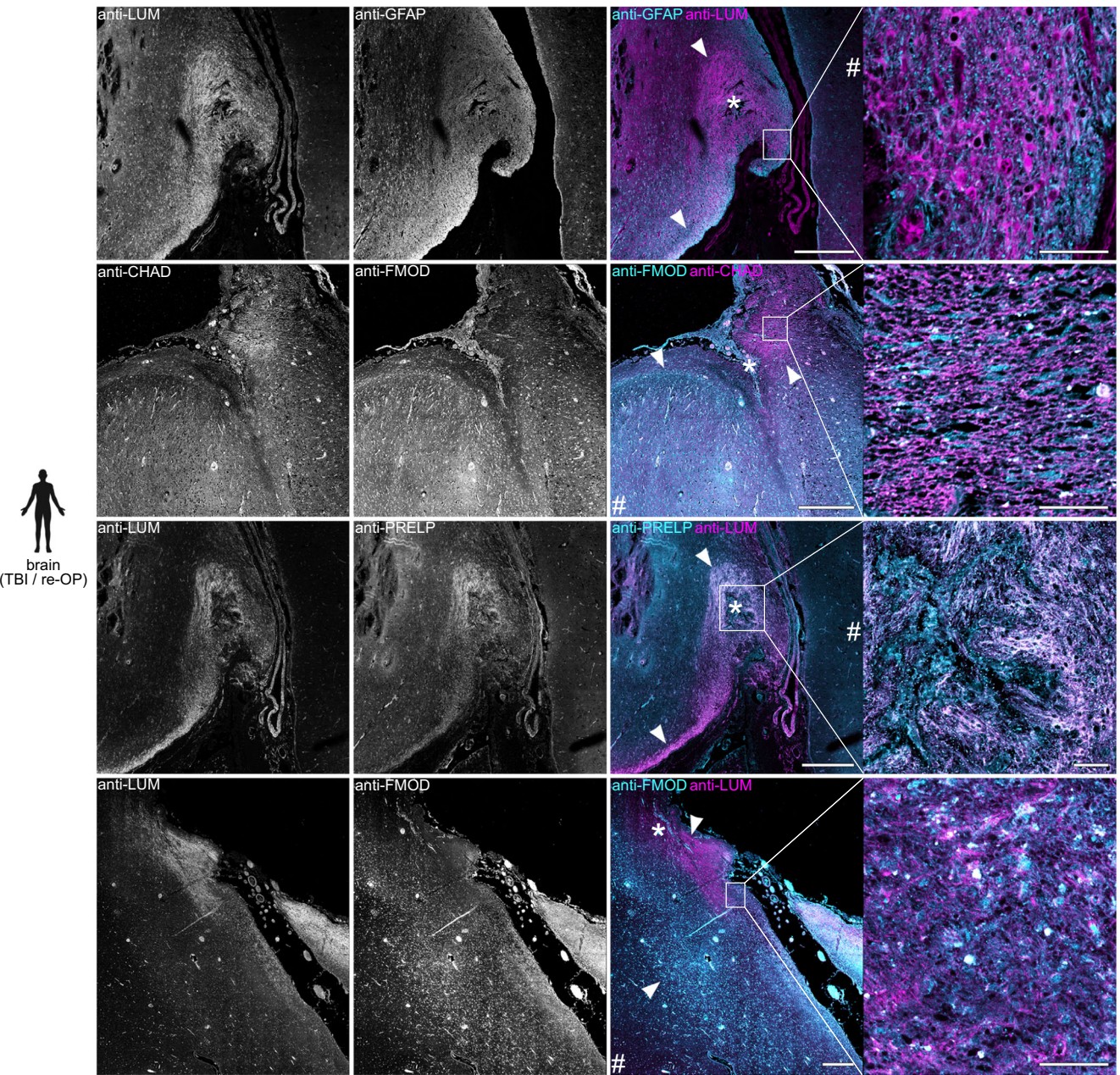

**Fig. 3 | SLRPs are enriched in human brain lesions.** Anti-CHAD, anti-FMOD, anti-LUM, and anti-PRELP immunoreactivity is locally increased in areas of scarring caused by contusion, local hemorrhage, or previous surgery (asterisks indicate lesion center, arrowheads mark scar boundaries) in the human brain, as compared to regions distant to the primary lesion site (hash sign), as well as human brain controls with no signs of fibrotic scarring (see Supplementary Fig. 3). Shown are coronal sections of brain tissue from patients with traumatic brain injury (TBI) or previous surgery (re-OP; bottom panel). Six ($n = 6$) cases with scars following TBI or previous surgery were analyzed and showed similar results (Supplementary Table 1). Scale bars: 500 μm, 50 μm (insets). The human icon in the panel was created using BioRender.

distance) (Fig. 5h). The effect on axon regeneration and functional recovery was further enhanced when two SLRPs (Chad, Lum) were concomitantly targeted to the injury ECM (73% reduced axonal bridge thickness, 60% reduced swim distance; Fig. 5i, j). Notably, targeting Aspn to the injury ECM had no effect on axon regeneration and recovery of swimming function, thus supporting the specificity of the observed phenotypes (Fig. 5g, h; Supplementary Fig. 4g). Our data therefore identify Chad, Fmod, Lum, and Prelp as ECM factors that inhibit CNS axon regeneration in vivo. Interestingly, bridging of astroglia-like processes, which cross the non-neural lesion site independently of regenerating axons, was also reduced by 44-54% when Chad, Fmoda, Lum, or Prelp was targeted to injury ECM (analyzed in *her4.3*:GFP-F transgenic animals; Supplementary Fig. 4h)[14,37,38]. This

suggests that the injury ECM may be less permissive to neural regeneration in the presence of SLRPs.

## SLRPs do not directly act on neurons to inhibit neurite extension

To elucidate the mechanism by which SLRPs inhibit neurite growth, we first examined a potential direct (biochemical) interaction with neurons, which has been reported for CSPGs and myelin-associated inhibitors[39–41]. To that aim, we used an *Xla.Tubb* promoter-driven Tet-activator line to target *chad*, *fmoda*, *lum*, or *prelp* expression specifically to neurons, including brainstem descending and spinal interneurons (Fig. 6a; Supplementary Fig. 5a, b)[14,42]. Induction of *chad-mcherry*, *fmoda-mcherry*, *lum-mcherry*, or *prelp-mcherry* expression in *Xla.Tubb*:TetA;*TetRE*:SLRP-mCherry transgenic animals (henceforth

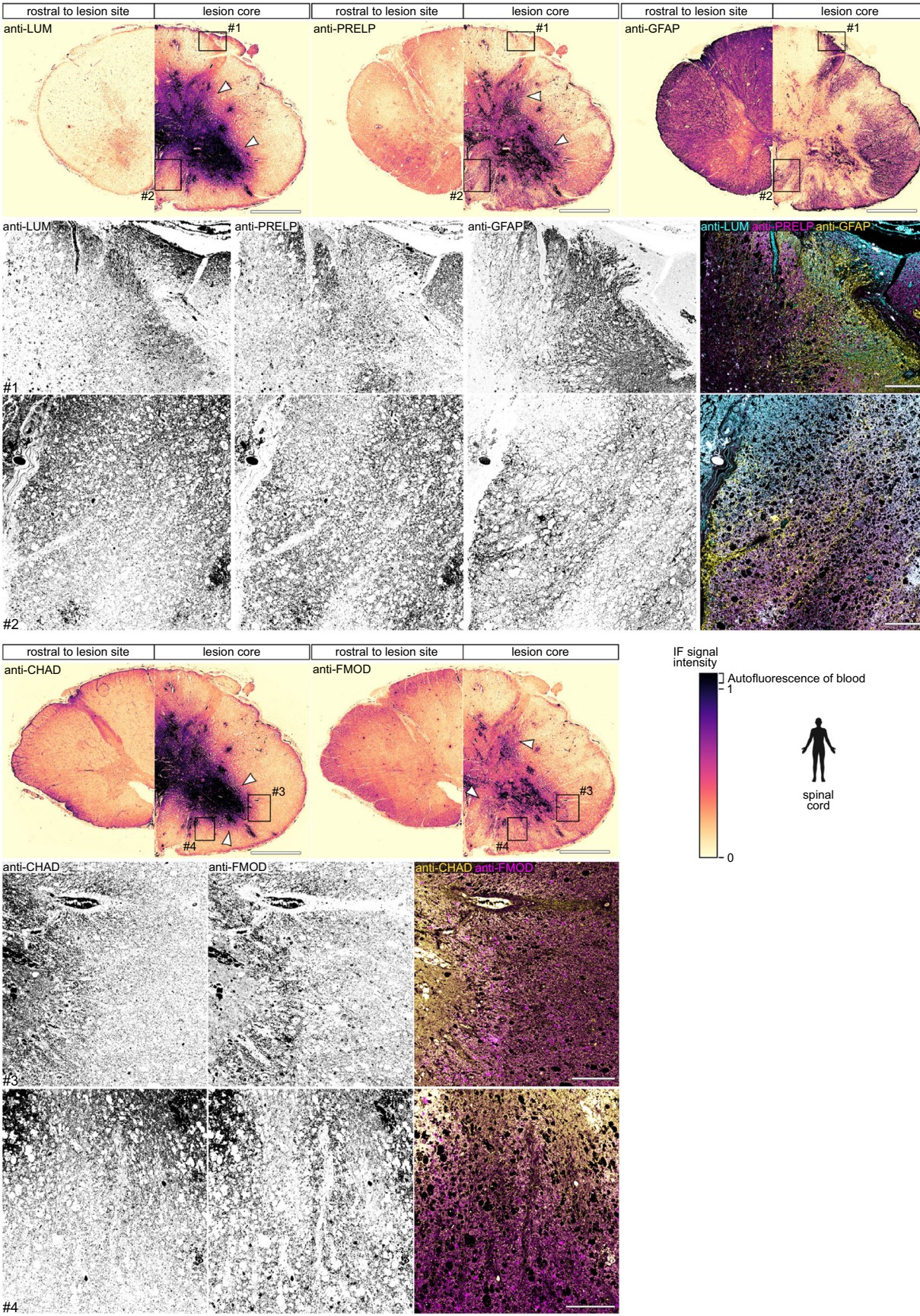

**Fig. 4 | SLRPs are enriched in human spinal cord lesions.** Anti-LUM, anti-PRELP, anti-CHAD, and anti-FMOD immunoreactivity is increased (arrowheads; see calibration bar of lookup table (LUT)) in the epicenter of the injured human spinal cord at nine days-post injury, as compared to rostral control segments of the same patient. Note that immunoreactivity is mainly observed in proximity to the hemorrhage (black; see calibration bar of LUT). Also note that with the exception of blood-derived autofluorescence, individual channels show distinct pattern of immunofluorescence (IF) signal (insets). Shown are transversal sections (dorsal is up). Similar results were obtained for anti-LUM in five out of six cases, for anti-PRELP and anti-FMOD in four out of six cases, and anti-CHAD in two out of six cases (Supplementary Table 3). Scale bars: 2 mm, 200 μm (insets). The human icon in the panel was created using BioRender.

abbreviated as *Xla.Tubb*:SLRP) led to robust and largely confined fluorescent labeling of the brain and spinal cord (Fig. 6b, c; Supplementary Fig. 5c). Comparing the fluorescence pattern of SLRP-mCherry fusion proteins in *Xla.Tubb*:SLRP transgenics to that of cytoplasmic DsRed proteins in *Xla.Tubb*:DsRed transgenic animals revealed prominent mCherry signal also in DsRed-negative regions of the spinal cord, including the central canal and its ependymal lining (Supplementary Fig. 5d). Moreover, primary cell culture from dissociated *Xla.Tubb*:SLRP;*elavl3*:GFP-F transgenic animals showed perineuronal mCherry fluorescence (Supplementary Fig. 5e). These data support that neuronal SLRP-mCherry fusion proteins are secreted into the extracellular space. At 1 dpl, we detected negligible mCherry fluorescence in the lesion core of *Xla.Tubb*:SLRP transgenic animals, which is consistent with the acute neuronal loss and secondary cell death following complete spinal cord transection (Fig. 6c, Supplementary Fig. 5b)[21,43]. Thus, cell type-specific manipulations allow us to distinguish in vivo between direct interactions of SLRPs with neurons and indirect effects on axon growth through modulation of the non-neural lesion environment. We found that targeting *chad*, *fmoda*, *lum*, or *prelp* expression specifically to neurons did not impair axon regeneration, as determined by measuring the axonal bridge thickness in live *elavl3*:GFP-F transgenic animals at 2 dpl (Fig. 6d; Supplementary Fig. 5f).

To corroborate these findings, we seeded adult primary murine dorsal root ganglion (DRG) neurons on laminin-coated glass coverslips and assessed neurite outgrowth in the presence of soluble or substrate-bound SLRPs (Fig. 6e–g). These 2D in vitro assays allow to test primarily for direct biochemical (ligand-receptor) interactions of SLRPs with the axonal growth cone. Potential indirect effects of SLRPs on axon regeneration through modulating inflammation, ECM composition, ECM assembly, or mechanical ECM properties, can be excluded. Although substrate-bound high molecular weight CSPGs potently reduced the average neurite length by 66%, a mixture of human CHAD, FMOD, LUM, and PRELP proteins had no effect (Fig. 6f). Similarly, average neurite length was not inhibited when SLRP proteins were supplemented to the growth medium but decreased by 29% in the presence of soluble CSPGs (Fig. 6g). Taken together, in vivo and in vitro data support that SLRPs inhibit axon growth indirectly.

## SLRPs do not prevent inflammation resolution

To test whether the presence of SLRPs in the injury ECM delays inflammation resolution, we analyzed the clearance of neutrophils and expression of the proinflammatory cytokine *il1b*, both of which processes must be tightly controlled for successful axon regeneration[3,21]. In larval zebrafish, the influx of Mpx+ neutrophils peaks as early as two hours following SCI, after which cell numbers rapidly decrease in the lesion site[21]. We found that targeting Chad, Fmod, Lum, or Prelp to the injury ECM in *pdgfrb*:SLRP transgenic animals did not lead to a higher number of Mpx+ neutrophils in the lesion site at 1 dpl (Supplementary Fig. 6a). Consistent with this, transcript levels of neutrophil-derived *il1b* were not increased in the lesion site of *pdgfrb*:SLRP transgenic animals at 1 dpl (determined by qRT-PCR; Supplementary Fig. 6b)[21]. Altogether, this suggests that the inhibition of axon regeneration by SLRPs does not occur via the prevention of inflammation resolution.

## SLRPs do not alter the fibroblast response

Accumulation of *pdgfrb*+ fibroblasts in the lesion site and secretion of a growth-conducive ECM is critical for axon regeneration after SCI in the zebrafish[13,14]. Hence, we sought to determine whether SLRPs inhibit axon regeneration by altering the fibroblast response in *pdgfrb*:SLRP transgenic animals. We did not detect significant changes in the area coverage of *pdgfrb*+ fibroblasts in the lesion site at 1 dpl (analyzed in *pdgfrb*:GFP transgenics), indicating that their recruitment was largely unperturbed (Supplementary Fig. 7). This is consistent with our findings that *pdgfrb*+ cell-specific induction of *slrp-mcherry* fusions does not trigger apoptosis of *pdgfrb*+ cells (Supplementary Fig. 4e). We next analyzed the composition of the injury ECM in *pdgfrb*:SLRP transgenic animals at 1 dpl, using MS-based quantitative proteomics. We found that induction of either *chad*, *fmoda*, *lum*, or *prelp* in *pdgfrb*+ cells did not lead to significant changes (FDR < 0.1, |FC| ≥ 1.3) in the abundance of detected matrisome proteins as compared to their respective controls (Supplementary Fig. 8a–c). To further validate the MS results, we used ISH to evaluate the expression pattern and transcript levels of five genes, coding for matrisome proteins that showed the highest (yet non-significant) fold change in each of the four experimental conditions. Consistent with the MS analysis, ISH signals in the lesion site were comparable between *pdgfrb*:SLRP transgenics and their respective controls for all genes analyzed (Supplementary Fig. 8d). We thus conclude that targeting SLRPs to *pdgfrb*+ cells does not lead to major changes in the biochemical composition of the injury ECM.

## Fmod, Lum, and Prelp modify the structural properties of the lesion environment

The structure and mechanical properties of the ECM within CNS scars differ significantly from that of healthy tissue[4,9,11]. SLRPs play instructive and structural roles in ECM organization and assembly to control the strength and biomechanical properties of tissues[44,45]. We thus hypothesized that SLRPs alter the structural and mechanical properties of the injury ECM, thereby making it adverse to axon regeneration. In order to assess this in vivo, we first utilized cross-polarized optical coherence tomography (CP-OCT). CP-OCT reports on relative changes in the polarization of incident light and can provide additional contrast to the native tissue based on its structural differences, including ECM structure[46,47]. Consistent with the changes in tissue structure which occur after CNS injury, the co-polarization ratio (ratio of preserved polarization to total reflectivity) prominently increased in the zebrafish spinal lesion site at 1 dpl, as compared to adjacent uninjured trunk tissue (Fig. 7a, b). Targeting Chad to the injury ECM (analyzed in *pdgfrb*:SLRP transgenics) did not alter the co-polarization ratio in the lesion site at 1 dpl, as compared to controls (Fig. 7b). By contrast, an increase in the co-polarization ratio was observed when Fmoda, Lum, or Prelp were targeted to the injury ECM (control$^{Fmoda}$: 0.709 ± 0.008, Fmoda: 0.750 ± 0.006; control$^{Lum}$: 0.702 ± 0.010, Lum: 0.733 ± 0.005; control$^{Prelp}$: 0.713 ± 0.006, Prelp: 0.738 ± 0.004; Fig. 7b). Hence, the presence of Fmod, Lum, and Prelp in the injury ECM coincides with alterations in the structural properties of the lesion environment and impaired axonal regrowth after SCI.

## Fmod, Lum, and Prelp modify the mechanical properties of the lesion environment

We next explored whether the observed structural alterations relate to changes in the mechanical properties of the lesion environment. To test this in vivo, we utilized confocal Brillouin microscopy (BM), a non-invasive, label-free, and all-optical method for assessing viscoelastic properties of biological samples in three dimensions[48–50]. BM measures an inelastic scattering process of incident light (photons) by density fluctuations (acoustic phonons) in the sample, called Brillouin

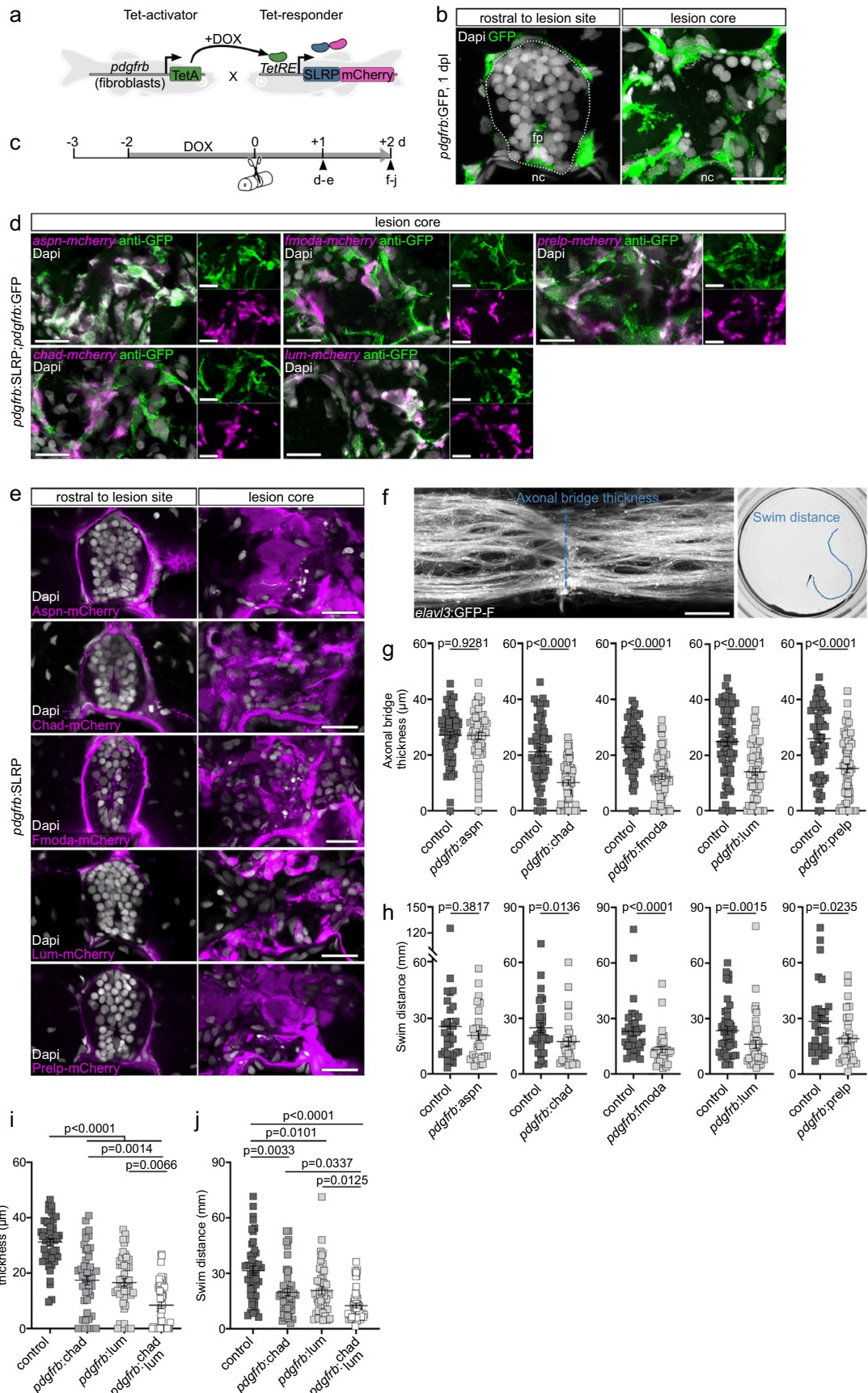

**Fig. 5 | SLRPs are inhibitory to CNS axon regeneration. a** Strategy to achieve *pdgfrb*⁺ fibroblast-specific expression of *slrp-mCherry* fusions using the TetON system. **b** *pdgfrb*⁺ fibroblasts (green) accumulate in the lesion core (*n* ≥ 10 animals)[13]. Dashed line indicates boundaries of the intact spinal cord. **c** Schematic timeline of the experimental design using the TetON system. **d** In the lesion core, mRNA expression of indicated *slrp-mCherry* fusions (magenta) is selectively induced in *pdgfrb*⁺ fibroblasts (green) in *pdgfrb*:TetA;*TetRE*:SLRP-mCherry (*pdgfrb*:SLRP);*pdgfrb*:GFP transgenic zebrafish. Similar results were obtained in *n* ≥ 10 animals for each experimental condition. **e** *pdgfrb*⁺ fibroblast-specific induction of indicated *slrp-mCherry* fusions leads to increased mCherry fluorescence (magenta) in the lesion core. Similar results were obtained in *n* ≥ 10 animals for each experimental condition. **f–h** *pdgfrb*⁺ fibroblast-specific induction of *chad*, *fmoda*, *lum*, and *prelp* but not *aspn* reduces the thickness of the axonal bridge (**f, g**) and impairs recovery of swim distance (**f, h**). Two-tailed Mann-Whitney test (**g**: Aspn, Chad; **h**), Two-tailed Student's *t*-test (**g**: Fmoda, Lum, Prelp). $n_{control}$ = 59, $n_{Aspn}$ = 61; $n_{control}$ = 59, $n_{Chad}$ = 60; $n_{control}$ = 63, $n_{Fmoda}$ = 64; $n_{control}$ = 60, $n_{Lum}$ = 60;

$n_{control}$ = 60, $n_{Prelp}$ = 60 animals over four independent experiments (**g**). $n_{control}$ = 30, $n_{Aspn}$ = 30; $n_{control}$ = 30, $n_{Chad}$ = 39; $n_{control}$ = 34, $n_{Fmoda}$ = 32; $n_{control}$ = 43, $n_{Lum}$ = 43; $n_{control}$ = 31, $n_{Prelp}$ = 31 animals (**h**). Image of the axonal bridge shown in **f** is a maximum intensity projection of the center of the spinal lesion site (lateral view; rostral is left). Note that neuronal somata are largely absent from the lesion core due to the acute mechanical trauma of the spinal cord transection. **i–j** Combinatorial induction of *chad* and *lum* in *pdgfrb*⁺ fibroblasts enhances the inhibitory effect of individual SLRPs on axon regeneration (**i**) and recovery of swim distance (**j**). Kruskal-Wallis test followed by Dunn' multiple comparison. $n_{control}$ = 45, $n_{Chad}$ = 48; $n_{Lum}$ = 48; $n_{Chad+Lum}$ = 48 animals over three independent experiments (**i**). $n_{control}$ = 45, $n_{Chad}$ = 45; $n_{Lum}$ = 45; $n_{Chad+Lum}$ = 45 (**j**). **a–j** Each data point represents one animal. Data are means ± SEM. Images shown in **b**, **d**, and **e** are transversal views of the unlesioned trunk or lesion core (dorsal is up). Scale bars: 20 μm. d days, dpl days post-lesion, DOX doxycycline, fp floor plate, nc notochord. Source data are provided as a Source Data file.

scattering. The energy transfer between the photons and acoustic phonons occurring during the scattering process can be quantified as the Brillouin frequency shift ($v_B$), which depends on the material's elastic properties, denoted by the longitudinal modulus ($M'$). $v_B$ is proportional to the square root of $M'$, which describes a material's elastic deformability under a distinct type of mechanical loading (i.e., longitudinal compressibility). Notably, $v_B$ has been shown in vivo to be sensitive to changes in the mechanical properties of the ECM during both physiological and pathological processes[51–53]. We acquired Brillouin images of the region in the spinal lesion site through which the regenerating axons preferentially grow at 1 dpl (Fig. 7c). Targeting Chad or Fmoda to the injury ECM (analyzed in *pdgfrb*:SLRP transgenics) did not alter $v_B$ of the lesion site (Fig. 7c). By contrast, a decrease of $v_B$ was observed when Lum or Prelp were targeted to the injury ECM (control$^{Lum}$: (5.324 ± 0.004) GHz, Lum: (5.298 ± 0.006) GHz; control$^{Prelp}$: (5.321 ± 0.005) GHz, Prelp: (5.295 ± 0.005) GHz; Fig. 7c). This suggests that Lum and Prelp increase the tissue compressibility of the local microenvironment in the spinal lesion site. $v_B$ depends not only on $M'$ but also on the refractive index ($n$) and density ($\rho$) of the sample in the focal volume (Supplementary Note 1). To exclude the possibility that the observed changes in $v_B$ are solely due to differences in $n$, we quantified its local distribution in the lesion site in vivo using optical diffraction tomography (ODT) (Supplementary Fig. 9). This revealed $\bar{n} = 1.3628 ± 0.0005$ in *pdgfrb*:lum transgenics, and $\bar{n} = 1.3636 ± 0.0005$ in control animals. In *pdgfrb*:prelp transgenics, we measured $\bar{n} = 1.3635 ± 0.0005$ and $\bar{n} = 1.3621 ± 0.0003$ for their respective control animals. To estimate the impact of the uncertainties in $\bar{n}$ and $\bar{v}_B$ on $M'$ in our experimental framework, we employed Gaussian propagation of uncertainty (Supplementary Note 2). This showed that the uncertainty in $n$ is overall contributing less to the propagated uncertainty in $M'$ than the one of $v_B$ for the values reported above. We conclude that the differences in $M'$ between control and *pdgfrb*:SLRP transgenic animals stem primarily from the observed changes in $v_B$ rather than $n$, including respective uncertainties. Hence, the longitudinal moduli with respective propagated uncertainties of $n$, $v_B$, refractive index increment of the dry mass content ($\alpha$), and the partial specific volume of the dry mass content ($\theta$) for the different conditionals are calculated to be: $M'(\text{control}^{Lum}) = (2.396 ± 0.006)$ GPa, $M'(\text{Lum}) = (2.373 ± 0.007)$ GPa, $M'(\text{control}^{Prelp}) = (2.394 ± 0.006)$ GPa, and $M'(\text{Prelp}) = (2.370 ± 0.006)$ GPa. Thus, targeting Lum or Prelp to the injury ECM leads to changes in $M'$ of the lesion environment.

We next performed atomic force microscopy (AFM)-based nanoindentation measurements on acute, living tissue preparations to quantify the viscoelastic properties (Young's modulus $E$, viscosity $\eta$) of the lesion core in the presence and absence of SLRP proteins (analyzed in *pdgfrb*:SLRPs transgenics; Fig. 7d). Applying two independent fitting models to the recorded force-indentation curves (Kelvin–Voigt—Maxwell, Hertz), we found that targeting Fmoda, Lum, or Prelp to the

injury ECM reduced the apparent Young's modulus by 44−66% at 2 dpl (Fig. 7e; Supplementary Fig. 10). Furthermore, the apparent viscosity was reduced by 23−25% in *pdgfrb:fmoda* and *pdgfrb:prelp* transgenic animals as compared to controls (Fig. 7f). Thus, targeting Fmoda, Lum, or Prelp to injury ECM leads to changes in the viscoelastic response of the lesion environment.

Altogether, these data show that SLRPs alter the mechanical properties of the injury ECM and inhibit axon regeneration after SCI.

## Discussion

Zebrafish regenerate severed axons and recover locomotor function after CNS injury, in sharp contrast to adult mammals including humans. Here, we demonstrate that differences in injury-associated ECM deposits account for the high regenerative capacity of zebrafish. We show that zebrafish establish a favorable composition of the injury ECM, which is characterized by the low abundance of inhibitory molecules rather than the presence of species-specific axon growth-promoting factors. Our study characterizes the SLRPs Chad, Fmod, Lum, and Prelp as such inhibitory molecules that govern the permissiveness of the injury ECM for regeneration. Increasing the abundance of any one of these proteins in the zebrafish injury ECM is sufficient to impair the capacity of severed axons to regrow across CNS lesions. This identifies SLRPs as potent inhibitors of CNS axon regeneration in vivo. Furthermore, we demonstrate that CHAD, FMOD, LUM, and PRELP are enriched in the injured human brain and spinal cord. Reducing the abundance of these SLRPs in the injury ECM or attenuating their activity may therefore offer a therapeutic strategy to enhance the permissiveness of CNS lesions for axon growth. Current evidence suggests a context-dependent cellular origin of SLRPs in the mammalian CNS, including neurons (ref. 54, this study), astrocytes (this study), and fibroblast lineage cells[55–57]. Hence, future studies will need to precisely map the cellular source of SLRPs in CNS lesions.

So far, much attention has been focused on components of the injury ECM that inhibit regeneration through direct interactions with the axonal growth cone[4,5]. However, mechanisms beyond growth cone collapse, repellence, and entrapment are emerging. For example, CSPGs impair inflammation resolution, thereby contributing to a lesion environment hostile to axon growth[3]. Here, we propose a new class of inhibitors of CNS axon regeneration which may function in part through modifying the structural and mechanical properties of the lesion environment. Our functional experiments support that SLRPs neither inhibit regeneration through i) direct interactions with the axonal growth cone, nor ii) prevention of inflammation resolution, nor iii) altering the composition of the fibroblast-derived injury ECM. Although we cannot rule out the possibility of SLRPs influencing the availability of neurotrophic factors in the lesion environment due to sensitivity limitations of the mass spectrometry analysis, our in vivo experiments provide evidence against this scenario. Specifically,

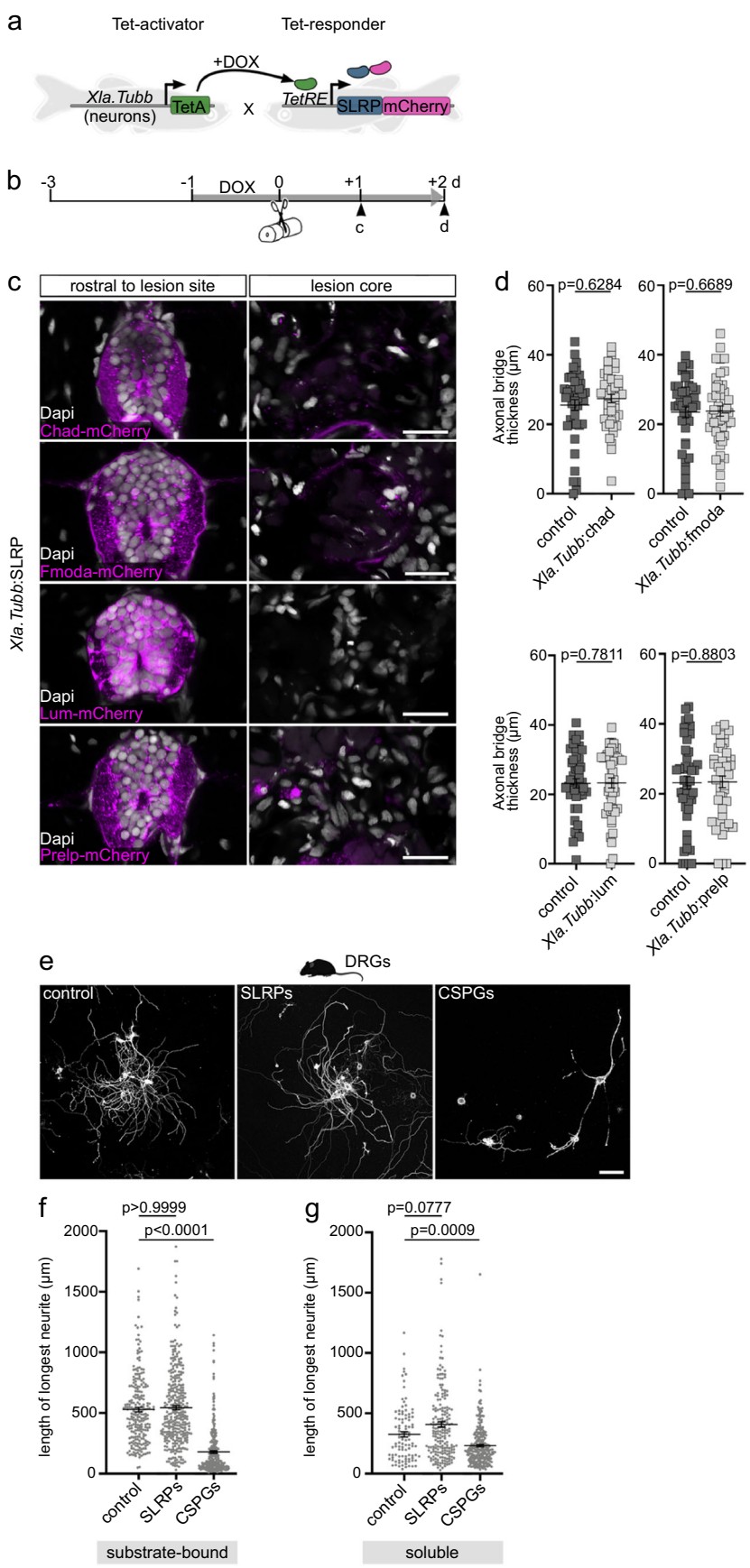

**Fig. 6 | SLRPs do not directly act on neurons to inhibit axon regeneration.**
**a** Strategy to achieve neuron-specific expression of *slrp-mCherry* fusions using the TetON system. **b** Schematic timeline of the experimental design using the TetON system. **c** Neuron-specific induction of indicated *slrp-mCherry* fusions in *Xla.Tubb*:TetA;*TetRE*:SLRP-mCherry (short *Xla.Tubb*:SLRP) transgenic zebrafish leads to negligible mCherry fluorescence (magenta) in the lesion core. Note that the lesion core lacks neuronal somata. Also note that strong mCherry fluorescence (magenta) is detectable in intact spinal cord tissue rostral to the lesion core. The same results were obtained in $n \geq 10$ animals for each experimental condition. Images shown are transversal views of the unlesioned trunk or lesion site (dorsal is up). **d** Neuron-specific induction of *chad*, *fmoda*, *lum*, or *prelp* in *Xla.Tubb*:SLRP transgenic zebrafish does not reduce the thickness of the axonal bridge. Each data point represents one animal. Two-tailed Mann-Whitney test. $n_{control} = 45$, $n_{Chad} = 45$; $n_{control} = 45$, $n_{Fmoda} = 45$; $n_{control} = 47$, $n_{Lum} = 45$; $n_{control} = 46$, $n_{Prelp} = 44$ animals over

three independent experiments. **e**–**g** Neurite outgrowth of primary adult murine dorsal root ganglion (DRG) neurons is not reduced in the presence of human SLRP proteins (mixture of CHAD, FMOD, LUM, PRELP), as compared to controls. Neurite growth is significantly reduced in the presence of CSPGs. **f** DRGs were cultured on glass coverslips with substrate-bound laminin, laminin and SLRPs, or laminin and CSPGs. **g** DRGs were cultured on laminin-coated glass coverslips and growth medium was supplemented with either SLRPs, CSPGs, or solvent control. Each data point represents one DRG neuron. Kruskal-Wallis test followed by Dunn' multiple comparison. $n_{control} = 229$, $n_{SLRPs} = 321$, $n_{CSPGs} = 315$ (**f**); $n_{control} = 100$, $n_{SLRPs} = 194$, $n_{CSPGs} = 266$ (**g**). The experiment was repeated once. **a**–**g** Data are means ± SEM. Scale bars: 100 μm (**e**), 20 μm (**c**). d days, dpl days post-lesion, DOX doxycycline. The mouse icon in the panel was created using BioRender. Source data are provided as a Source Data file.

neuron-specific induction of *chad*, *fmod*, *lum*, or *prelp* expression did not impact axonal regrowth. Utilizing AFM-based nanoindentation measurements, intravital CP-OCT and Brillouin microscopy, we show that the presence of individual SLRP family members in the injury ECM leads to changes in the structural (Fmod, Lum, Prelp) and mechanical (Lum, Prelp: reduced longitudinal modulus; Fmod, Lum, Prelp: reduced apparent Young's modulus; Fmod, Prelp: reduced apparent viscosity) properties of the lesion environment, which coincides with an impaired regenerative capacity of axons. This indicates that a single ECM factor can direct CNS repair toward inhibitory scarring by altering the mechano-structural properties of the local microenvironment.

Previous studies have shown that CNS injuries in mammals are accompanied by an increased compliance of the tissue, whereas the spinal cord of zebrafish transiently stiffens (increased apparent Young's modulus)[9,11,58]. This work provides in vivo evidence for a direct relationship between tissue mechanics (longitudinal modulus, apparent Young's modulus, apparent viscosity) and regenerative outcome upon CNS injury. Furthermore, we propose SLRPs as a causal factor of the differential response in zebrafish and humans. LUM has previously been implicated in the modulation of local tissue stiffness during the induction of fold formation in the developing human neocortex, further supporting a role of SLRPs in CNS tissue mechanics[59]. While the apparent Young's modulus of the injury ECM was reduced in the presence of Fmoda, Lum, or Prelp proteins, effective changes in the longitudinal modulus or apparent viscosity were only observed in a subset of these SLRPs. This suggests that different SLRP family members have specific functions in modulating the mechanical properties of CNS lesions. Interestingly, targeting Chad to the injury ECM did not yield effective changes in ECM structure and mechanics. Hence, the mechanism through which Chad inhibits CNS regeneration remains to be determined.

In conclusion, our data establish the composition, structural organization, and mechanical properties of the ECM as critical determinants of regenerative success after CNS injury. These findings reveal targets to make CNS lesions more conducive to axon growth in mammals, in which regeneration fails.

## Methods
### Human tissue collection and ethical compliance
This study was approved by the Ethics Committee of the Friedrich-Alexander-University (FAU) Erlangen-Nürnberg, Germany (Refs.#18–193_1-Bio, 193_18B, 92_14B; brain samples) and the Ethics Council of the Max Planck Society (Ref.#2021_40; spinal cord samples) and was conducted in accordance with the Declaration of Helsinki.

For brain samples, informed and written consent was obtained from all patients, their parents, or legal representatives if underage. We reviewed clinical and histological data of individuals who underwent surgery for the treatment of their focal drug-resistant epilepsy and were diagnosed with a scar, i.e., extensively transformed fibrotic and gliotic brain tissue. *En bloc* resections were carried out and tissue was

dissected into 5 mm-thick slices along the anterior-posterior axis. Tissue samples were fixed overnight in 4% formalin and routinely processed into liquid paraffin. Paraffin-embedded tissue was sectioned in coronal orientation at a thickness of 4 μm. Six patients with histologically proven scarred brain tissue following traumatic brain injury (TBI, $n = 3$), or repeated surgery (re-OP, $n = 3$) were selected for further investigation. Furthermore, a reference group including no-seizure autopsy controls ($n = 2$) and focal epilepsy patients with a cortical malformation as the primary lesion but no fibrous scarring of resected tissue upon visual inspection ($n = 4$) was also analyzed. In addition to routine hematoxylin and eosin (H&E) staining, immunohistochemical examination of all surgical brain specimens was performed using the following panel of antibodies: mouse monoclonal anti-NeuN (clone A-60, Millipore Cat#MAB377; RRID:AB_2298772, 1:1000), mouse monoclonal anti-GFAP (clone 6F2, Dako Cat#M0761; RRID:AB_2109952, 1:800), and recombinant rabbit monoclonal anti-LUM (lumican, Invitrogen Cat#MA5-29402; RRID:AB_2785270, 1:500-1:600). The samples were subsequently digitized using a NanoZoomer Hamamatsu S60 digital slide scanner.

Human spinal cord injury (SCI) samples, control samples, and related clinical and neuropathological information were obtained from the International Spinal Cord Injury Biobank (ISCIB; Vancouver, Canada). The Clinical Research Ethics Board of the University of Columbia (Vancouver, Canada) granted the permission for post-mortem spinal cord acquisition and for sharing biospecimens. Spinal cord biospecimens were collected from consented participants or their next-of-kin and provided as paraffin-embedded tissue sections at a thickness of 5 μm.

Supplementary Tables 1 and 2 detail the clinical and neuropathological data of subjects included in this study.

### Zebrafish husbandry, transgenic lines, and ethical compliance
All zebrafish experimental procedures were in accordance with institutional and internationally recognized guidelines and were approved by the Regierung von Unterfranken (Government of Lower Franconia, Würzburg, Germany; RUF 55.2.2-2532.2-1120-15) and the BRFAA Animal Ethics Committee and the Attica Veterinary Department (no: 534930/20, no: 534935/20, no: 1189597, 236/03-04-2023 for EL 25 BIOexp 03).

All zebrafish lines were kept and raised under a 14/10 h light/dark cycle and according to FELASA recommendations[60]. We used AB and WIK wild-type strains of zebrafish (*Danio rerio*) and the following transgenic zebrafish lines: BAC(*pdgfrb*:Gal4ff)[ncv24 61], *UAS*:EGFP[zf82 62], *UAS-E1b*:Eco.NfsB-mCherry[c264 63], *pdgfrb*:TetA AmCyan[mps7 13], *Xla.Tubb*:TetA AmCyan[ue103 14], *Xla.Tubb*:DsRed[zf148 42], *TetRE*:lum-mCherry[mps3 13], and *her4.3*:GFP-F[mps9]. *elavl3*:GFP-F[mps10], *TetRE*:aspn-mCherry[mps11], *TetRE*:chad-mCherry[mps12], *TetRE*:fmoda-mCherry[mps13], and *TetRE*:prelp-mCherry[mps14] transgenic zebrafish lines were established using the DNA constructs and methodology described below.

Combinations of different transgenic zebrafish lines used in this study were abbreviated as follows:

*pdgfrb*:aspn (*pdgfrb*:TetA AmCyan;*TetRE*:aspn-mCherry),

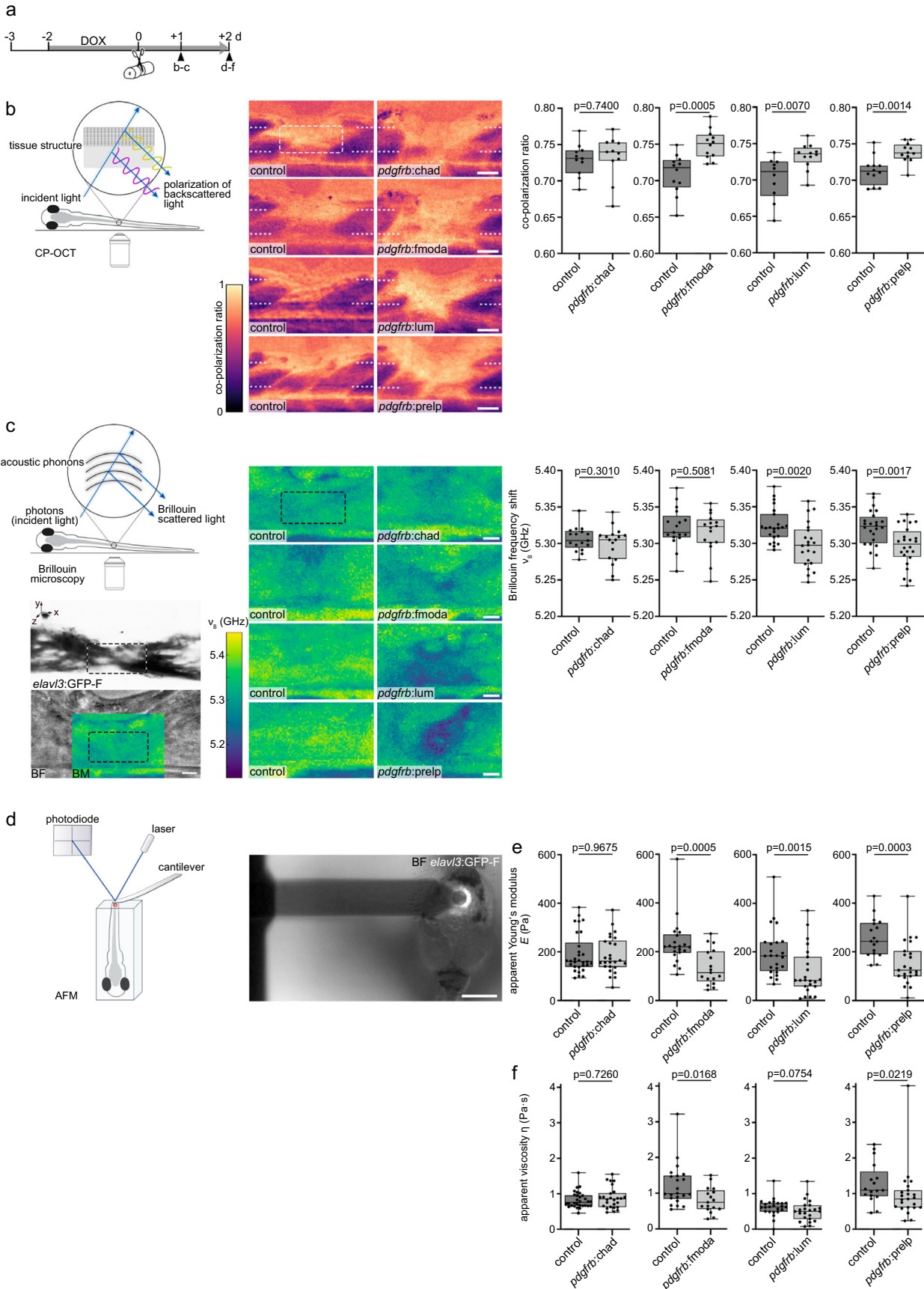

**Fig. 7 | SLRPs modulate the structural and mechanical properties of the lesion environment. a** Timeline for experimental treatments shown in **b**–**f**. **b** Targeting Fmoda, Lum, or Prelp to the injury ECM in *pdgfrb*:SLRP transgenic zebrafish increases the co-polarization ratio in the spinal lesion site, as determined by cross-polarized optical coherence tomography (CP-OCT). Images shown are average intensity projections of the lesion site (lateral view; rostral is left). The dashed lines indicate the location of the severed spinal cord, the dashed rectangle indicates the region of quantification. Two-tailed Student's *t*-test. $n_{control} = 11$, $n_{Chad} = 11$; $n_{contro} = 12$, $n_{Fmoda} = 12$; $n_{contro} = 10$, $n_{Lum} = 12$; $n_{control} = 12$, $n_{Prelp} = 12$ animals. **c** Targeting Lum or Prelp to the injury ECM in *pdgfrb*:SLRP transgenic zebrafish decreases the Brillouin frequency shift ($v_B$) in the spinal lesion site, as determined by Brillouin microscopy. Images shown are sagittal optical sections (brightfield intensity, confocal fluorescence, or Brillouin frequency shift map) through the center of the lesion site of an *elavl3*:GFP-F transgenic zebrafish (lateral view; rostral is left). The dashed rectangle indicates the region of quantification. Two-tailed

Student's *t*-test. $n_{control} = 17$, $n_{Chad} = 17$; $n_{control} = 17$, $n_{Fmoda} = 16$; $n_{control} = 23$, $n_{Lum} = 22$; $n_{control} = 24$, $n_{Prelp} = 24$ animals over three (Chad, Fmoda) or four (Lum, Prelp) independent experiments. **d**–**f** Targeting Fmoda, Lum, or Prelp to the injury ECM in *pdgfrb*:SLRP transgenic zebrafish decreases the apparent Young's modulus (*E*) (**e**; Fmoda, Lum, Prelp) and apparent viscosity (η) (**f**; Fmoda, Prelp) in the spinal lesion site, as determined by atomic force microscopy-based nanoindentation measurements (AFM). Recorded force-indentation curves were fitted to the Kelvin–Voigt–Maxwell model. Two-tailed Mann-Whitney test. $n_{control} = 30$, $n_{Chad} = 26$; $n_{control} = 22$, $n_{Fmoda} = 17$; $n_{control} = 24$, $n_{Lum} = 23$; $n_{control} = 17$, $n_{Prelp} = 23$ animals over three (Chad, Fmoda, Prelp) or four (Lum) independent experiments. **a**–**f** Each data point represents one animal. Box plots show the median, first, and third quartile. Whiskers indicate the minimum and maximum values. Scale bars: 500 μm (**d**), 50 μm (**b**), 10 μm (**c**). BF brightfield intensity, BM Brillouin frequency shift map, d days, dpl days post-lesion, DOX doxycycline. Source data are provided as a Source Data file.

*pdgfrb*:chad (*pdgfrb*:TetA AmCyan;*TetRE*:chad-mCherry),
*pdgfrb*:fmoda (*pdgfrb*:TetA AmCyan;*TetRE*:fmoda-mCherry),
*pdgfrb*:lum (*pdgfrb*:TetA AmCyan;*TetRE*:lum-mCherry),
*pdgfrb*:prelp (*pdgfrb*:TetA AmCyan;*TetRE*:prelp-mCherry),
*Xla.Tubb*:aspn (*Xla.Tubb*:TetA AmCyan;*TetRE*:aspn-mCherry),
*Xla.Tubb*:chad (*Xla.Tubb*:TetA AmCyan;*TetRE*:chad-mCherry),
*Xla.Tubb*:fmoda (*Xla.Tubb*:TetA AmCyan;*TetRE*:fmoda-mCherry),
*Xla.Tubb*:lum (*Xla.Tubb*:TetA AmCyan;*TetRE*:lum-mCherry),
*Xla.Tubb*:prelp (*Xla.Tubb*:TetA AmCyan;*TetRE*:prelp-mCherry),
*pdgfrb*:GFP (BAC(*pdgfrb*:Gal4ff);*UAS*:EGFP),
*pdgfrb*:NTR-mCherry (BAC(*pdgfrb*:Gal4ff);*UAS-E1b*:Eco.NfsB-mCherry).

For live microscopy, embryos were treated with 0.00375% 1-phenyl-2-thiourea (PTU, Sigma-Aldrich Cat#P7629), beginning at 24 hpf (hours post-fertilization), to prevent pigmentation.

### Generation of transgenic zebrafish lines
All primer sequences for molecular cloning are given in Supplementary Data 1, plasmid maps and sequences are given in Supplementary Data 2 to 6. To create the donor plasmid for *elavl3*:GFP-F transgenic zebrafish, the sequence coding for the membrane-localized GFP (EGFP fused to farnesylation signal from c-HA-Ras) was amplified from the pEGFP-F vector (Clonetech) using primer pair #1, and cloned downstream of the zebrafish *elavl3* promoter (Addgene plasmid Cat#59530)[64]. To create the donor plasmids for generation of *TetRE*:aspn-mCherry, *TetRE*:chad-mCherry, *TetRE*:fmoda-mCherry, and *TetRE*:prelp-mCherry transgenic fish, the sequences coding for zebrafish *aspn* (ENSDART00000064798.6), *chad* (ENSDART00000066264.4), *fmoda* (ENSDART00000065985.5), and *prelp* (ENSDART00000155521.2) were amplified from cDNA of developing zebrafish (primer pairs #2 to #5), fused to mCherry at the C-terminus and cloned downstream of the tetracycline operator sequence as described[65,66]. Generation of *elavl3*:GFP-F, *TetRE*:aspn-mCherry, *TetRE*:chad-mCherry, *TetRE*:fmoda-mCherry, and *TetRE*:prelp-mCherry transgenic zebrafish lines was achieved by injection of 35 pg of the respective donor plasmid together with in vitro synthesized capped sense mRNA of the Tol2 transposase into 1-cell embryos[67].

Further information and requests for unique biological reagents should be directed to and will be fulfilled by the corresponding author.

### Drug treatments
Drug treatments were performed according to the schematic timelines shown with each experiment. Doxycycline (DOX; Sigma-Aldrich Cat#D9891) was dissolved in reverse osmosis $H_2O$ at 50 mg mL$^{-1}$ and used at a final concentration of 25 μg mL$^{-1}$. Metronidazole (MTZ; Sigma-Aldrich Cat#M3761) was dissolved in DMSO (VWR Cat#WN182) at 800 mM stock concentration and used at a final concentration of 2 mM.

### Zebrafish spinal cord lesions and behavioral recovery
A detailed protocol for inducing spinal cord lesions in zebrafish larvae has been previously described[68]. Briefly, zebrafish larvae (3 dpf) were anesthetized in E3 medium containing 0.02% MS-222 (PharmaQ Cat# Tricaine PharmaQ). A 30 G x ½" hypodermic needle was used to transect the spinal cord by either incision or perforation at the level of the urogenital pore. After surgery, larvae were returned to E3 medium for recovery and kept at 28.5 °C. Larvae that had undergone extensive damage to the notochord were excluded from further analysis. For lesions, the experimenter was blinded to the experimental treatment. Larvae used for lesions were randomly taken from Petri dishes containing up to 50 animals, however, no formal randomization method was used.

Analysis of behavioral recovery after spinal cord transection in larval zebrafish was performed as previously described[18], using Etho-Vision XT software (Noldus, version 16.0.1536). Behavioral data are shown as the distance traveled within 10 s after touch, averaged for triplicate measures per larva.

Adult spinal cord lesions were performed as described previously[69]. Briefly, zebrafish (>3 months post-fertilization) were anesthetized by immersion in 0.02% MS-222 in PBS. A longitudinal incision was made at the side of the fish to expose the vertebral column. The spinal cord was transected under visual control, 5 mm caudal to the brainstem-spinal cord junction.

### Adult mice spinal cord lesions, daily care, and ethical compliance
All mice experimental procedures were conducted in accordance with the standards and rules of KU Leuven and Belgian national regulations (protocol no: P122-2017). Mice were group housed, and all mice were kept on a 12/12 h light/dark cycle in a facility where temperature was kept at 22 ± 2 °C and ~50% humidity. For spinal cord lesions, adult mice of both sexes, aged between one and two month were used. Mice were maintained on a mixed genetic background (129/C57Bl6). All surgical procedures were performed under aseptic conditions and under full general anesthesia with isoflurane in oxygen-enriched air (1-2%). Adult mice underwent a mid-dorsal skin incision to access the spinal cord and muscles were separated along the midline to expose the vertebral column. Laminectomy was performed using forceps at spinal segment ~T10, and the spinal cord was completely transected using micro scissors. After surgery, mice were placed in a heat chamber until fully awake. Analgesia (Metacam, Boehringer Ingelheim, 0.5 mg mL$^{-1}$) was provided after surgery and antibiotics (Cefazoline, Sandoz, 5 mg kg$^{-1}$) were provided for 3 days post-surgery. Mice were housed with a 12/12 h light/dark cycle with ad libitum access to food and water, and cared for twice daily with manual bladder voiding. Complete transections were confirmed post-mortem by visual inspection of the spinal cord. At one month after injuy, all mice were deeply anesthetized with ketamine

(100 mg kg$^{-1}$, *i.p.*) and xylazine (5 mg kg$^{-1}$, *i.p.*) and perfused with 4% PFA. All collected tissue was cryoprotected in 30% sucrose in PBS.

## Image acquisition and processing

Images were acquired using the systems described in each subsection. Images were processed using ImageJ (http://rsb.info.nih.gov/ij/; version 2.3.0/1.53 f), Adobe Photoshop CC (version 22.3.1), and Zeiss ZEN blue software (version 3.1 blue edition). Figures were assembled using Adobe Photoshop CC.

## Live imaging of larval zebrafish

For live confocal imaging, zebrafish larvae were anesthetized in E3 medium containing 0.02% MS-222 and mounted in the appropriate orientation in 1% low melting point agarose (Ultra-Pure™ Low Melting Point, Invitrogen Cat#16520) between two microscope cover glasses. During imaging, larvae were covered with 0.01% MS-222-containing E3 medium to keep preparations from drying out. Imaging was done using a Plan-Apochromat 10x/0.45 M27 objective, Plan-Apochromat 20x/0.8 objective, and C-Apochromat 40x/1.2 W Korr UV-VIS-IR objective on a Zeiss LSM 980 confocal microscope.

## Sectioning of zebrafish

Adult zebrafish were terminally anesthetized by immersion in 0.1% MS-222 in PBS, transcardially fixed by injection of 4% paraformaldehyde (PFA; Thermo Fisher Scientific Cat#28908) solution (methanol-free) in PBS into the vascular system, and post-fixed in 4% PFA overnight at 4 °C. Cryoprotection and sectioning were performed as previously described[13].

Terminally anesthetized zebrafish larvae were fixed in 4% PFA in PBS for 1 h at room temperature or for overnight at 4 °C. After two washes in PBT (0.1% Tween-20 in PBS), the larvae were embedded in 4% agarose or 4% agar in PBS. Sections of 100 μm thickness were obtained using a vibratome (Leica, VT1200S). The sections were then processed for in situ hybridization, as described below, or immediately counterstained with DAPI (Thermo Fisher Scientific Cat#62248) to visualize nuclei and mounted in 75% glycerol. Images were acquired using a Plan-Apochromat 20x/0.8 and C-Apochromat 40x/1.2 W Korr UV469 VIS-IR objective on a Zeiss LSM 980 confocal microscope.

## Zebrafish primary cell culture

Larvae with neuron-specific expression of membrane-tethered GFP (*elavl3*:GFP-F) and *slrp-mCherry* fusions (*Xla.Tubb*:SLRP-mCherry) were dissociated in DMEM (Thermo Fisher Cat#31885023) containing 0.08% trypsin (Thermo Fisher Cat#25200072), 625 μg ml$^{-1}$ collagenase P (Roche Cat#11213865001), and 17.5 μM dispase II (Sigma Aldrich Cat#D4693). Cells were resuspended in MEM (Thermo Fisher Cat#11095-080) supplemented with 1% Neuropan2 (PanBiotech Cat#P07-11010), 1 mM sodium pyruvate (Thermo Fisher Cat#11360070), 0.6% glucose (Merck Cat#G8270), 0.22% sodium bicarbonate (Sigma Aldrich Cat#S5761), 2 mM L-Glutamine (Gibco Cat#25030149), and 25 μg ml$^{-1}$ doxycycline, and seeded in 8-well glass bottom plates (ibidi Cat# 80827) pre-coated with Collagen I (Merck Cat#08115). Cells were grown at 5% CO$_2$ and 28 °C for 24 h and imaged using a C-Apochromat 40x/1.2 W Korr UV469 VIS-IR objective on a Zeiss LSM 980 confocal microscope.

## Tissue dissociation and FACS

GFP$^+$ and GFP$^-$ cells were isolated from *pdgfrb*:GFP transgenic zebrafish at 4 dpf. 120 animals were pooled for each experiment. Before tissue dissociation, head, yolk sac, and tail fin were removed, using micro scissors. Dissociation, live cell staining, and FACS sorting of cells were done as previously described[13]. Briefly, excised tissue was enzymatically dissociated using 0.25% Trypsin-EDTA (Gibco Cat#25200072) followed by mechanical dissociation using a fire-polished glass Pasteur pipette. Cell suspension was filtered through a

20 μm cell strainer (pluriSelect Cat#43-10020-40). Following centrifugation for 10 min at 300 g, cells were resuspended in 500 μL HBSS medium (Gibco Cat#14065-056). To stain for viable cells, Calcein Blue (Invitrogen Cat#C1429) was added to the cell suspension. Cells were directly sorted into lysis buffer using a MoFlo Astrios EQ sorter (Beckman Coulter). For the detection of GFP, a 488 nm excitation laser and a 526/52 nm bandpass filter were used. Calcein Blue was detected following 405 nm excitation using a 431/28 nm bandpass emission filter.

## Quantitative RT-PCR (qRT-PCR)

For qRT-PCR on isolated *pdgfrb$^+$* cells, total RNA was extracted from ~70,000 FACS-sorted GFP$^+$ cells of dissected trunks from unlesioned *pdgfrb*:GFP transgenic animals at 4 dpf, using Total RNA Purification Plus Micro Kit (Norgen Biotek Cat#48500). The total RNA extracted from an equal number of GFP$^-$ cells obtained from the same animals served as control.

For quantification of transcript levels in the lesion site at 1 dpl, trunk tissue of 75 animals spanning approximately three somites in length and containing the lesion site were isolated using micro scissors. Corresponding trunk tissue of unlesioned age-matched clutch mates served as control. Total RNA was extracted using TRIzol reagent (Invitrogen Cat#15596026).

Reverse transcription was performed with Maxima™ H Minus cDNA Synthesis Master Mix with dsDNase (Thermo Fisher Scientific Cat#1681), using a combination of oligo(dT) and random hexamer primers. qRT-PCR was performed at 60 °C using PowerUp™ SYBR™ Green Master Mix (Applied Biosystems Cat#A25918) on a StepOne-Plus™ Real-Time PCR system. Samples were run in triplicates, and expression levels were normalized to *actb1* control. Normalized relative mRNA levels were determined using the ΔΔCt method.

Primers were designed to span an exon−exon junction using Primer-BLAST software (https://ncbi.nlm.nih.gov/tools/primer-blast/). All primer sequences are given in Supplementary Data 1.

## Sample preparation for mass spectrometry (MS) analysis

For label-free MS-based quantitative proteomics, proteins were isolated from trunk tissue of 75 wild type (strain: AB) zebrafish larvae spanning approximately three somites in length and containing the lesion site (1 dpl, 2 dpl) or corresponding unlesioned age-matched trunk tissue (4 dpf, 5 dpf). To identify injury-induced changes in protein abundance, MS analysis was performed on three biological replicates for each condition. To identify changes in protein abundance caused by targeting SLRPs to *pdgfrb$^+$* cells, MS analysis was performed on four biological replicates for each condition. Isolation of proteins was done as previously described with minor modifications[70]. Proteins were extracted in three distinct fractions, and each fraction was analyzed separately by LC-MS/MS. Zebrafish tissue was homogenized in PBS for 10×30 s at high intensity and 10 × 30 s pauses in between, using the Bioruptor Plus sonication system (Diogenode Cat#UCD-300). After centrifugation at 16,000 g for 2 min, the supernatant containing the soluble proteins was collected (fraction 1). The pellet was resuspended in detergent-containing buffer (50 mM Tris (Sigma-Aldrich Cat#T1503), 5% glycerin (Roth Cat#4043.2), 500 mM NaCl (VWR Cat#27810.295), 1% NP40 (Fluka Cat#74385), 2% SDC (Sigma-Aldrich Cat#SER0046), 1% SDS (Roth Cat#4360.2), 1% DNAse I (made in house, ProteinProduction Facility), 1 mM MgCl$_2$ (Sigma-Aldrich Cat#M9272)), followed by incubation at 0 °C for 20 min and homogenization using the Bioruptor Plus sonication system. After centrifugation at 16,000 g for 2 min, the supernatant containing the detergent-soluble proteins (fraction 2) and the detergent-insoluble protein pellet (fraction 3) were collected. Proteins of fraction 1 and fraction 2 were precipitated by incubation with acetone at −20 °C for overnight. The pellets of all three fractions were dissolved in reduction and alkylation buffer (6 M guanidinium chloride (Merck

Cat#1.04220.1000), 100 mM Tris-HCl pH 8.5, 10 mM TCEP (Thermo Fisher Scientific Cat#77720), 50 mM CAA (Sigma-Aldrich Cat#C0267)), followed by incubation at 99 °C for 15 min and sonication in the Bioruptor Plus sonication system. The proteins were diluted 1:3 with urea buffer (4.5 mM urea (Sigma-Aldrich Cat#U1250), 10 mM Tris-HCl, 3% acetonitrile (VWR Cat#83640.320), 1 µg LysC (Wako Cat#129-02541)), and incubated at 37 °C for 3 h. Thereafter, the digestion mixture was diluted 1:3 with 10% acetonitrile in MS grade water (VWR Cat#83645.320), followed by sonication in the Bioruptor Plus sonication system and incubation with 1 µg LysC and 2 µg trypsin (Promega Cat#V511A) at 37 °C for overnight. Acetonitrile was removed using a SpeedVac (Christ Cat#RVC 2-25 together with CT 02-50), and peptides were further purified using in-house produced three plug SCX stage tips (Empore™ Cation Solid Phase Extraction Disks Cat#2251). After elution with 60 µL 5% ammonia solution (Roth Cat#5460.1) in 80% acetonitrile, samples were vacuum-dried in the SpeedVac.

## LC-MS/MS acquisition

Peptides were solubilized in 6 µL buffer A (100% MS-LC grade water and 0.1% formic acid (VWR Cat#84865.180)), and a total volume of 3 µL was loaded onto a 30 cm-long column (75 µm inner diameter (Polymicro Cat#TSP075375); packed in-house with ReproSil-Pur 120 C18-AQ 1.9 µm beads (Dr. Maisch GmbH Cat#r119.aq)) via the autosampler of the Thermo Scientific Easy-nLC 1200 (Thermo Fisher Scientific) at 60 °C. For the identification of injury-induced changes in protein abundance (experimental conditions: unlesioned 4 dpf, 1 dpl; unlesioned 5 dpf, 2 dpl), eluted peptides were directly sprayed onto the Q Exactive HF Orbitrap LC-MS/MS system (Thermo Fisher Scientific), using the nanoelectrospray interface. As a gradient, the following steps were programmed with increasing addition of buffer B (80% acetonitrile, 0.1% formic acid): linear increase from 30% over 120 min, followed by a linear increase to 60% over 10 min, followed by a linear increase to 95% over the next 5 min, and finally buffer B was maintained at 95% for another 5 min. The mass spectrometer was operated in a data-dependent mode with survey scans from 300 to 1750 m/z (resolution of 60000 at m/z = 200), and up to 15 of the top precursors were selected and fragmented using higher energy collisional dissociation (HCD with a normalized collision energy of value of 28). The MS2 spectra were recorded at a resolution of 15k (at m/z = 200). AGC target for MS1 and MS2 scans were set to 3E6 and 1E5, respectively, within a maximum injection time of 100 ms for MS1 and 25 ms for MS2. Dynamic exclusion was set to 16 ms. For identification of changes in protein abundance caused by overexpression of SLRPs, eluting peptides were directly sprayed onto the timsTOF Pro LC-MS/MS system (Bruker). As gradient, the following steps were programmed with increasing addition of buffer B (80% acetonitrile, 0.1% formic acid): linear increase from 5% to 25% over 75 min, followed by a linear increase to 35% over 30 min, followed by a linear increase to 58% over the next 5 min, followed by a linear increase to 95% over the next 5 min, and finally buffer B was maintained at 95% for another 5 min. Data acquisition on the timsTOF Pro was performed using timsControl. The mass spectrometer was operated in data-dependent PASEF mode with one survey TIMS-MS and ten PASEF MS/MS scans per acquisition cycle. Analysis was performed in a mass scan range from 100-1700 m/z and an ion mobility range from $1/K0 = 0.85$ Vs cm$^{-2}$ to 1.30 Vs cm$^{-2}$, using equal ion accumulation and ramp time in the dual TIMS analyzer of 100 ms each at a spectra rate of 9.43 Hz. Suitable precursor ions for MS/MS analysis were isolated in a window of 2 Th for m/z < 700 and 3 Th for m/z > 700 by rapidly switching the quadrupole position in sync with the elution of precursors from the TIMS device. The collision energy was lowered as a function of ion mobility, starting from 59 eV for $1/K0 = 1.6$ Vs cm$^{-2}$ to 20 eV for 0.6 Vs cm$^{-2}$. Collision energies were interpolated linearly between these two 1/K0 values and kept constant above or below these base points. Singly charged precursor ions were excluded with a polygon filter mask and further m/z and ion mobility information was used for dynamic exclusion to avoid re-sequencing of precursors

that reached a target value of 20,000 a.u. The ion mobility dimension was calibrated linearly using three ions from the Agilent ESI LC/MS tuning mix (m/z, 1/K0: 622.0289, 0.9848 Vs cm$^{-2}$; 922.0097 Vs cm$^{-2}$; 1.1895 Vs cm$^{-2}$; 1221.9906 Vs cm$^{-2}$, 1.3820 Vs cm$^{-2}$).

## Computational MS data analysis

Raw data were processed using the MaxQuant computational platform (versions 1.6.17.0 and 2.0.1.0)[71] with standard settings applied. Briefly, the peak list was searched against the zebrafish (*Danio rerio*) proteome database (SwissProt and TrEMBL, 46847 entries) with an allowed precursor mass deviation of 4.5 ppm and an allowed fragment mass deviation of 20 ppm. MaxQuant enables individual peptide mass tolerances by default, which was used in the search. Cysteine carbamidomethylation was set as static modification, and methionine oxidation and N-terminal acetylation as variable modifications. Proteins were quantified across samples using the label-free quantification algorithm in MaxQuant, generating label-free quantification (LFQ) intensities. The match-between-runs option was enabled.

## Bioinformatic analysis of proteomics data

LFQ intensity values were imported to Perseus software (version 1.6.14.0)[72]. Filters were set to exclude proteins identified by site, matching to the reverse database, or contaminants. For the identification of injury-induced changes in protein abundance (lesioned *vs.* unlesioned), proteins that exhibited ≥ 1 invalid value were excluded from further analysis. However, we included proteins that showed invalid values in all three replicates of one experimental condition (either lesioned or unlesioned). This was necessary to include proteins undetectable in unlesioned animals but with enrichment after SCI. For identification of changes in protein abundance caused by cell-type specific induction of SLRPs, all proteins that exhibited ≥ 1 invalid value were excluded. Proteins exhibiting significantly altered abundance between experimental and control samples were identified by permutation-based analysis[73] with FDR < 0.1, $s_0 = 0.1$, and |FC| ≥ 1.3. Principal component analysis plots and volcano plots were created using Perseus software.

## Reactome pathway analysis

To identify overrepresented biological pathways based on the observed protein abundance ratios, Reactome pathway analysis was performed using g:Profiler[74,75]. Proteins exhibiting significantly increased abundance in lesioned samples as compared to corresponding unlesioned age-matched control samples (FDR < 0.1, FC ≥ 1.3, $s_0 = 0.1$) were analyzed. Reactome pathways with an adjusted (Bonferroni) *P*-value ≤ 0.05 were considered significantly enriched.

## Heatmaps

The zebrafish matrisome was obtained from http://matrisome.org[19] and manually updated for missing UniProt identifiers. The updated matrisome list was imported to Perseus software and matched with the MS dataset to add missing UniProt identifiers, gene symbols, as well as further specifications, including division (core matrisome, matrisome-associated, and putative matrisome proteins) and protein symbol of mammalian orthologues. The list of matrisome proteins exhibiting significantly altered abundance between lesioned and corresponding unlesioned age-matched control samples (FDR < 0.1, |FC| ≥ 1.3, $s_0 = 0.1$) was extracted, and the Z-score was calculated. Heatmaps were created using Perseus software.

## Cross-species comparison of proteomics data

For cross-species comparison of proteomics data, a published dataset from adult Sprague-Dawley rats at seven days post-contusion spinal cord injury (T10 injury level) and uninjured control T10 spinal cord segments was re-analyzed[31,76]. LFQ intensity values of ECM-enriched 4 M guanidine spinal cord tissue extracts (https://data.mendeley.com/datasets/npkwh5vsss/1) were imported to Perseus software. Filters

were set to exclude proteins identified by site, matching the reverse database or contaminants. Additionally, proteins were excluded that did not exhibit valid values for all three replicates of at least one experimental condition (unlesioned, 7 dpl). Proteins exhibiting significantly altered abundance between lesioned and unlesioned control samples were identified by permutation-based analysis[73] with FDR < 0.1, $s_0 = 0.1$, and $|FC| \geq 1.3$. A pairwise comparison of the obtained protein lists using Venny 2.1 software (https://bioinfogp.cnb.csic.es/tools/venny) was performed to identify differentially regulated matrisome proteins between rat and zebrafish[77].

## In situ hybridization (ISH)

A detailed protocol for ISH on whole-mount zebrafish larvae with digoxygenin (DIG)-labeled antisense probes has been previously described[68]. In brief, terminally anesthetized larvae were fixed in 4% PFA (Thermo Fisher Scientific Cat#28908) in PBS and treated with proteinase K (Invitrogen Cat#25530-049) followed by re-fixation for 15 min in 4% PFA in PBS. DIG-labeled antisense probes were hybridized overnight at 65 °C. This protocol allows efficient probe penetration in whole-mount preparations of 5 dpf larvae[14]. Color reaction was performed after incubation with anti-DIG antibody conjugated to alkaline phosphatase (Sigma-Aldrich Cat#11093274910; RRID:AB_2313640, 1:3000) using NBT/BCIP substrate (Roche Cat#11697471001). Samples were mounted in 75% glycerol in PBS and imaged in multi-focus mode using a Leica M205 FCA stereo microscope equipped with a Leica DMC6200 C color camera. Information on ISH probes, including primer sequences used for molecular cloning, is provided in Supplementary Data 1. All ISH probes showed specific staining in developmental domains.

Fluorescence ISH and simultaneous detection of mCherry transcripts and GFP protein on floating sections of larval zebrafish was performed as previously described[14].

HCR RNA fluorescence ISH (Molecular Instruments) on adult zebrafish cryosections was performed according to the manufacturer's instructions. All genes were processed simultaneously. Sections comprising the lesion core or rostral control tissue of the same animal were stained on the same microscope slide. Sections were mounted in Fluoromount-G with DAPI (Invitrogen Cat#E132139) and imaged using a Plan-Apochromat 20x/0.8 objective on a Zeiss LSM 980 confocal microscope. Images of all genes were taken with the same settings. Information on HCR ISH probes is provided in Supplementary Data 1.

## Immunofluorescence (IF)

A detailed protocol for IF on whole-mount zebrafish larvae has been previously described[68]. Briefly, terminally anesthetized larvae were fixed in 4% PFA in PBS for 1 h at room temperature. After removing the head and tail using micro scissors, larvae were permeabilized by subsequent incubation in acetone and proteinase K. Samples were re-fixed in 4% PFA in PBS, blocked in PBS containing 1% Triton X-100 (Sigma-Aldrich Cat#T8787; PBTx) and 4% bovine serum albumin, and incubated over two to three nights with the primary antibody of interest. After several washes in PBTx, samples were incubated over two nights with the secondary antibody of interest. Thereafter, samples were washed in PBTx and mounted in 75% glycerol (Sigma-Aldrich Cat#G5516) in PBS. Imaging was performed using a Plan-Apochromat 20x/0.8 objective on a Zeiss LSM 980 confocal microscope.

For neurite outgrowth assay, dorsal root ganglion (DRG) neuron cultures were fixed in 4% PFA in PBS for 15 min at room temperature. After permeabilization with 0.1% PBTx for 20 min at 37 °C, DRG neurons were blocked with 5% goat serum (Sigma-Aldrich Cat#G9023) in PBS for 45 min at 37 °C. Neurites were labeled by subsequent incubation with anti-Tubulin β3 antibody in PBS for 2 h and fluorophore-conjugated secondary antibody in PBS for 1.5 h at 37 °C. DRG neurons were mounted in Fluoromount-G with DAPI and imaged using a Plan-Apochromat 20x/0.8 objective and digital camera system on a Zeiss LSM 980 microscope.

IF on mouse and human paraffin-embedded tissue sections was performed as previously described[78]. Briefly, tissue sections were deparaffinized in xylene (3x for 10 min each), stepwise hydrated by successive incubation in 100%, 96%, and 70% isopropanol for 5 min each, and rinsed in distilled water. For antigen retrieval, sections were microwaved in citrate buffer at 850 W for 3 min. The sections were then microwaved twice at 250 W for 10 min each. Sections were allowed to cool down on ice for 15 min. Sections were then rinsed in PBS for 15 min under gentle agitation. Sections were blocked in 0.1% PBTx containing 3% fetal calf serum (Gibco Cat#A4766801), and either 1% goat (Gibco #Cat16210-064) or donkey (Biowest #CatS2170) serum. Sections were then sequentially incubated with primary antibodies diluted in blocking solution for 1 h at 37 °C, and fluorophore-conjugated secondary antibodies for 1 h at room temperature. Imaging was performed using Plan-Apochromat 10x/0.45 M27 and Plan-Apochromat 20x/0.8 objectives on a Zeiss LSM 980 confocal microscope.

We used rabbit polyclonal anti-Collagen IV (Abcam Cat#ab6586; RRID:AB_305584, 1:250), mouse monoclonal anti-Tubulin β3 (TUBB3, Biolegend Cat#801201; RRID:AB_2313773, 1:500), rabbit polyclonal anti-Mpx (GeneTex Cat#GTX128379; RRID:AB_2885768, 1:300), mouse monoclonal anti-GFAP (clone GA5, Cell Signaling Technology Cat#mAB3670; RRID:AB_561049, 1:300), recombinant rabbit polyclonal anti-GFAP (clone RM1003, Abcam Cat#ab278054, 1:2000), rabbit monoclonal anti-LUM (Invitrogen Cat#MA5-29402; RRID:AB_2785270, 1:500-1:600), rabbit monoclonal anti-LUM (ThermoFisher Scientific Cat#MA5-34828; RRID: AB_2848736, 1:600), sheep polyclonal anti-PRELP (R&D Systems Cat#AF6447; RRID:AB_10889844, 1:400), rabbit polyclonal anti-PRELP (ThermoFisher Scientific Cat#PA5-78631; RRID:AB_2736456, 1:300), rabbit polyclonal anti-CHAD (Invitrogen Cat#PA-553761; RRID:AB_2639742, 1:300), mouse monoclonal anti-CHAD (clone 8B7, Thermo Fisher Scientific Cat#H00001101-M01; RRID:AB_530000, 1:300), and sheep polyclonal anti-FMOD (clone 549302, R&D Systems Cat#AF5945; RRID:AB_1964583, 1:600). Secondary fluorophore-conjugated antibodies were from Invitrogen.

## Whole-mount TUNEL/anti-GFP co-labeling

Terminally anesthetized larvae were fixed in 4% PFA in PBS for overnight at 4 °C. After removing the head and tail using micro scissors, larvae were permeabilized by subsequent incubation in acetone and proteinase K as described for whole-mount IF. Samples were re-fixed in 4% PFA in PBS and Click-iT TUNEL Alexa Fluor 647 Imaging Assay (Thermo Fisher Cat#C10247) was performed according to the manufacturer's protocol to label apoptotic cells. Briefly, samples were equilibrated in TdT reaction buffer for 30 min at room temperature, followed by incubation in TdT reaction cocktail for overnight at room temperature. The click-it reaction was performed for 60 min at room temperature. Thereafter, samples were blocked in PBTx containing 3% bovine serum albumin and incubated over three nights with chicken anti-GFP antibody (Abcam Cat#ab13790; RRID:AB_300798, 1:500). After several washes in PBTx, samples were incubated over two to three nights with the secondary antibody of interest. Thereafter, samples were washed in PBTx and mounted in 75% glycerol in PBS. Imaging was done using a Plan-Apochromat 10x/0.45 M27 objective on a Zeiss LSM 980 confocal microscope.

## Primary dorsal root ganglia (DRG) culture/neurite outgrowth assay

Handling of mice was performed in accordance with animal welfare laws, complied with ethical guidelines, and was approved by the responsible local committees and government bodies (University of Erlangen, Amt für Veterinärwesen der Stadt Erlangen, and the Regierung von Unterfranken). Adult female and male (4–6 months old) C57BL/6 J mice were sacrificed by cervical dislocation, and spinal cords were removed. DRG neurons were dissected from the spinal cord, incubated in Neurobasal medium (NB, Thermo Fisher Scientific

Cat#21103049) containing 2.5 mg mL$^{-1}$ collagenase P (Sigma-Aldrich Cat#11213857001) and maintained in an incubator for 1 h at 37 °C in 5% CO$_2$. The DRG tissue was homogenized by pipetting using finely fire-polished drawn glass pipettes. Dissociated DRG neurons were separated from axon stumps and myelin debris via a 14% bovine serum albumin layer (Sigma-Aldrich Cat#A9205) and centrifugation at 1 x g for 8 min. The pellet with DRG neurons was resuspended in NB containing 20 μL mL$^{-1}$ B27 supplement 50x (Gibco Cat#17504044), 2 mM Glutamax (ThermoFisher Cat#35050-061), 10 μg mL$^{-1}$ antibiotic-antimycotic (Gibco Cat#15240062), and 0.01 μg mL$^{-1}$ nerve growth factor (ThermoFisher Cat#3257-019). Isolated DRG neurons were seeded on glass coverslips pre-coated with 0.1 mg mL$^{-1}$ Poly-D-lysine (PDL, Gibco Cat#A3890401), followed by coating with either 10 μg mL$^{-1}$ laminin (ThermoFisher Cat#23017015), a mix of 10 μg mL$^{-1}$ laminin and 5 μg mL$^{-1}$ of each SLRP protein (recombinant human CHAD (R&D Systems Cat#8218-CH), FMOD (R&D Systems Cat#9840-FM), LUM (Abcam Cat#ab221400), and PRELP (R&D Systems Cat#6447-PR)), or a mix of 10 μg mL$^{-1}$ laminin and 5 μg mL$^{-1}$ CSPGs (Sigma-Aldrich Cat#CC117). For coating of glass coverslips, a standard protocol was employed that is predicted to apply a monomolecular layer of 3-5 nm thickness[79]. Alternatively, DRG neurons were seeded on glass coverslips pre-coated with PDL and laminin and cultured in the presence of either 1 μg mL$^{-1}$ CSPGs, a mix of 1 μg mL$^{-1}$ of each SLRP protein, or solvent control. All recombinant SLRP proteins used were produced in mammalian expression systems. DRG cultures were incubated for 48 h at 37 °C and 5% CO$_2$ to allow neurite outgrowth.

### Cross-polarized optical coherence tomography (CP-OCT)
For in vivo CP-OCT[47], zebrafish larvae were anesthetized in E3 medium containing 0.015% MS-222 and mounted in a lateral position in 1% low gelling temperature agarose (Sigma-Aldrich Cat#A0701) between two microscope cover glasses. During imaging, larvae were covered with 0.01% MS-222-containing E3 medium to keep preparations from drying out. Cross-polarized images of the spinal lesion site were acquired using a custom-built CP-OCT system[80]. Light from a broadband supercontinuum laser (YSL Photonics Cat#SC-OEM) was filtered to acquire a spectrum centered at 885 nm with a full width at half maximum of 80 nm. The laser was operated at 200 MHz. The filtered spectrum from the laser was coupled to a single-mode optical fiber and collimated using a collimator (Thorlabs Cat#F230APC-850). The light was further split using a 90/10 beam splitter (Thorlabs Cat#BS025) into a reference beam and a sample beam. The sample was illuminated with 15 mW of optical power. A combination of a quarter-wave plate (Thorlabs Cat#SAQWP05M-1700) and a lens (Thorlabs Cat#AC254-030-AB) was inserted in the reference and the sample arm to control the polarization of the light. A galvano mirror (Thorlabs Cat#GVS012) was used to scan the laser beam over the sample. The reflected reference and the sample signals were acquired using a custom-designed spectrometer consisting of a reflective collimator (Thorlabs Cat#RC08APC), a holographic grating (Wasatch Photonics Cat#1200 l/mm@840 nm), a lens (Thorlabs Cat#AC-254-080-B), and a line scan camera (Basler Cat#2048 pixels, ral2048-48gm) operating at 25 kHz line scan rate. The spectrometer signal was processed using LabVIEW-based custom-designed software (National Instruments). The acquired interference spectrum from the camera was spectrally recalibrated from wavelength space to wavenumber space[81] and a fast Fourier transform was performed to obtain an axial profile of the sample.

### Combined confocal fluorescence and Brillouin microscopy (BM)
For in vivo BM, zebrafish larvae were anesthetized in E3 medium containing 0.015% MS-222 and mounted in a lateral position in 1% low gelling temperature agarose on a 35 mm glass-bottom dish (ibidi Cat#81158). During imaging, larvae were covered with 0.015% MS-222-containing E3 medium to keep preparations from drying out. Brillouin frequency shift images were acquired by BM employing a confocal

configuration and a Brillouin spectrometer consisting of a two-stage virtually imaged phase array (VIPA) etalon, as previously described in detail elsewhere[53]. Briefly, the sample was illuminated by a frequency-modulated diode laser beam ($\lambda$ = 780.24 nm, DLC TA PRO 780, Toptica), which was stabilized to the D$_2$ transition of rubidium $^{85}$Rb. A Fabry-Perot interferometer in two-pass configuration and a monochromatic grating (Toptica) were employed to further suppress the contribution of amplified spontaneous emission to the laser spectrum. The laser light was coupled into a single-mode fiber and guided into the backside port of a commercial inverted microscope stand (Axio Observer 7, Zeiss). An objective lens (20x, NA = 0.5, EC Plan-Neofluar, Zeiss) illuminated the sample on a motorized stage with an optical focus. The laser power at the sample plane was set at 15 mW. The backscattered light from the sample was collected by the same objective lens, coupled into the second single-mode fiber to achieve confocality, and delivered to a Brillouin spectrometer. In the Brillouin spectrometer, the backscattered light was collimated and passed through a molecular absorption cell filled with rubidium $^{85}$Rb (Precision Glassblowing Cat#TG-ABRB-I85-Q), in which the intensity of the Rayleigh scattered and reflected light was significantly suppressed. After passing through the molecular absorption cell, the beam was guided to two VIPA etalons (Light Machinery Cat#OP-6721-6743-4) with the free spectral range of 15.2 GHz, which convert the frequency shift of the light into the angular dispersion in the Brillouin spectrum. The Brillouin spectrum was acquired by a sCMOS camera (Teledyne Cat#Prime BSI), with the exposure time of 0.5 s per measurement point. The two-dimensional Brillouin frequency map of the injured region was measured by scanning the motorized stage on the microscope stand, with the translational step size of 0.5 μm. The Brillouin microscope was controlled with custom acquisition software written in C++ (https://github.com/BrillouinMicroscopy/BrillouinAcquisition; version 0.3.4). Confocal fluorescence imaging was performed in the same region of interest (ROI) as Brillouin measurement using a re-scan confocal microscopy (RCM) module (RCM2, Confocal.nl) that was attached to one side port of the microscope stand. The RCM module consists of a sCMOS camera (Prime BSI Express, Teledyne) and a multi-line laser unit (Skyra, Cobolt) as an excitation illumination source for four laser lines ($\lambda$ = 405 nm, 488 nm, 562 nm, and 637 nm). A pinhole of the diameter of 50 μm and the re-scanning imaging principle[82] provides rapid confocal fluorescence imaging with a lateral resolution of 120 nm. By focusing a confocal fluorescence image in the center of the spinal cord of *elavl3*:GFP-F transgenic zebrafish larvae, the axial plane of Brillouin imaging was determined.

### Optical diffraction tomography (ODT)
For in vivo ODT, zebrafish larvae were anesthetized in E3 medium containing 0.015% MS-222 and mounted in a lateral position in 1% low gelling temperature agarose on a 35 mm glass-bottom dish (ibidi Cat#81158). During imaging, larvae were covered with E3 medium containing 0.015% MS-222 and 20% of refractive index (RI)-matching agent Iodaxanol (OptiPrep™; Sigma-Aldrich Cat#D1556)[83]. Iodaxanol was used to reduce the RI difference between the zebrafish larvae and the surrounding medium. The final RI of the medium was 1.351, which was determined by an ABBE refractometer (Kern & Sohn GmbH Cat#ORT1RS). The RI distribution of the zebrafish spinal lesion site was measured by ODT employing Mach-Zehnder interferometry to measure multiple complex optical fields from various incident angles, as previously described[84]. A solid-state laser beam ($\lambda$ = 532 nm, 50 mW, CNI Optoelectronics Technology Co.) was split into two paths using a beamsplitter. One beam was used as a reference beam and the other beam illuminated the sample on the stage of an inverted microscope (Axio Observer 7, Carl Zeiss AG) through a tube lens ($f$ = 175 mm) and a water-dipping objective lens (40x, NA = 1.0, Carl Zeiss AG). A high numerical aperture objective lens (40x, water immersion, NA = 1.2, Carl Zeiss AG) collected the beam diffracted by the sample. To reconstruct a 3D RI tomogram of the sample, the sample was illuminated from 150 different incident angles scanned by a

dual-axis galvano mirror (Thorlabs Cat#GVS212/M) located at the conjugate plane of the sample. The diffracted beam interfered with the reference beam at an image plane and generated a spatially modulated hologram, which was recorded with a CMOS camera (XIMEA Cat#MQ042MG-CM-TG). The field-of-view of the camera covers 205.0 µm × 205.0 µm. The complex optical fields of light scattered by the samples were retrieved from the recorded holograms by applying a Fourier transform-based field retrieval algorithm[85]. The 3D RI distribution of the samples was reconstructed from the retrieved complex optical fields via the Fourier diffraction theorem, employing the first-order Rytov approximation[86,87]. A more detailed description of tomogram reconstruction can be found elsewhere[88]. The MATLAB script for ODT reconstruction can be found at https://github.com/OpticalDiffractionTomography/ODT_Reconstruction (version 1.0.0).

## Atomic force microscopy-based nanoindentation measurements (AFM)

For AFM measurements, zebrafish larvae were terminally anesthetized by immersion in 0.1% MS-222 in PBS and mounted in an upright position in 3% low gelling temperature agarose. The trunk tissue was cut transversely using a vibratome to expose the lesion core. The agarose block containing the larva was glued onto a plastic tissue culture dish. During measurements, larvae were covered with 0.015% MS-222 containing PBS to keep preparations from drying out. AFM measurements were performed using a Nanowizard 4 (JPK BioAFM, Bruker Nano GmbH). For indentation measurements, 37 µm diameter polystyrene beads (microParticles #CatPS/Q-F-B878) were glued to tipless cantilevers (NanoWorld #CatARROW-TL1-50) with two-component adhesive (Araldite Rapid #Cat3131.1). Cantilevers (accurate spring constant ranged between 0.027–0.039 N m$^{-1}$) were calibrated prior to the measurements using the thermal noise method[89]. The indentation force was 12 nN, and the indentation speed was 7 µm s$^{-1}$. The ROI for indentation was identified by fluorescence signal from the rostral spinal cord stumps in *elavl3*:GFP-F transgenics using an upright Axio Zoom.V16 stereo microscope.

## Quantifications and statistics

Unless otherwise indicated, controls refer to DOX-treated clutch mates of wild type, single transgenic Tet-activator or single transgenic Tet-responder genotype. For quantifications, investigators were blinded to the experimental conditions.

For quantification of axonal bridge thickness, transgenic live animals (*elavl3*:GFP-F) were imaged using a Plan-Apochromat 20x/0.8 objective on a Zeiss LSM 980 confocal microscope. The length of a vertical line that covers the width of the axonal bridge at the center of the lesion site, was then determined using ImageJ software (https://imagej.nih.gov/ij/index.html). The quantification of the glial bridge thickness was done in *her4.3*:GFP-F transgenics and analogously to the axonal bridge measurements. Measurements of axonal bridge thickness were performed in samples of three independent experiments. Spinal cord lesions for one out of the three experimental replicates of axonal bridge measurements were induced by a second operator.

Quantification of fluorescence signal in the lesion site was performed on captured images of whole-mount samples using ImageJ software, following previously published protocols[14,21]. Confocal image stacks were collapsed before analysis. For the quantification of area coverage of *pdgfrb*:GFP$^+$ in the lesion site, collapsed confocal stacks were converted to a binary image using the automated mean of grey levels thresholding function of ImageJ software. A pre-set ROI of constant size was applied to all images of the same experiment and the number of pixels determined (pixel area). The ROI was placed in the center of the lesion site and ventrally limited by the notochord.

Quantification of *pdgfrb*:GFP$^+$/TUNEL$^+$ cells was performed in a pre-set ROI of constant size on optical sections of a confocal stack using ImageJ software. The ROI was ventrally limited by the dorsal/caudal artery. *pdgfrb*:GFP$^+$/TUNEL$^+$ cells in the floor plate or median fin fold were excluded from analysis based on their location and morphology. Quantification of *Mpx*$^+$ cells in the lesion site was performed in a pre-set ROI of constant size on optical sections of a confocal stack using ImageJ software. The ROI was ventrally limited by the notochord.

The neurite length of adult murine DRG neurons was quantified using the semi-automated SNT Fiji-ImageJ plugin[90]. The longest neurite of each DRG neuron was identified and measured.

Extraction and quantification of Brillouin frequency shifts were performed in a pre-set ROI, using custom software written in Python (https://github.com/GuckLab/impose, version 0.4.1; https://github.com/BrillouinMicroscopy/BMicro, version0.8.1). The ROI was 40 µm x 20 µm, which corresponds to the determined average axonal bridge thickness and the average distance between the spinal cord stumps following a dorsal incision lesion at 1 dpl. The ROI was placed in the center of the lesion site and ventrally limited 5 µm dorsal to the notochord. The ROI was applied to all images. Brillouin frequency shifts were measured in samples of at least three independent experiments. The measured Brillouin frequency shift $\nu_B$ can be expressed in terms of the longitudinal modulus $M'$, refractive index (RI) $n$, and density $\rho$ of the specimen, as well as the incident wavelength $\lambda_0$ and scattering angle $\vartheta$ given by the setup: $\nu_B = \frac{2n\sqrt{M'}}{\lambda_0 \sqrt{\rho}} \sin\frac{\vartheta}{2}$. All measurements were performed in the backscattering configuration with $\vartheta = 180°$ and, accordingly $\sin\frac{\vartheta}{2} = 1$. The longitudinal compressibility $\kappa_L$ may be expressed as the inverse of the longitudinal modulus $\kappa_L = \frac{1}{M'}$. For the sake of conciseness $\kappa_L$ will be referred to as 'compressibility' in the rest of the text, despite the fact that this term is usually defined as the inverse of the bulk modulus $K'$. Bulk modulus $K'$, shear modulus $G'$, and longitudinal modulus $M'$ are related by the following equation in the case of an isotropic sample: $M' = K' + \frac{4}{3} G'$.

To quantify the co-polarization ratio (ratio of preserved polarization to total reflectivity) in the lesion site, two images of the tissue were acquired; one co-polarized and one cross-polarized image. For the acquisition of the co-polarized image, the angle of the quarter-wave plate (QWP) axis in the reference path was set in such a way that the light in the sample and the reference arm had the same polarization. To acquire the cross-polarized image, the QWP in the reference beam path was rotated by 45°, resulting in orthogonal polarization between reference and sample arm. Sample reflectivity $R(z)$ and co-polarization ratio $\delta(z)$ of the sample was calculated using the amplitude of the co- ($A_{co}$) and cross- ($A_{cross}$) polarized images using the following equations: $R(z) = \sqrt{A_{co}^2 + A_{cross}^2}$ and $\delta(z) = \frac{A_{co}}{A_{co} + A_{cross}}$. Using ImageJ software, the co-polarization ratio was determined in a pre-set ROI of constant size for the averaged intensity projection of the complete image stack. The ROI was placed in the center of the lesion site and ventrally limited by the notochord. A second independent observer validated the results.

Extraction and quantification of the RI and mass density distribution from reconstructed tomograms were performed using custom-written MATLAB (MathWorks, version R2018b) scripts. The mass density of the samples was calculated directly from the reconstructed RI tomograms, since the RI of the samples, $n(x,y,z)$, is linearly proportional to the mass density of the material, $\rho(x,y,z)$, as $n(x,y,z) = n_m + \alpha\rho(x,y,z)$, where $n_m$ is the RI value of the surrounding medium and $\alpha$ is the RI increment ($dn/dc$), with $\alpha = 0.1919$ mL g$^{-1}$ for proteins and nucleic acids[91,92]. RI and mass density distribution from reconstructed tomograms were evaluated in a pre-set ROI in a sagittal slice of the RI tomogram along the $x$-$y$ plane at the focused plane, using a custom-written MATLAB script. The ROI was the same as applied to Brillouin and ODT images. Note that the RI was measured in samples independent of those analyzed with BM.

AFM force-indentation analysis was performed using the open-source analysis software PyJibe (https://github.com/afm-analysis/

pyjibe; version 0.14.0). The Poisson's ratio was set to 0.5 for all analyses. We applied two different fitting models: (i) the Hertz model for a spherical indenter[58] with a limitation of the indentation curve to 3 µm, and (ii) the Kelvin–Voigt–Maxwell (KVM) model[93], in order to analyze the apparent Young's modulus and apparent viscosity.

Unless indicated, no data were excluded from analyses. Except for determining effect sizes and Gaussian propagation of uncertainty, all statistical analyses were performed using Graph Pad Prism 9 (GraphPad Software Inc., version 9.0.0). Mathematica software (Wolfram Research Inc., version 12.2) was used to calculate the Gaussian propagation of uncertainty. All quantitative data were tested for normal distribution using Shapiro-Wilk test. Parametric and non-parametric tests were used as appropriate. All statistical tests used in each experiment are given in the figure legends or in Supplementary Data 7. We used two-tailed Student's $t$-test, two-tailed Mann-Whitney test, paired two-tailed Student's $t$-test, and Kruskal-Wallis test followed by Dunn's test for multiple comparisons. Differences were considered statistically significant at $P$-values below 0.05 and effect sizes (Cohen's d for Student's $t$-test, common language effect size $\theta$ for Mann-Whitney and Dunn's multiple comparison tests) including the significance boundary ($d_c = 1$, $\theta_c = 0.5$) within their uncertainty. The $P$-value, effect size, and respective uncertainty for each experimental group is given in the figures or Supplementary Data 7. Respective effect sizes were determined according to[94] and[95] using custom scripts in Mathematica software. Unless indicated, variance for all group's data is presented as ± standard error of the mean (SEM). The sample size ($n$) for each experimental group is given in the figure legends or in Supplementary Data 7. Graphs were generated using GraphPad Prism 9.

### Reporting summary
Further information on research design is available in the Nature Portfolio Reporting Summary linked to this article.

## Data availability
Except for the proteomics data, all data are available in the main text or the supplementary materials. The shotgun zebrafish MS data generated in this study have been deposited to the ProteomeXchange Consortium (http://proteomecentral.proteomexchange.org) via the PRIDE partner repository with the dataset identifiers PXD037605 and PXD037590. The shotgun rat proteomic data used in this study[31] are available in the Mendeley database under the accession code 10.17632/npkwh5vsss.1 (https://data.mendeley.com/datasets/npkwh5vsss/1). Source data are provided with this paper.

## Code availability
All codes used in this study are available on GitHub: https://github.com/BrillouinMicroscopy/BrillouinAcquisition (version 0.3.4)[96]; https://github.com/GuckLab/impose (version 0.4.1)[97]; https://github.com/BrillouinMicroscopy/BMicro (version 0.8.1)[98]; https://github.com/OpticalDiffractionTomography/ODT_Reconstruction (version 1.0.0)[99]; https://github.com/afm-analysis/pyjibe (version 0.14.0)[100].

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

## Acknowledgements

The authors gratefully acknowledge the International Spinal Cord Injury Biobank (ISCIB) for generously providing the human spinal cord specimens used in this project, Dr. Johann Helmut Brandstätter (Biology Department, FAU Erlangen-Nürnberg) for providing mice, and Dr. Caterina Becker for sharing *her4.3*:GFP-F transgenic fish. We thank Casandra Cecilia Carrillo Mendez and Olga Stelmakh for excellent fish care, Dr. Jona Kayser and Dr. Catherine Xu for comments on the manuscript. The authors acknowledge financial support from the Else Kröner-Fresenius-Stiftung (to K.Ko.), the Hellenic Foundation for Research and Innovation (H.F.R.I – 7903 grant to V.T.), the Deutsche Forschungsgemeinschaft (project number 460333672 – CRC1540 Exploring Brain Mechanics (subproject B03, B05, and C03) to J.G., S.M., D.W., and K.Ko.; FR2758/3-1 to R.F.; GU612/8-1 to J.G.; project number 527729149 to D.W.), the Max Planck Society (to J.G.), the Research Foundation – Flanders (FWO Research Grant G097818N to A.T.), the GSKE/FMRE Queen Elizabeth Medical Foundation for Neuroscience (Young Investigator Grant to A.T.), and the Wings for Life Spinal Cord Research Foundation (grant 03371AYTA to A.T.). Mouse, rat, and human icons were created using BioRender.

## Author contributions

Conceptualization: D.W.; Formal analysis: J.K., K.K., A.P., C.M., G.S., K.S., S.A., D.W.; Investigation: J.K., V.T., N.J., K.K., C.M., O.L., S.M., G.R., V.K., A.P., G.S., K.Ka.; T.B., R.F., A.W., N. K., B.S., K.S., K.Ko., D.W.; Methodology: J.K., K.K., S.M., A.W., N.K., B.S., K.S., D.W.; Project administration: D.W.; Resources: A.T., D.B., I.B., B.S., K.S., J.G., K.Ko. D.W.; Software: K.K., P.M., R.S.; Supervision: K.Ko., D.W.; Visualization: J.K., D.W.; Writing—original draft: J.K., D.W.; Writing—review and editing: J.K., N.J., K.K., C.M., T.B., D.B., A.T., I.B., J.G., K.Ko., D.W.

## Funding

## Competing interests

The authors declare no competing interests.

## Additional information

[1]Max Planck Institute for the Science of Light, 91058 Erlangen, Germany. [2]Max-Planck-Zentrum für Physik und Medizin, 91058 Erlangen, Germany. [3]Department of Biology, Animal Physiology, Friedrich-Alexander-University Erlangen-Nürnberg, 91058 Erlangen, Germany. [4]Experimental Surgery, Clinical and Translational Research Center, Biomedical Research Foundation Academy of Athens, 11527 Athens, Greece. [5]Center of Basic Research, Biomedical Research Foundation, Academy of Athens, 11527 Athens, Greece. [6]Department of Physics, Friedrich-Alexander-University Erlangen-Nürnberg, 91058 Erlangen, Germany. [7]Department of Neuropathology, Universitätsklinikum Erlangen, Friedrich-Alexander-University Erlangen-Nürnberg, 91054 Erlangen, Germany. [8]Department of Medicine 1, Universitätsklinikum Erlangen, Friedrich-Alexander-University Erlangen-Nürnberg, 91054 Erlangen, Germany. [9]Biotechnology Center, Center for Molecular and Cellular Bioengineering, Technische Universität Dresden, 01307 Dresden, Germany. [10]Mass Spectrometry Core Facility, Max Planck Institute of Biochemistry, 82152 Martinsried, Germany. [11]Laboratory of Biological Chemistry, Faculty of Medicine, School of Health Sciences, University of Ioannina, 45110 Ioannina, Greece. [12]VIB-Neuroelectronics Research Flanders, 3001 Leuven, Belgium. [13]Department of Neuroscience and Leuven Brain Institute, KU Leuven, 3000 Leuven, Belgium. [14]These authors contributed equally: Vasiliki Tsata, Nora John. ✉e-mail: daniel.wehner@mpl.mpg.de

