## [Peer Review File · Nature Communications]

Small leucine-rich proteoglycans inhibit CNS regeneration by modifying the structural and mechanical properties of the lesion environmentREVIEWER COMMENTS

Reviewer #1 (Remarks to the Author):

In this manuscript, Kolb and colleagues show that small leucine-rich proteoglycans, so-called SLRPs are the causing factor for regeneration failure in the human spinal cord, whereas in zebrafish, their expression is low or absent, and therefore, the zebrafish can regenerate after injury. To a large extent, the authors use cutting-edge methods and approaches that are appropriate for the study.

While the experiments, data analyses, data presented, and images are well executed and presented, several significant issues hamper the excitement about the study. As I describe below, several interpretations, missing control experiments, experimental design problems, and inequitable comparisons need to be supported by the data, thus weakening the findings presented here.

Comments:

#1. While comparing species with and without regenerative capacity is essential, the comparison here must be more informative and demonstrate the importance of the differences. The authors compare human tissue from adult or adolescent humans with larvae zebrafish. This is problematic because the prenatal and neonatal mammalian nervous system has better regenerative ability than the limited one of the adults. Therefore, a fair comparison will be to validate the extracellular matrix depositions in early developmental stages in mammals, and for this possibility, the use of rodent species could demonstrate it better.

On the other hand, the authors could also use the adult zebrafish, where the regeneration process after the injury is slower lasting weeks. Therefore, if the hypothesis is correct, I could expect that SLRPs expression is higher in adult zebrafish than in larvae. Either way, these are essential experiments to validate the finding's importance and better support the overall claims.

#2. Another point for the unfair differential response reported here between zebrafish and humans is the crucial claimed link by the Authors that tissue mechanics in zebrafish (tissue stiffens) is related to the regenerative outcome. The authors have to consider the differential environmental pressures between the species. Zebrafish live in an aquatic environment the water pressure is far higher than the air. Therefore, we cannot exclude or preclude the possibility that tissue mechanics are related to these environmental burdens. In line with this, another point is that the aquatic environment is far more dangerous for the integrity of the nervous tissue as it causes osmotic shock to the neurons. Even if the buffered E3 medium is less hazardous than plain water still is not close to the extracellular solution (environment) of the zebrafish spinal cord. Therefore, as described here, the extracellular matrix (ECM) deposition could not be the mechanism that zebrafish use to counteract this stress. Accordingly, the human tissue obtained by biopsies or post-mortem subjects is likely never exposed to any harsh environment, including the open air. So, the cellular mechanisms employed are expected to be different.

#3. The authors use the genetic induction of SLRPs in zebrafish to validate their importance in delaying regeneration by studying swimming performance and axons passing from the bridge formation (Figure 5). This is a critical study yet a bit problematic as it is presented. The axonal

growth is a consequence of the bridge establishment, where the glia is a fundamental player. As such, and according to the finding that SLRPs do not act on neurons (Figure 6), the authors did not ask the question of what the contribution of glia is in this process.

#4. While the study focuses on the central nervous system and spinal cord (Figure 1 B) the authors demonstrate in panel 1E, the expression of genes is the whole multi-tissue injured area. Therefore, it is still being determined if the genes are expressed in the skin, muscles, and bones close to or within the spinal cord. Higher magnification images with fluorescent in situ with a labeled spinal cord will be more informative. A similar comment applies to Figure 2C, and to several supplementary figures (Fig S4,5,6). As presented here, how important the genes are or the responses for the spinal cord versus the other tissues must be clarified. In line with this comment, it is somewhat confusing to validate the perivascular cells (Fig S4), neutrophils (Fig. S5), apoptotic cells, and fibroblasts (Fig. S6) in the whole body and not specifically in the spinal cord. Are there any differences in apoptotic cells, vascularization, and inflammation response within the spinal cord?

Minor:

1. A question emerges regarding Fig. 5B, regarding where the neuron somata in the *elav3:GFP* animal that the authors used.
2. Fig. 7 while the authors used the CP-OCT and MB to validate the tissue, it is rather difficult for the readers to follow. A better indication of where is the spinal cord and/or post-hoc comparison of the CP-OCT with a classical microscope imaging is necessary.

Reviewer #2 (Remarks to the Author):

In this manuscript, Kold et al found that SLRPs accumulation prevented axon regeneration in spinal cord injury rat model and patients, however lacking of SLRPs in larva zebrafish spinal cord injury could promote axon regeneration. This is a very interesting manuscript describing the specie difference of axon regeneration after spinal cord injury between zebrafish and mammals. By using MS based quantitative proteomics, in situ mRNA validation as well as a few functional experiment tests, the authors identified that SLRPs Lum and Prelp accumulating in the human spinal cord injury sites, not in larva zebrafish, inhibit axon regeneration by changing the mechanical properties of the lesion environments.

I have only a few concerns as follow:

1. The mechanisms of axon regeneration in developing spinal cord injury and adult spinal cord injury were proven differently at a few aspects. In this study, authors compared the larva zebrafish spinal cord injury with adult rats and human spinal cord injury to conclude that the SLRPs accumulation preventing axon regeneration in spinal cord injury patients and rat, not in zebrafish. I understand it is very convincing for the data of larva zebrafish spinal cord injury. But this still leaves somehow a question mark here, unless the authors show additional similar results in adult zebrafish spinal cord injury, since it might be a specific event for the adult animals or human.
2. The SLRPs have been well detected and two of them were well identified to exert the inhibitory effects on axon regeneration. I wonder where do these SLRPs come from? Are *pdgfrb*⁺ myoseptal and perivascular cells the only source cells to produce SLRPs? The authors should at least perform

experiments to show these cell really can produce SLRPs. Or at least to deeply discuss this if any other published paper already showed this.

The data in Fig 5A stating SLRPs induction in fibroblasts in not convincing. The authors need to show better co-localization of the cells and the SLRPs, since this part is very important.

Minor points:

1. The font on most of the figures are too small to read. I suggest to enlarge them.
2. In Fig. 3, the images showed the distribution of SLRPs in the human spinal cord injury slides seems region-specific (for example. anti-LUM or PRELP seems stain only a little proportion of the images). I wonder if the SLRPs are only expressed at the injury sites. If possible, please indicates the non-injured tissue and the injury sites?
3. In the discussion, the authors mentioned the SLRPs might also influence the neural growth factors. I would suggest to compare the data on neuronal growth factor protein or genes at rat or zebrafish spinal cord injury and put these data in a supplementary figure.

Reviewer #3 (Remarks to the Author):

The manuscript by Kolb et al provides evidence for an enrichment of Small Leucine-Rich Proteoglycans (SLRPs) in the inhibitory scar that exists in mammals (in this study rats and humans), that may underlie their inability to regenerate. This is indeed a new and important finding because these were previously unknown ECM factors that are expressed upon injury in the context of human CNS.

However, to say that “This identifies SLRPs as previously unknown inhibitory ECM factors in the human CNS that impair axon regeneration by altering tissue mechanics and structure.” is an overstating because this is solely based on indirect evidence. All the functional experiments were done in the zebrafish in the context of SLRPs overexpression that show a negative impact on axon regeneration and locomotor recovery and no complementary loss-of-function experiments were done in a mammalian model to show any kind of improvement when these SLRPs were reduced at the lesion core.

In a comprehensive proteomic analysis between zebrafish and rats, that was cross validated with RNA in situ hybridizations and qRT-PCR for the zebrafish and immunostainings for brain and spinal cord human samples, the authors were able to demonstrate that the SLRPs Chondroadherin, Fibromodulin, Lumican, and Prolargin are enriched in rats and humans, but not zebrafish, CNS lesions. This is indeed a new and important finding because these were previously unknown ECM factors in the human CNS that respond to injury. And additionally, makes us think the other way around which is that the regeneration capacity of the zebrafish can be attributed to the lack of axon growth-limiting components rather than the presence of species-specific growth-promoting factors.

The authors went on to produce zebrafish transgenic lines that would allow them to overexpress the selected SLRPs under the control of pdgfrb-positive fibroblast that were also shown to be the

relevant cells that produce these ECM factors upon injury. With this transgenic line the authors were able to induce the expression SLRPs considerably at the lesion site (evaluated by the increased levels of mCherry in Fig.5A) and further show the reduction of the axonal bridge in these animals that was then translated into a smaller swim distance (Fig.5B). The effects are somehow mild and very variable, so I was wondering what would happen if more than one SLRP factors was overexpressed in a combinatorial manner?

To investigate if the action of SLRPs on axon regrowth was a direct effect on neurons the authors use the Xla.Tubb promotor to overexpress SLRPs on neuronal populations and found that the effect on axonal bridge thickness was negligible (Fig. 6B), however I find puzzling the fact that at the lesion site there is no mCherry signal, which would indicate that the overexpression was not successful (Fig. 6A).

Even more puzzling are the results on Fig. 6C, were the authors show that culturing DRGs in a substrate coated with a combination of the SLRPs (Chondroadherin+Fibromodulin+Lumican+Prolargin) had no effect on neurite outgrowth. If, as the authors purpose, these SLRPs are modulating the structural and mechanical properties of the lesion environment (Fig. 7), I would expect to see a strong effect on neurite growth.

Reviewer #4 (Remarks to the Author):

In this paper, the authors use proteomics to study spinal cord regeneration in zebrafish. Comparing this data to mammalian data, they identify extracellular matrix proteins which are up-regulated after spinal cord injury in mammals, including rat and humans, but not in zebrafish. The authors show that overexpression of these proteins in zebrafish impairs spinal cord regeneration, and changes the mechanical properties of the injury.

The paper has a very good quality. The experiments are in general performed very carefully, and well presented. Their results are potentially relevant for human health, thus highlighting the potential impact of the publication. However, there are some points that should be addressed by the authors.

1. Figure S1C and S1D. Points are distributed in a surprisingly clustered way. The authors should revise the analysis to find if there is something that can be improved. One possibility that may explain this clustering is the following. Log2FC cannot be calculated if the expression in one of the samples is zero, so a small value, for example, 0.001 counts, is usually added to all the samples. When this value is too low, this kind of “weird” clusters may appear. I suggest that the authors revise the analysis, and add a higher value if it makes sense, which may reduce these artifacts. In any case, I would not expect that the main conclusions change after these small changes.

2. In section “SLRPs are abundant in human CNS lesions”, the authors do not analyze anymore the Dcn and Ogn proteins. The authors should justify why they decided to focus on the other four proteins, and not to continue the Dcn and Ogn analysis.

3. I can find some problems in the interpretation of the section “SLRPs do not directly act on neurons

to inhibit neurite extension". In the first part (Fig 6A and 6B), the authors show that expressing SLRPs in neurons has no effect on axons. This experiment does not rule out a direct action of SLRPs in neurons, it could just be that neurons are not able to secrete SLRPs, and SLRPs may only have an effect if they are secreted, so this experiment is not very conclusive.

I do not think that the second part of the section (Fig. 6C) corroborates the same findings. It is just a different experiment, which is more conclusive, showing that extracellular SLRPs do not affect axon growth. However, the interpretation of this experiment also generates some doubts. I would expect that adding these proteins would affect the mechanical properties of the plate, and according to the main claim of the paper, changes in these mechanical properties should affect axon regeneration. How do the authors explain the absence of effect?

The authors should explain better the implications of each of the experiments, and interpret them more carefully following the indications above.

4. After a first reading, it is difficult to understand why the authors do the MTZ experiment (S6A), as it is not explained in the main text. After reading the S6A figure legend, it becomes clear that it is just a positive control. The authors could make easier the understanding of this experiment by including a sentence in the main text.

Also, this TUNEL experiment is not really connected with the title and conclusions of its section: "SLRPs do not alter the fibroblast response". TUNEL signal is not detected in fibroblasts, but it is also not detected in any other tissue.

5. In S7D, initially, it is also difficult to understand why the authors choose 5 different genes for each experiment, instead of analyzing the 25 genes in the 4 different experiments. It would be helpful to add one sentence in the main text explaining this. Also, I count 20 genes, not 25.

6. Brillouin microscopy experiment are correctly performed. However, not all the required information is displayed. In Fig 7C, the authors should show a representative image of each of the conditions that they analyze, as they do in Fig 7B. The same applies to Fig 8. Also, the overlay of brightfield, Brillouin frequency shift and fluorescence in 7C is almost impossible to see. The authors should show this information in separated channels.

7. In Fig. 7C and Fig S8, the authors should avoid the use of the "jet" colorscale, and use instead viridis, inferno, magma or plasma, which are perceptually uniform and robust to colorblindness.

8. Although the authors provide information about how the plasmids were generated, it would not be possible to exactly reproduce these experiments unless the authors provide the full sequence of the plasmids they used. The authors should attach the plasmids maps as supplementary data.

REVIEWER COMMENTS

Reviewer #1 (Remarks to the Author):

In this manuscript, Kolb and colleagues show that small leucine-rich proteoglycans, so-called SLRPs are the causing factor for regeneration failure in the human spinal cord, whereas in zebrafish, their expression is low or absent, and therefore, the zebrafish can regenerate after injury. To a large extent, the authors use cutting-edge methods and approaches that are appropriate for the study.

While the experiments, data analyses, data presented, and images are well executed and presented, several significant issues hamper the excitement about the study. As I describe below, several interpretations, missing control experiments, experimental design problems, and inequitable comparisons need to be supported by the data, thus weakening the findings presented here.

RESPONSE: We thank the reviewer for the positive evaluation of our experiments, data analyses, data presentation, and methodology.

Comments:

#1. While comparing species with and without regenerative capacity is essential, the comparison here must be more informative and demonstrate the importance of the differences. The authors compare human tissue from adult or adolescent humans with larvae zebrafish. This is problematic because the prenatal and neonatal mammalian nervous system has better regenerative ability than the limited one of the adults. Therefore, a fair comparison will be to validate the extracellular matrix depositions in early developmental stages in mammals, and for this possibility, the use of rodent species could demonstrate it better. On the other hand, the authors could also use the adult zebrafish, where the regeneration process after the injury is slower lasting weeks. Therefore, if the hypothesis is correct, I could expect that SLRPs expression is higher in adult zebrafish than in larvae. Either way, these are essential experiments to validate the finding's importance and better support the overall claims.

RESPONSE: We appreciate the reviewer's comment and agree that it would be of interest to determine whether CNS lesions in neonatal mice lack SLRP proteins. Unfortunately, establishing a neonatal mouse SCI model is beyond the scope of this study. However, as per the reviewer's suggestion, we have examined the expression of SLRPs in the spinal lesion site of adult zebrafish, whose regenerative capacity is comparable to that of larval zebrafish ¹. We performed sensitive HCR fluorescence RNA *in situ* hybridization on tissue sections of the lesion core and rostral control tissue of adult zebrafish at 7 dpl, which corresponds to the early phase of axonal regrowth ². We found that expression of *aspn*, but not *chad*, *fmoda*, *lum*, and *prelp* is increased in the spinal lesion core as compared to baseline levels in control spinal cord tissue of the same animal (see new Fig. 2d). This further supports that mechanisms of axon regeneration are conserved between larval and adult stages of zebrafish ^{3,4}.

To obtain additional evidence that the capacity for CNS axon regeneration scales with SLRP protein abundance in the injury ECM, we also assessed immunofluorescence of anti-Chad, anti-Fmod, anti-Lum, and anti-Prelp on tissue sections of adult mice after complete spinal cord transection. Consistent with the findings in human specimens, we observed prominent immunoreactivity for all four proteins in both the spinal cord stumps and the lesion core (see new Fig. 2e).

Collectively, the data obtained from adult rats ⁵, adult mice (this study), adult humans (this study), larval zebrafish (this study), and adult zebrafish (this study) strongly support a reciprocal correlation between the abundance of SLRPs in the injury ECM and the regenerative capacity of the CNS. We have added the

new data as Fig. 2d and Fig. 2e, and describe it (page 8, lines 121-133, changes marked in yellow) as follows:

[...] To obtain further evidence that the capacity for axon regeneration scales with SLRP protein abundance, we analyzed their expression after SCI in adult zebrafish and adult mice, which possess high and low regenerative capacity for the CNS, respectively. We decided to concentrate on Fmod, Lum, and Prelp due to their structural similarities as class II SLRPs. Additionally, we focused on Chad which showed the highest abundance among SLRPs in the rat spinal lesion site (Fig. 2a). Consistent with our findings in larval animals, fluorescence ISH on tissue sections of adult zebrafish revealed upregulation of *aspn* expression in the highly disorganized lesion core at 7 dpl, whereas expression of *chad*, *fmoda*, *lum*, and *prelp* was not increased over baseline levels (Fig. 2d). In contrast, we found prominent anti-Chad, anti-Fmod, anti-Lum, and anti-Prelp immunoreactivity in spinal cord stumps and the lesion core of adult mice at one month after spinal cord transection (Fig. 2e). The abundance of Chad, Fmod, Lum, and Prelp proteins in the injury ECM therefore reciprocally correlates with the CNS regenerative capacity. [...]

#2. Another point for the unfair differential response reported here between zebrafish and humans is the crucial claimed link by the Authors that tissue mechanics in zebrafish (tissue stiffens) is related to the regenerative outcome. The authors have to consider the differential environmental pressures between the species. Zebrafish live in an aquatic environment the water pressure is far higher than the air. Therefore, we cannot exclude or preclude the possibility that tissue mechanics are related to these environmental burdens.

RESPONSE: This is an interesting and thought-provoking hypothesis. We would like to clarify that our manuscript does not claim differential mechanical properties (e.g., stiffness) of injured CNS tissue in regeneration-competent zebrafish and non-regenerating mammals. This has already been established by others⁶⁻⁸. Here, we are advancing insights from this prior work. By leveraging the zebrafish as an attractive model system, we were able to identify which specific ECM proteins may contribute to changes in mechanical (scar) tissue properties, and thus impair axon regeneration in the mammalian CNS.

Our study provides an inroad into dissecting the notoriously complex fibrous scar tissue, a major factor limiting CNS axon regeneration in humans and other mammals. This is very challenging in mammalian animal models due to the complexity of the injury ECM and the plethora of co-existing inhibitory neuron-intrinsic and -extrinsic mechanisms. In the zebrafish, we can dissect the growth-modulating properties of specific ECM components and interrogate the underlying mechanisms *in vivo*, due to its intrinsically high regenerative capacity and axon growth-conducive lesion environment. We demonstrate this with four SLRP family members, which were previously unknown as inhibitors of axon regeneration and to be abundant in human CNS lesions (this has been explicitly appreciated by reviewer #3). Furthermore, our data show that genetically increasing the abundance of a single ECM factor in the zebrafish injury ECM (Lum, Prelp) is sufficient to impair axon regeneration and concomitantly alter both the structural and mechanical (longitudinal modulus) properties of the lesion environment. This suggests that SLRPs confer mechano-structural properties to the injury ECM that are adverse to axon growth; a previously undocumented mechanism limiting CNS regeneration.

Although we did not find evidence that SLRPs inhibit regeneration through direct (biochemical) interaction with the axonal growth cone (via ligand-receptor interaction), by preventing the resolution of inflammation, or by altering the composition of the fibroblast-derived injury ECM, we cannot fully exclude these mechanisms. We have explicitly stated this in the discussion of our manuscript. However, to avoid potential confusion, we have toned down this statement in the revised manuscript as follows (page 18, lines 378-380, changes marked in yellow):

[...] Here, we propose a new class of inhibitors of CNS axon regeneration which may function in part through modifying the structural and mechanical properties of the lesion environment. [...]

While our data does not exclude or preclude the possibility that tissue mechanics of regenerating CNS tissue is related to the environmental burden, this hypothesis is not supported by the literature. To our knowledge, no correlation between the CNS regenerative capacity and the species' habitat

(aquatic/terrestrial) exists. There are several known aquatic species with high (zebrafish⁹, goldfish¹⁰, lamprey¹¹, eel¹², carp¹³) and low (marine mammals, killifish¹⁴, medaka^{15,16}, frogs¹⁷⁻¹⁹) regenerative capacity. The same holds true for terrestrial species, among which some exhibit a high regenerative capacity for the CNS, including lizards/geckos (*Anolis carolinensi*²⁰, *Lepidodactylus lugubri*²¹, *Eublepharis macularius*²², *Gekko japonicus*²³) and, most notably, the African spiny mouse (*Acomys cahirinus*^{24,25}). Finally, urodele amphibians which are both land and water-based animals, are well-established to possess a high regenerative capacity for the CNS (*Notophthalmus viridescens*²⁶, *Pleurodeles waltlii*²⁷, *Ambystoma mexicanum*²⁸⁻³¹). Furthermore, the pressure (p) difference between a terrestrial habitat at sea level (p = 1 atm = 1.01325 bar) and that of larval zebrafish at the time of surgery and thereafter (aqueous environment at a maximum depth of 15 mm, p = 1.01475 bar) is only 0,0015038 bar or 0.15% at most. The individual species listed above experience a much greater range of environmental pressure regardless of their regenerative abilities. But even in cases where they occupy habitats that are comparable with respect to environmental pressure, they still display greatly varying regenerative capacity.

Crucially, we would like to emphasize that all our Brillouin measurements were done in the same species (zebrafish) and that the conditions were the same with the exception of a single ECM factor whose abundance we altered in the spinal injury ECM via genetic tools. Hence, we can exclude that the observed differences in mechanical tissue properties (longitudinal modulus) and regeneration outcome were a consequence of differential environmental burden.

Nevertheless, to obtain additional supportive evidence for a role of SLRP proteins in modulating the mechanical properties of the injury ECM in CNS lesions, we performed atomic force microscopy (AFM)-based nanoindentation measurements on acute, living tissue preparations to quantify the viscoelastic properties of the lesion core in the presence and absence of SLRPs. We found that targeting Fmoda, Lum, or Prelp to the injury ECM reduced the apparent Young's modulus by 44-66%. The apparent viscosity was reduced by 24-25% in *pdgfrb:fmoda* or *pdgfrb:prelp* transgenic fish as compared to controls. We have added these data to Fig. 7d and Supplementary Fig. 10, and describe it (page 16-17, lines 343-354, changes marked in yellow) as follows:

[...] We next performed atomic force microscopy (AFM)-based nanoindentation measurements on acute, living tissue preparations to quantify the viscoelastic properties (Young's modulus E , viscosity η) of the lesion core in the presence and absence of SLRP proteins (analyzed in *pdgfrb:SLRPs* transgenics; Fig. 7d). Applying two independent fitting models to the recorded force-indentation curves (Kelvin-Voigt-Maxwell, Hertz), we found that targeting Fmoda, Lum, or Prelp to the injury ECM reduced the apparent Young's modulus by 44-66% at 2 dpl (Fig. 7d'; Supplementary Fig. 10). Furthermore, the apparent viscosity was reduced by 24-25% in *pdgfrb:fmoda* and *pdgfrb:prelp* transgenic animals as compared to controls (Fig. 7d''). Thus, targeting Fmoda, Lum, or Prelp to injury ECM leads to changes in the viscoelastic response of the lesion environment. Altogether, these data show that SLRPs alter the mechanical properties of the injury ECM and inhibit axon regeneration after SCI. [...]

In line with this, another point is that the aquatic environment is far more dangerous for the integrity of the nervous tissue as it causes osmotic shock to the neurons. Even if the buffered E3 medium is less hazardous than plain water still is not close to the extracellular solution (environment) of the zebrafish spinal cord. Therefore, as described here, the extracellular matrix (ECM) deposition could not be the mechanism that zebrafish use to counteract this stress. Accordingly, the human tissue obtained by biopsies or post-mortem subjects is likely never exposed to any harsh environment, including the open air. So, the cellular mechanisms employed are expected to be different.

RESPONSE: We appreciate the reviewer's comment. Although differences clearly exist between a CNS injury in humans and the zebrafish SCI model used in our study, neurons will be exposed to a harsh environment in both systems. In humans, tissue swelling due to edema (intracellular and interstitial increase in CNS tissue water), oxidative stress and hyperosmotic shock due to hemorrhage, and sustained inflammation causes additional (secondary) injury often in excess to the primary trauma³²⁻³⁴. Of note,

some of the human brain samples used in this study were derived from patients that underwent repeated surgery. Thus, the CNS tissue has even been exposed to the “*open air*”. Consistent with this, we observed the accumulation of SLRPs mainly in areas of scarring that were caused by contusion, local hemorrhage, or previous surgery, supporting a causal connection.

Furthermore, we respectfully disagree with the reviewer that ECM deposition cannot be a factor contributing to the differential regenerative capacity among species. A large body of literature exists on the regeneration-inhibiting function of injury-induced ECM deposits (fibrous CNS scars) in mammals e.g., 35-39. Importantly, there is accumulating evidence indicating that the composition of the injury ECM differs between regeneration-competent and non-regenerating species (e.g., zebrafish vs. rodents/humans ^{this study, 3}, *Axolotl* vs. rat ⁴⁰, African spiny mouse vs. *mus musculus* ^{24,41}), and that quantitatively altering the ECM can, depending on the context, impair or promote axon regeneration ^{3,4,42-49, this study}. Clearly, there must be differences in the cellular mechanism that will lead to the observed differences in ECM composition. In support of this, we have recently shown that zebrafish fibroblasts selectively suppress the expression of inhibitory matrix molecules in order to establish an axon growth-permissive ECM after SCI ³. Taken together, our findings are both novel and consistent with existing literature.

#3. The authors use the genetic induction of SLRPs in zebrafish to validate their importance in delaying regeneration by studying swimming performance and axons passing from the bridge formation (Figure 5). This is a critical study yet a bit problematic as it is presented. The axonal growth is a consequence of the bridge establishment, where the glia is a fundamental player. As such, and according to the finding that SLRPs do not act on neurons (Figure 6), the authors did not ask the question of what the contribution of glia is in this process.

RESPONSE: We appreciate the reviewer’s comment. However, we respectfully disagree with the notion that axonal (re-)growth is a consequence of glial bridge establishment across the spinal lesion site in zebrafish. While bridging of glial cells (also called astrocyte-like glia or ependymo-radial glia) has been proposed to provide a substrate/scaffold for regenerating axons to cross the non-neural lesion site ⁵⁰, this view is highly controversial. To date, no study has provided unambiguous evidence (through glial cell-specific manipulation) that establishes astrocyte-like glia as a fundamental player in long-distance axonal regeneration. Instead, histological analysis in the eel ⁵¹, ultrastructural analysis of the adult goldfish ⁵², and live observations of glial and neuronal processes in larval zebrafish ⁴ provided evidence that axonal fascicles precede glial cells when navigating the spinal lesion site. Even in the elegant publication by Goldsmith et al. it was reported, that in adult zebrafish a substantial proportion of regenerating axons was found to not grow alongside GFAP⁺ glial processes (25% at 10 dpl; see their Fig. 2e) ⁵⁰. Crucially, axonal bridging was also established in larval zebrafish when GFAP⁺ glial cells were selectively ablated using the Nitroreductase/Metronidazole system, which we have reported in 2017 in *Nat. Commun.* 8:126 ⁴. This indicates that glial bridging is dispensable for axonal regrowth across the spinal lesion site, at least in larval zebrafish. However, we agree with the reviewer that it would be interesting to know if glial bridging is affected when SLRPs are abundant in the injury ECM. We have investigated this and found that targeting *Chad*, *Fmoda*, *Lum*, or *Prelp* to the injury ECM (in *pdgfrb*:SLRP transgenics) also impairs glial bridging. These data further strengthen our working model that the injury ECM is less permissive to neural regeneration in the presence of SLRPs. We have added these data to Supplementary Fig. 4h and describe it (page 11, lines 206-210, changes marked in yellow) as follows:

[...] Our data therefore identify *Chad*, *Fmod*, *Lum*, and *Prelp* as ECM factors that inhibit CNS axon regeneration *in vivo*. Interestingly, bridging of astroglia-like processes, which cross the non-neural lesion site independently of regenerating axons, was also reduced by 44-54% when *Chad*, *Fmoda*, *Lum*, or *Prelp* was targeted to injury ECM (analyzed in *her4.3*:GFP-F transgenic animals; Supplementary Fig. 4h) ^{4,51,52}. This suggests that the injury ECM may be less permissive to neural regeneration in the presence of SLRPs.[...]

#4. While the study focuses on the central nervous system and spinal cord (Figure 1 B) the authors demonstrate in panel 1E, the expression of genes is the whole multi-tissue injured area. Therefore, it is still being determined if the genes are expressed in the skin, muscles, and bones close to or within the spinal cord. Higher magnification images with fluorescent in situ with a labeled spinal cord will be more informative. A similar comment applies to Figure 2C, and to several supplementary figures (Fig S4,5,6). As presented here, how important the genes are or the responses for the spinal cord versus the other tissues must be clarified. In line with this comment, it is somewhat confusing to validate the perivascular cells (Fig S4), neutrophils (Fig. S5), apoptotic cells, and fibroblasts (Fig. S6) in the whole body and not specifically in the spinal cord. Are there any differences in apoptotic cells, vascularization, and inflammation response within the spinal cord?

RESPONSE: We apologize for any lack of clarity that may have caused confusion. The experimental design was based on the anatomy and the previously well-characterized cellular responses to SCI in the larval zebrafish model^{3,4,53}. At the analyzed larval stage (3-5 days post-fertilization), the trunk is boneless and the spinal cord is avascular (e.g., see⁵⁴). This precludes analyzing re-vascularization (and associated cells) of the injured spinal cord following the manipulation of SLRPs. Furthermore, re-vascularization of the injured trunk tissue occurs after axon regeneration is completed⁵³, indicating a negligible role in axon regeneration in the larval SCI model.

The injury paradigm used in our study involves the complete transection of the spinal cord, which is situated immediately dorsal to a distinct and easily identifiable structure known as notochord. Spinal cord transection is achieved through either perforation or incision, which also damages surrounding trunk tissue, including muscle, vasculature, and skin^{3,4,53,55}. While we agree with the reviewer that the observed changes in protein abundance/gene expression after injury may not be exclusive to spinal cord tissue or the lesion core, this was not a claim made in our study. In fact, we anticipate that injury-induced expression of certain matrix genes is common to several tissues since fibroblasts reacting to SCI are associated with the dermis, vasculature, the notochord, and myotendinous junctions^{3,4,56}. The purpose of the experiment presented in Fig. 1e was to validate our mass spectrometry (MS) analysis, which was performed on whole-trunk tissue containing the spinal lesion site. Using whole-trunk tissue was necessary since it is technically extremely challenging to obtain sufficient material for MS from laser-dissected spinal lesion sites of larval zebrafish. Nevertheless, our MS analysis identified several ECM proteins that have previously been implicated in SCI and repair in zebrafish and other species (e.g., see page 5/6, lines 54-55 and lines 59-63), which supports the utility of our approach. However, we acknowledge the reviewer's suggestion of providing more spatial information regarding the injury-induced gene expression, which would be of interest to the scientific community. Therefore, in our revised manuscript we employed fluorescence *in situ* hybridization (FISH) on transversal tissue sections to assess the expression pattern of 10 genes shown in Fig. 1e. This revealed expression of all examined genes in the spinal cord stump, its immediate vicinity, or the highly disorganized lesion core. This further supports the utility of our model system in identifying genes associated with SCI and repair. We have added the data to Fig. 1f and describe it (page 6, lines 73-77, changes marked in yellow) as follows:

[...] Furthermore, fluorescence ISH on tissue sections for 10 selected genes confirmed transcript induction in the spinal cord stump, its immediate vicinity, or the highly disorganized lesion core (Fig. 1f). This reinforces the findings of our proteomics profiling. Collectively, these data identify the dynamics of the matrix landscape after SCI in a vertebrate species which exhibits a high regenerative capacity for the CNS. [...]

The purpose of the experiment shown in Fig. 2c was to demonstrate that expression of the SLRPs *chad*, *dcn*, *fmoda/b*, *ogn*, *lum*, and *prelp* is not upregulated in the lesion core at the time point of active axonal regrowth in the larval zebrafish SCI model (1 dpl). Therefore, we fail to see how FISH, whether performed on tissue sections or whole-mounts, would be more informative, considering that we did not detect transcripts in the lesion site at the multi-tissue level. However, we have performed FISH on tissue sections to detect transcripts of *aspn*, the only SLRP family member whose expression was found to be upregulated in the lesion site (in whole-mount preparations). This showed expression predominantly in the lesioned

spinal cord (see above and Fig. 1f). Furthermore, we have analyzed expression of the relevant SLRP family members *aspn*, *chad*, *fmoda*, *lum*, and *prelp* by FISH on tissue sections of adult zebrafish (also see our response to your comment #1). This confirmed upregulation of *aspn* expression in the highly disorganized lesion core whereas expression of *chad*, *fmoda*, *lum*, and *prelp* was not increased over baseline levels. We have added the data to Fig. 2d and describe it (page 8, lines 121-133, changes marked in yellow) as follows:

[...] To obtain further evidence that the capacity for axon regeneration scales with SLRP protein abundance, we analyzed their expression after SCI in adult zebrafish and adult mice, which possess high and low regenerative capacity for the CNS, respectively. We decided to concentrate on Fmod, Lum, and Prelp due to their structural similarities as class II SLRPs. Additionally, we focused on Chad which showed the highest abundance among SLRPs in the rat spinal lesion site (Fig. 2a). Consistent with our findings in larval animals, fluorescence ISH on tissue sections of adult zebrafish revealed upregulation of *aspn* expression in the highly disorganized lesion core at 7 dpl, whereas expression of *chad*, *fmoda*, *lum*, and *prelp* was not increased over baseline levels (Fig. 2d). In contrast, we found prominent anti-Chad, anti-Fmod, anti-Lum, and anti-Prelp immunoreactivity in spinal cord stumps and the lesion core of adult mice at one month after spinal cord transection (Fig. 2e). The abundance of Chad, Fmod, Lum, and Prelp proteins in the injury ECM therefore reciprocally correlates with the CNS regenerative capacity. [...]

The purpose of the experiment shown in Fig. S4a (now Supplementary Fig. 4a) was to verify that *pdgfrb*⁺ fibroblasts express endogenous *aspn*, *chad*, *fmoda*, *lum*, and *prelp*, and thus, to obtain evidence for their ability to produce the SLRP-mCherry fusion proteins. *pdgfrb*⁺ fibroblasts were isolated using FACS from trunk tissue of uninjured animals for two reasons: i) uninjured tissue was used since reactive *pdgfrb*⁺ fibroblasts downregulate expression of SLRPs after SCI³. ii) *pdgfrb*⁺ fibroblasts are exclusively located outside the spinal cord in larval zebrafish, occupying niches including the vasculature and myotendinous junctions³. This precludes isolating *pdgfrb*⁺ fibroblasts from spinal cord tissue.

The purpose of the experiment shown in Fig. S6b (now Supplementary Fig. 4e) was to ascertain that TetON system-mediated expression of *slrp-mCherry* fusions in *pdgfrb*⁺ fibroblasts does not induce apoptosis. This analysis cannot be limited to the spinal cord alone since *pdgfrb*⁺ fibroblasts are exclusively located outside the spinal cord, as explained above. A similar logic applies to the data presented in Fig. S6c (now Supplementary Fig. 7). Although reactive *pdgfrb*⁺ fibroblasts have been observed from distances up to 150 μm away from the spinal lesion center³, the region of quantification was limited to the spinal cord stumps, including the lesion site along the rostro-caudal and dorso-ventral axis. After the reviewer's comment, we indeed realized that we inadvertently missed to indicate the region of quantification in the submitted manuscript, which we have corrected in the revised version to better illustrate our analysis. Although we cannot assess recruitment of *pdgfrb*⁺ fibroblasts from within the spinal cord for the reasons discussed above, we have re-analyzed our data and limited the region of quantification in the confocal image to the level of the spinal cord including the transection site in all three dimensions. However, we found no major difference in the area coverage of *pdgfrb*⁺ fibroblast, demonstrating the robustness of our assay (see Fig. R1).

Fig. R1 | *pdgfrb*⁺ cell-specific induction of SLRPs does not prevent recruitment of *pdgfrb*⁺ fibroblast-like cells to the lesion site. Dashed lines indicate the location of the severed spinal cord. The rectangle indicates the region of quantification, which was limited to the level of the spinal cord, including the transection site, in all three dimensions. Image shown is a maximum

intensity projection, which is limited to the level of the spinal cord. Each data point represents one animal. a.u., arbitrary unit; n.s., not significant; sc, spinal cord; BF, brightfield.

The dynamics of immune cell recruitment in the larval zebrafish SCI model have been characterized in great detail⁵³. Spinal cord-resident microglia are negligible in promoting axon regeneration whereas peripheral neutrophils and macrophages are rapidly recruited after SCI. To exclude that we missed a potential effect in SLRP-expressing animals, we have re-analyzed our data and limited the region of quantification in the confocal images to the level of the spinal cord including the transection site in all three dimensions. We found no difference in the number of Mpx⁺ neutrophils, demonstrating the robustness of our assay (see Fig. R2).

Fig. R2 | Targeting SLRPs to the injury ECM does not prevent inflammation resolution. Dashed lines indicate the location of the severed spinal cord. The rectangle indicates the region of quantification, which was limited to the level of the spinal cord, including the transection site, in all three dimensions. Image shown is a maximum intensity projection, which is limited to the level of the spinal cord. Each data point represents one animal. n.s., not significant; sc, spinal cord; BF, brightfield.

Finally, we went through the manuscript and revised the pertinent parts to make it easier for the reader to follow.

Minor:

1. A question emerges regarding Fig. 5B, regarding where the neuron somata in the *elav3*:GFP animal that the authors used.

RESPONSE: We thank the reviewer for the question. As part of this study, we created the *elav3*:GFP-F transgenic zebrafish line to improve neurite labeling. The transgene consists of regulatory elements of the zebrafish *elva3* gene⁵⁷, which drive expression of a membrane-localized GFP (EGFP fused to farnesylation signal from c-HA-Ras) in all neurons of the brain and spinal cord (also see Fig. 1b of our manuscript). As can be appreciated in the optical section through the unlesioned spinal cord in the image presented below (see lateral view in Fig. R3a), the neuronal somata are well-labeled by the membrane-localized GFP. However, only few neuronal somata can be seen in Fig. 5b of our manuscript (now Fig. 5f), which shows the spinal cord at two days after complete transection (days-post lesion; dpl) due to the following reasons. First, due to i) the anatomy of the (larval zebrafish) spinal cord (white matter tracts are located in the outer region, cell somata are situated closer to the center) and ii) the reduction of fluorescence intensity signal in deeper tissue layers caused by an increase in light scattering, most of the neuronal somata cannot be seen in a maximum intensity projection of a confocal z-stack of whole mount preparations (lateral view), even in unlesioned animals. Second, the image shown in Fig. 5f of our manuscript has been cropped to focus on the center of the spinal lesion site. Our lesioning paradigm, which involves complete transection of the spinal cord, leads to immediate and massive neuronal loss in the lesion center due to the acute mechanical trauma (also see Fig. 1b of our manuscript), followed by secondary loss of neurons due to necrosis and/or apoptosis toward the edges of the lesion^{53,58}. Hence, after spinal cord transection no or only very few neuronal somata are present in the lesion center but towards the lesion edges at 2 dpl. This can be appreciated in the uncropped image of Fig. 5f, which is presented below (Fig. R3b; arrow).

Fig. R3 | Confocal image of the spinal cord of *e/av3:GFP-F* transgenic animals before (a) and at 2 days after (dpl) complete spinal cord transection (b). The arrow indicates neuronal somata located at the edge of the lesion site. Scale bars: 20 μm .

We have edited the manuscript (page 5, lines 37-41; marked in yellow) and the figure legend of Fig. 5f (page 60, changes marked in yellow) to provide more details:

• **Results:**

[..] To that aim, we applied label-free mass spectrometry (MS)-based quantitative proteomics to a larval zebrafish spinal cord injury (SCI) model. In this injury paradigm, the spinal cord is completely transected, which allows axon regeneration across an ECM-rich, non-neural lesion environment and recovery of swimming function within two days post-lesion (dpl; Fig. 1a,b)^{4,59}. [...]

• **Figure legend Fig. 5f:**

pdgfrb⁺ cell-specific induction of the SLRPs *chad*, *fmoda*, *lum*, and *prelp* but not *aspn* in *pdgfrb:SLRP* transgenic zebrafish reduces the thickness of the axonal bridge (f'; analyzed in *e/av3:GFP-F* transgenics) and impairs recovery of swimming distance (f'') at 2 dpl. Image of the axonal bridge shown in (f) is a maximum intensity projection of the center of the spinal lesion site in *e/av3:GFP-F* transgenics at 2 dpl (lateral view; rostral is left). Note that neuronal somata are largely absent from the lesion core due to the acute mechanical trauma of the spinal cord transection.

2. Fig. 7 while the authors used the CP-OCT and MB to validate the tissue, it is rather difficult for the readers to follow. A better indication of where is the spinal cord and/or post-hoc comparison of the CP-OCT with a classical microscope imaging is necessary.

RESPONSE: We thank the reviewer for the helpful suggestion. In larval zebrafish, the spinal cord is located immediately dorsal to the notochord, a prominent structure that can be easily recognized with all optical techniques used in this manuscript. In the revised Fig. 7b, we have indicated the location of the spinal cord stumps in each presented CP-OCT image with dashed lines (note that the dashed rectangle indicates the region of quantification). The dimension of the intact spinal cord along the dorsal-ventral axis (50 μm) was identified in optical sections of the corresponding cross-polarized image stack. Additionally, we have revised Fig. 7c to facilitate orientation in the image.

Reviewer #2 (Remarks to the Author):

In this manuscript, Kold et al found that SLRPs accumulation prevented axon regeneration in spinal cord injury rat model and patients, however lacking of SLRPs in larva zebrafish spinal cord injury could promote axon regeneration. This is a very interesting manuscript describing the specie difference of axon regeneration after spinal cord injury between zebrafish and mammals. By using MS based quantitative proteomics, in situ mRNA validation as well as a few functional experiment tests, the authors identified that SLRPs Lum and Prelp accumulating in the human spinal cord injury sites, not in larva zebrafish, inhibit axon regeneration by changing the mechanical properties of the lesion environments.

RESPONSE: We are glad that the reviewer appreciates our work as very interesting, and thank them for their specific suggestions, which allowed us to improve the manuscript.

I have only a few concerns as follow:

1. The mechanisms of axon regeneration in developing spinal cord injury and adult spinal cord injury were proven differently at a few aspects. In this study, authors compared the larva zebrafish spinal cord injury with adult rats and human spinal cord injury to conclude that the SLRPs accumulation preventing axon regeneration in spinal cord injury patients and rat, not in zebrafish. I understand it is very convincing for the data of larva zebrafish spinal cord injury. But this still leaves somehow a question mark here, unless the authors show additional similar results in adult zebrafish spinal cord injury, since it might be a specific event for the adult animals or human.

RESPONSE: We thank the reviewer for the comment. To address this, we have performed highly sensitive HCR fluorescence *in situ* hybridization on sections of the lesion core and rostral control tissue of adult zebrafish at 7 dpl, which corresponds to the early phase of axonal regrowth^{1,2}. We found that expression of *aspn* but not *chad*, *fmoda*, *lum*, and *prelp* is increased in the spinal lesion core of adult zebrafish at 7 dpl, as compared to baseline levels in rostral control segments of the same animal. This further supports that mechanisms of axon regeneration are conserved between larval and adult stages^{3,4}. We have added these data to Fig. 2d and describe it (page 8, lines 121-133, changes marked in yellow) as follows:

[...] To obtain further evidence that the capacity for axon regeneration scales with SLRP protein abundance, we analyzed their expression after SCI in adult zebrafish and adult mice, which possess high and low regenerative capacity for the CNS, respectively. We decided to concentrate on Fmod, Lum, and Prelp due to their structural similarities as class II SLRPs. Additionally, we focused on Chad which showed the highest abundance among SLRPs in the rat spinal lesion site (Fig. 2a). Consistent with our findings in larval animals, fluorescence ISH on tissue sections of adult zebrafish revealed upregulation of *aspn* expression in the highly disorganized lesion core at 7 dpl, whereas expression of *chad*, *fmoda*, *lum*, and *prelp* was not increased over baseline levels (Fig. 2d). In contrast, we found prominent anti-Chad, anti-Fmod, anti-Lum, and anti-Prelp immunoreactivity in spinal cord stumps and the lesion core of adult mice at one month after spinal cord transection (Fig. 2e). The abundance of Chad, Fmod, Lum, and Prelp proteins in the injury ECM therefore reciprocally correlates with the CNS regenerative capacity. [...]

2. The SLRPs have been well detected and two of them were well identified to exert the inhibitory effects on axon regeneration. I wonder where do these SLRPs come from? Are pdgfrb+ myoseptal and perivascular cells the only source cells to produce SLRPs? The authors should at least perform experiments to show these cell really can produce SLRPs. Or at least to deeply discuss this if any other published paper already showed this.

RESPONSE: We thank the reviewer for the question. Unfortunately, from the comment it was not entirely clear to us what the reviewer precisely referred to. From our interpretation, the reviewer asked if *pdgfrb*⁺ (myoseptal and perivascular) cells are a source or the only source of endogenous SLRP proteins in the zebrafish, and thus, can be considered competent to secrete the SLRP-mCherry fusion proteins. In Supplementary Fig. 4a, we have performed quantitative RT-PCR (qPCR) on *pdgfrb*⁺ and *pdgfrb*⁻ cells that were isolated from unlesioned *pdgfrb*:GFP transgenic animals by fluorescence-activated cell sorting (FACS). This revealed several-fold higher transcript levels of *aspn*, *chad*, *lum*, and *prelp* in *pdgfrb*⁺ cells as compared to *pdgfrb*⁻ cells. This indicates that *pdgfrb*⁺ cells are a main source of endogenous SLRP proteins in uninjured larval zebrafish, albeit not the only. Furthermore, this supports that *pdgfrb*⁺ cells are competent to produce/secrete SLRP proteins. Finally, using RNA-sequencing of FACS-isolated *pdgfrb*⁺ cells, we have previously shown that *pdgfrb*⁺ cells accumulating in the spinal lesion site downregulate the expression of *fmoda*, *lum*, and *prelp* but upregulate *aspn* in larval zebrafish³. We have edited the manuscript to better describe these data (page 10, lines 163-170, changes marked in yellow):

[...] *pdgfrb*⁺ cells are rapidly recruited in response to SCI to secrete an injury-specific ECM in the lesion site (Fig. 5b)³. Moreover, qRT-PCR on GFP⁺ and GFP⁻ cells isolated from uninjured *pdgfrb*:GFP transgenic animals by FACS revealed several-fold higher expression levels of endogenous *aspn* ($\overline{FC} = 22.3$), *chad* ($\overline{FC} = 3.8$), *fmoda* ($\overline{FC} = 4.8$), *lum* ($\overline{FC} = 6.6$), and *prelp* ($\overline{FC} = 10.5$) transcripts in *pdgfrb*⁺ cells as compared to *pdgfrb*⁻ cells (Supplementary Fig. 4a). This indicates that *pdgfrb*⁺ cells are a main source of SLRP proteins under physiological conditions. Following SCI, *fmoda*, *lum*, and *prelp* expression is downregulated whereas *aspn* is upregulated in *pdgfrb*⁺ cells in zebrafish, which we have reported previously³. Together, this makes *pdgfrb*⁺ fibroblast-like cells an ideal target for manipulating the ECM in the zebrafish spinal lesion site. [...]

Alternatively, the reviewer asked us to clarify whether *pdgfrb*⁺ cells are capable of secreting the SLRP-mCherry fusion proteins. To address this, we have added magnified views of the mCherry fluorescence in *pdgfrb*:SLRP-mCherry animals (Supplementary Fig. 4b,d), which show distinct mesh and fiber-like structures in the case of *Aspn*, *Fmoda*, and *Prelp* (compare new Supplementary Fig. 4c showing *pdgfrb* promoter-driven cytoplasmic GFP). This supports the binding of the SLRP-mCherry fusions to fibrillar ECM. Furthermore, we have added three-dimensional (3D) reconstructions of high-resolution images of *pdgfrb*:SLRP;*pdgfrb*:GFP transgenic animals, which show accumulation of mCherry signal in the pericellular environment of *pdgfrb*⁺ cells (Supplementary Fig. 4d). Together, these data support that *pdgfrb*⁺ cell-derived SLRP-mCherry fusions are secreted into the extracellular space. We describe these data (page 10, lines 172-181, changes marked in yellow) as follows:

[...] Induction of *aspn-mCherry*, *chad-mCherry*, *fmoda-mCherry*, *lum-mCherry*, or *prelp-mCherry* expression in unlesioned *pdgfrb*:SLRP transgenic animals resulted in fluorescent labeling of mainly the myosepta and vasculature, thus resembling the *pdgfrb*⁺ cell niche (Supplementary Fig. 4b,c)³. 3D reconstruction of high-resolution images of *pdgfrb*:SLRP;*pdgfrb*:GFP transgenics revealed accumulation of mCherry signal in the pericellular environment of *pdgfrb*⁺ cells (Supplementary Fig. 4d). Furthermore, we observed variations in the mCherry fluorescence pattern among different SLRP proteins, including distinct mesh and fiber-like structures in the case of *Aspn*, *Fmoda*, and *Prelp* (Supplementary Fig. 4b,d). Together, this supports the secretion and binding of the SLRP-mCherry fusion proteins to fibrillar ECM⁶⁰. [...]

Another possibility may be that the reviewer asked to discuss current knowledge on the cellular origin of SLRPs in the mammalian CNS. In the revised manuscript, we cite the pertinent literature in the discussion section as follows (page 17, lines 370-373, changes marked in yellow):

[...] Current evidence suggests a context-dependent cellular origin of SLRPs in the mammalian CNS, including neurons⁶¹, this study), astrocytes (this study), and fibroblast lineage cells^{42,62,63}. Hence, future studies will need to precisely map the cellular source of SLRPs in CNS lesions. [...]

The data in Fig 5A stating SLRPs induction in fibroblasts is not convincing. The authors need to show better co-localization of the cells and the SLRPs, since this part is very important.

RESPONSE: We thank the reviewer for the comment and apologize for any confusion that may have been caused by the presentation of our data. The purpose of Fig. 5a (now Fig. 5e) was to show that *pdgfrb*⁺ fibroblast-derived SLRP proteins accumulate in the injury ECM of the highly disorganized lesion core. In the revised manuscript, we have edited the labels of the figure to better indicate that the fluorescence signal corresponds to the SLRP-mCherry fusion proteins but not the transcripts. Additionally, we have added more detail to Fig. 5 (see Fig. 5a,b) to better guide the reader of the manuscript.

While the *pdgfrb*⁺ cell type-specific Tet-activator zebrafish line (*pdgfrb*:TetA) has been previously characterized by our group ³, we agree that it would be helpful to provide additional data showing the specific induction of SLRPs in *pdgfrb*⁺ fibroblasts accumulating in the lesion core. Hence, we have co-stained GFP protein (anti-GFP) and *mCherry* mRNA on sections of the lesion site of *pdgfrb*:SLRP;*pdgfrb*:GFP transgenic animals. This showed co-labeling of mCherry transcripts and GFP protein. We have added these data to Fig. 5d and describe it (pages 10-11, lines 186-190, changes marked in yellow) as follows:

[...]). At 1 dpl, in the highly disorganized lesion core of *pdgfrb*:SLRP;*pdgfrb*:GFP transgenic animals, *pdgfrb*⁺ fibroblasts selectively expressed the *slrp-mCherry* fusions (Fig. 5c,d). Furthermore, mCherry fluorescence signal accumulated in the lesion core, indicating enrichment of the secreted proteins (Fig. 5e). [...]

Minor points:

1. The font on most of the figures are too small to read. I suggest to enlarge them.

RESPONSE: We thank the reviewer for pointing this out. We have increased the font size in all main and supplementary figure panels, and hope that this has improved the readability.

2. In Fig. 3, the images showed the distribution of SLRPs in the human spinal cord injury slides seems region-specific (for example. anti-LUM or PRELP seems stain only a little proportion of the images). I wonder if the SLRPs are only expressed at the injury sites. If possible, please indicates the non-injured tissue and the injury sites?

RESPONSE: We thank the reviewer for the question. Fig. 3 of our manuscript shows human brain tissue from patients with traumatic brain injury (TBI) or previous surgery (re-OP). Consistent with the reviewer's notion, immunoreactivity of anti-CHAD, anti-FMOD, anti-LUM, and anti-PRELP was observed to be mainly increased in areas of fibrotic scarring, marking the lesion core of, e.g., contusions, local hemorrhage, or previous surgery. In the revised manuscript, we have edited Fig. 3 to indicate the lesion core with an asterisk. Furthermore, we have indicated regions distant to the lesion core that show little or no signs of fibrotic scarring with the hash sign. However, since scarring is frequently observed in a gradient with no sharp boundaries to healthy tissue, we would like to avoid the term 'uninjured tissue' in this context. Instead, we are referring to Supplementary Fig. 3 which shows 'uninjured' brain control autopsy and biopsy tissue without signs of scarring. We have edited the figure legend of Fig. 3 accordingly to incorporate the above changes (page 59, changes marked in yellow).

• *Figure legend Fig. 3:*

Anti-CHAD, anti-FMOD, anti-LUM, and anti-PRELP immunoreactivity is locally increased in areas of scarring caused by contusion, local hemorrhage, or previous surgery (asterisks indicate lesion center, arrowheads mark scar boundaries) in the human brain, as compared to regions distant to the primary lesion site (hash sign), as well as human brain controls with no signs of fibrotic scarring (see Supplementary Fig. 3). Shown are coronal sections of brain tissue from patients with traumatic brain injury (TBI) or previous surgery (re-OP; bottom panel). Scale bars: 500 µm, 50 µm (insets).

3. In the discussion, the authors mentioned the SLRPs might also influence the neural growth factors. I would suggest to compare the data on neuronal growth factor protein or genes at rat or zebrafish spinal cord injury and put these data in a supplementary figure.

RESPONSE: This is a great suggestion. Unfortunately, sensitivity limitations of the mass spectrometry (MS) analysis did not permit the reliable detection of neuronal growth factors across experimental conditions in the zebrafish, such as neurotrophins, neurokinins, fibroblast growth factors, transforming growth factor β superfamily proteins (with the exception of Bmp1a and Tgfb2), epidermal growth factors, etc., which are usually present at very low ng/mL concentrations in tissue samples. Similarly, in the 4M guanidine rat spinal cord tissue extract dataset from Didangelos *et al.* 2017. *bioRxiv*, doi:10.1101/184713 (2017) we detected only very few secreted proteins (Fgf1, FGF12, Tgfa) that may act as neuronal growth factors. We have edited the discussion to point out this limitation of the MS analysis more clearly (page 18, lines 380-387, changes marked in yellow):

[...] Our functional experiments support that SLRPs neither inhibit regeneration through i) direct interactions with the axonal growth cone, nor ii) prevention of inflammation resolution, nor iii) altering the composition of the fibroblast-derived injury ECM. Although we cannot rule out the possibility of SLRPs influencing the availability of neurotrophic factors in the lesion environment due to sensitivity limitations of the mass spectrometry analysis, our *in vivo* experiments provide evidence against this scenario. Specifically, neuron-specific induction of *chad*, *fmod*, *lum*, or *prelp* expression did not impact axonal regrowth. [...]

Reviewer #3 (Remarks to the Author):

The manuscript by Kolb et al provides evidence for an enrichment of Small Leucine-Rich Proteoglycans (SLRPs) in the inhibitory scar that exists in mammals (in this study rats and humans), that may underlie their inability to regenerate. This is indeed a new and important finding because these were previously unknown ECM factors that are expressed upon injury in the context of human CNS.

RESPONSE: We thank the reviewer for evaluating our findings as new and important, as well as for the helpful comments and suggestions to improve the manuscript.

However, to say that “This identifies SLRPs as previously unknown inhibitory ECM factors in the human CNS that impair axon regeneration by altering tissue mechanics and structure.” is an overstating because this is solely based on indirect evidence. All the functional experiments were done in the zebrafish in the context of SLRPs overexpression that show a negative impact on axon regeneration and locomotor recovery and no complementary loss-of-function experiments were done in a mammalian model to show any kind of improvement when these SLRPs were reduced at the lesion core.

RESPONSE: We appreciate the comment and agree with the reviewer that our statement should be phrased more carefully to better reflect the findings of our study. Hence, we have revised the pertinent parts in the abstract (changes marked in yellow) and introduction (pages 4-5, lines 23-31, changes marked in yellow) as follows:

• *Abstract:*

Extracellular matrix (ECM) deposition after central nervous system (CNS) injury leads to inhibitory scarring in mammals, whereas it facilitates axon regeneration in the zebrafish. However, the molecular basis of these different fates is not understood. Here, we identify small leucine-rich proteoglycans (SLRPs) as a causal factor in regeneration failure. We demonstrate that the SLRPs Chondroadherin, Fibromodulin, Lumican, and Prolargin are enriched in **rodent** and human but not zebrafish CNS lesions. Targeting SLRPs to the zebrafish injury ECM inhibits axon regeneration and functional recovery. Mechanistically, we find that SLRPs confer **mechano-structural** properties to the lesion environment that are adverse to axon growth. **Our study reveals SLRPs as previously unknown inhibitory ECM factors that impair axon regeneration by modifying tissue mechanics and structure, and identifies their enrichment as a feature of human brain and spinal cord lesions. These findings imply that SLRPs may be targets for therapeutic strategies to promote CNS regeneration.**

• *Introduction:*

[...] We demonstrate that the **SLRPs** Chondroadherin, Fibromodulin, Lumican, and Prolargin are enriched in human **and rodent** but not zebrafish CNS lesions. Increasing the abundance of SLRPs in the zebrafish injury ECM inhibits axon regeneration and functional recovery. Mechanistically, we find that SLRPs confer **mechano-structural** properties to the lesion environment that render it adverse to axon growth. **Our data identify SLRPs as previously unknown inhibitory ECM factors that impair axon regeneration by altering tissue mechanics and structure, and reveal their enrichment as a feature of human CNS lesions.** Targeting SLRPs therefore presents itself as a potential therapeutic strategy to promote axon growth across CNS lesions. [...]

In a comprehensive proteomic analysis between zebrafish and rats, that was cross validated with RNA in situ hybridizations and qRT-PCR for the zebrafish and immunostainings for brain and spinal cord human samples, the authors were able to demonstrate that the SLRPs

Chondroadherin, Fibromodulin, Lumican, and Prolargin are enriched in rats and humans, but not zebrafish, CNS lesions. This is indeed a new and important finding because these were previously unknown ECM factors in the human CNS that respond to injury. And additionally, makes us think the other way around which is that the regeneration capacity of the zebrafish can be attributed to the lack of axon growth-limiting components rather than the presence of species-specific growth-promoting factors.

RESPONSE: We thank the reviewer again for emphasizing the novelty and importance of our findings.

The authors went on to produce zebrafish transgenic lines that would allow them to overexpress the selected SLRPs under the control of *pdgfrb*-positive fibroblast that were also shown to be the relevant cells that produce these ECM factors upon injury. With this transgenic line the authors were able to induce the expression SLRPs considerably at the lesion site (evaluated by the increased levels of mCherry in Fig.5A) and further show the reduction of the axonal bridge in these animals that was then translated into a smaller swim distance (Fig.5B). The effects are somehow mild and very variable, so I was wondering what would happen if more than one SLRP factors was overexpressed in a combinatorial manner?

RESPONSE: We thank the reviewer for this excellent suggestion. We have performed the experiment, which involved the analysis of in total 369 (live) animals. We found that targeting the two SLRPs Chad and Lum to the injury ECM in a combinatorial manner further enhanced the inhibitory effect of individual SLRPs on axon regeneration and functional recovery. We have added these data as Fig. 5g to the revised manuscript and describe it (page 11, lines 199-202, changes marked in yellow) as follows:

[...] We found that the average axonal bridge thickness was reduced by 41-52% when Chad, Fmoda, Lum, or Prelp was targeted to the injury ECM, as compared to their controls (Fig. 5f; Supplementary Fig. 4g). Moreover, quantification of swimming distance at 2 dpl showed that *pdgfrb*:SLRP transgenics exhibited worse functional recovery than their controls (30-42% reduced swimming distance) (Fig. 5f'). The effect on axon regeneration and functional recovery was further enhanced when two SLRPs (Chad, Lum) were concomitantly targeted to the injury ECM (73% reduced axonal bridge thickness, 60% reduced swimming distance; Fig. 5g). [...]

To investigate if the action of SLRPs on axon regrowth was a direct effect on neurons the authors use the *Xla.Tubb* promotor to overexpress SLRPs on neuronal populations and found that the effect on axonal bridge thickness was negligible (Fig. 6B), however I find puzzling the fact that at the lesion site there is no mCherry signal, which would indicate that the overexpression was not successful (Fig. 6A).

RESPONSE: We apologize for any lack of clarity that may have caused confusion. The absence of mCherry fluorescence signal in the lesion center is a consequence of the deployed SCI paradigm. Our paradigm involves complete transection of the spinal cord, leading to an immediate and massive neuronal loss in the lesion center due to the acute mechanical trauma. This is followed by secondary neuronal death due to necrosis and/or apoptosis towards the edges of the lesion^{4,53,58}. Hence, after SCI (and during the course of regeneration) no or only very few spared neuronal somata are present in the lesion core to secrete SLRPs, explaining the observed negligible mCherry fluorescence signal (also see new Supplementary Fig. 5a-b; and our response to reviewer #1, Fig. R1). Of note, mCherry fluorescence signal is robustly detected distant to the lesion core as well as in the brain (see Fig. 6c and the new Supplementary Fig. 5c). This is in stark contrast to inducing *slrp-mCherry* expression in *pdgfrb*⁺ fibroblasts, which are recruited to the non-neural lesion site (see Fig. 5e)³. We have added new data and edited the manuscript to better explain this:

i) We have edited the results section "*Matrisome dynamics of zebrafish spinal cord regeneration*" (page 5, lines 35-41; changes marked in yellow) to clearly state that our SCI paradigm involves a complete transection, creating a non-neural lesion site:

[...] In order to identify factors that confer axon growth-limiting properties to CNS scars by altering tissue mechanics, we first set out to map the changes in ECM composition in a regeneration context. To that aim, we applied label-free mass spectrometry (MS)-based quantitative proteomics to a larval zebrafish spinal cord injury (SCI) model. In this injury paradigm, the spinal cord is completely transected, which allows axon regeneration across an ECM-rich, non-neural lesion environment and recovery of swimming function within two days post-lesion (dpl; Fig. 1a,b)^{4,59}. [...]

ii) We have edited the result section “SLRPs do not directly act on neurons to inhibit neurite extension” (pages 11-12, lines 213-234; changes marked in yellow):

[...] To elucidate the mechanism by which SLRPs inhibit neurite growth, we first examined a potential direct (biochemical) interaction with neurons, which has been reported for CSPGs and myelin-associated inhibitors⁶⁴⁻⁶⁶. To that aim, we used an *Xla.Tubb* promoter-driven Tet-activator line to target *chad*, *fmoda*, *lum*, or *prelp* expression specifically to neurons, including brainstem descending and spinal interneurons (Fig. 6a; Supplementary Fig. 5a,b)^{4,67}. Induction of *chad-mCherry*, *fmoda-mCherry*, *lum-mCherry*, or *prelp-mCherry* expression in *Xla.Tubb:TetA; TetRE:SLRP-mCherry* transgenic animals (henceforth abbreviated as *Xla.Tubb:SLRP*) led to robust and largely confined fluorescent labeling of the brain and spinal cord (Fig. 6b,c; Supplementary Fig. 5c). Comparing the fluorescence pattern of SLRP-mCherry fusion proteins in *Xla.Tubb:SLRP* transgenics to that of cytoplasmic DsRed proteins in *Xla.Tubb:DsRed* transgenics animals revealed prominent mCherry signal also in DsRed-negative regions of the spinal cord, including the central canal and its ependymal lining (Supplementary Fig. 5d). Moreover, primary cell culture from dissociated *Xla.Tubb:SLRP;elavl3:GFP-F* transgenic animals showed perineuronal mCherry fluorescence (Supplementary Fig. 5e). These data support that neuronal SLRP-mCherry fusion proteins are secreted into the extracellular space. At 1 dpl, we detected negligible mCherry fluorescence in the lesion core of *Xla.Tubb:SLRP* transgenic animals, which is consistent with the acute neuronal loss and secondary cell death following complete spinal cord transection (Fig. 6c, Supplementary Fig. 5b)^{53,58}. Thus, cell type-specific manipulations allow us to distinguish *in vivo* between direct interactions of SLRPs with neurons and indirect effects on axon growth through modulation of the non-neural lesion environment. [...]

iii) We have added new data to Supplementary Fig. 5a-e supporting the above. Furthermore, we carefully edited the figure legend to avoid any potential confusion.

Even more puzzling are the results on Fig. 6C, where the authors show that culturing DRGs in a substrate coated with a combination of the SLRPs (Chondroadherin+ Fibromodulin+ Lumican+ Prolargin) had no effect on neurite outgrowth. If, as the authors propose, these SLRPs are modulating the structural and mechanical properties of the lesion environment (Fig. 7), I would expect to see a strong effect on neurite growth.

RESPONSE: We apologize for any lack of clarity that may have caused confusion. The proposed role of SLRPs in modulating the mechanical properties of the lesion environment is considered to be indirect and is related to their known function in regulating (collagen) fibril growth, fibril formation, and ECM assembly⁶⁸. Our experimental *in vivo* data suggest that the presence of SLRPs in CNS lesions influences the structural properties of injury-induced ECM deposits, which in turn, confers mechanical properties to the ECM that are unfavorable to axon growth. In this context, mechanical properties refer to how the tissue responds to mechanical load, including compression, bending, or shearing.

The aim of the *in vitro* experiment shown in Fig. 6C (now Fig. 6e') was to determine whether SLRPs can inhibit neurite growth by directly interacting with neurons through biochemical signaling (that is receptor-ligand interactions), which has been reported for several other CNS scar components, including CSPGs⁶⁴, myelin-associated glycoprotein (MAG)⁶⁵, and NogoA⁶⁶. To isolate the direct effects of SLRPs on axon regeneration from potential indirect effects through modulating the immune response, ECM composition, or ECM structure/mechanical ECM properties, we used a defined two-dimensional *in vitro* system^{47,68}. We seeded adult primary dorsal root ganglion (DRG) neurons on glass coverslips and assessed neurite outgrowth in the presence of substrate-bound SLRP proteins. Before seeding the DRGs, we adhered laminin proteins, a mixture of laminin and SLRP proteins, or a mixture of laminin and CSPG proteins to the

glass coverslip using a standard protocol that is predicted to apply a monomolecular layer of 3-5 nm thickness⁶⁹. Two main reasons support the exclusion of any relevant impact of SLRP proteins on the mechanical properties (viscoelasticity) of the substrate that are sensed by the cells:

i) Laminin-coated glass coverslips do not recapitulate the complexity of the injury ECM composition, nor its three-dimensional structure. Therefore, the known roles of SLRPs in controlling mechanical ECM properties through regulating ECM structure can be disregarded in this context⁶⁸.

ii) Using glass as a substrate provides a defined and constant stiffness, which is important due to the well-established role of substrate stiffness in influencing neurite extension^{e.g., 70-76}. The elastic modulus (E modulus; describes the material's resistance to deformation as mechanical forces are applied to it) of glass is in the GPa range, which is several orders of magnitude higher than that of CNS tissue (which ranges from 0.1-10 kPa)^{8,73,77}. When a thin layer of proteins is adhered to the glass surface, it may, at best alter the E modulus at the sub-nanoscale. While cells are capable of sensing and interpreting very small changes in the E modulus, they do sense the stiffness of the underlying substrate up until a depth of multiple tens of μm depending on the site of the cell (approximate radius of the cell)⁷⁸. Furthermore, the relative differences between the stiffness of glass and the stiffness of glass with protein coating is too small to be detected by the neurons due to the very high absolute value of the E modulus of the glass substrate. Several studies have shown that neurite extension scales with substrate stiffness, even when the substrate is coated with (laminin) proteins^{e.g., 73,74,79}. In other words, neurons will sense and interpret the E modulus of the glass substrate underneath an adhered thin layer of proteins. Hence, the mechanical properties of the substrate (i.e., the glass) remain unaltered in the presence of adhered SLRP proteins.

To further verify that adhering SLRPs to glass coverslips does not lead to major changes in the E modulus, we performed atomic force microscopy-based nanoindentation. This confirmed the absence of a measurable difference in the E modulus between glass coverslips coated with laminin and a mixture of laminin and SLRPs (see Fig. R4).

Fig. R4: Atomic force microscopy-based nanoindentation measurements confirm the absence of a difference in E modulus between glass coverslips coated with laminin proteins or a mixture of laminin and SLRP proteins.

Furthermore, we have expanded on the two-dimensional (2D) *in vitro* assay. We have added new data showing that the growth medium supplemented with SLRP proteins does neither affect neurite growth negatively. We have added these data to Fig. 6e”.

Finally, we have edited the pertinent parts in the results section (pages 12-13, lines 237-248, changes marked in yellow) and methods section (page 36, lines 859-863, changes marked in yellow) in order to provide more detail and make the manuscript easier to follow.

• **Results:**

[...] To corroborate these findings, we seeded adult primary murine dorsal root ganglion (DRGs) neurons on laminin-coated glass coverslips and assessed neurite outgrowth in the presence of soluble or substrate-bound SLRPs. These 2D *in vitro* assays allow to test primarily for direct biochemical (ligand-receptor) interactions of SLRPs with the axonal growth cone. Potential indirect effects of SLRPs on axon regeneration through modulating inflammation, ECM composition, ECM assembly, or mechanical ECM properties, can be excluded. Although substrate-bound high molecular weight CSPGs potently reduced the average neurite length by 66%, a mixture of human CHAD, FMOD, LUM, and PRELP proteins had no effect (Fig. 6e'). Similarly, average neurite length was not inhibited when SLRP proteins were supplemented to the growth medium but decreased by 29% in the presence of soluble CSPGs (Fig. 6e''). Taken together, *in vivo* and *in vitro* data support that SLRPs inhibit axon growth indirectly. [...]

• **Methods:**

[...] For coating of glass coverslips, a standard protocol was employed that is predicted to apply a monomolecular layer of 3-5 nm thickness⁶⁹. Alternatively, DRG neurons were seeded on glass coverslips pre-coated with PDL and laminin and cultured in the presence of either 1 µg/ml CSPGs, a mix of 1 µg/ml of each SLRP protein, or solvent control. [...]

Reviewer #4 (Remarks to the Author):

In this paper, the authors use proteomics to study spinal cord regeneration in zebrafish. Comparing this data to mammalian data, they identify extracellular matrix proteins which are up-regulated after spinal cord injury in mammals, including rat and humans, but not in zebrafish. The authors show that overexpression of these proteins in zebrafish impairs spinal cord regeneration, and changes the mechanical properties of the injury. The paper has a very good quality. The experiments are in general performed very carefully, and well presented. Their results are potentially relevant for human health, thus highlighting the potential impact of the publication. However, there are some points that should be addressed by the authors.

RESPONSE: We thank the reviewer for their very favorable opinion about our work, it's very good quality and potential impact, and for their suggestions to further improve the manuscript.

1. Figure S1C and S1D. Points are distributed in a surprisingly clustered way. The authors should revise the analysis to find if there is something that can be improved. One possibility that may explain this clustering is the following. Log2FC cannot be calculated if the expression in one of the samples is zero, so a small value, for example, 0.001 counts, is usually added to all the samples. When this value is too low, this kind of “weird” clusters may appear. I suggest that the authors revise the analysis, and add a higher value if it makes sense, which may reduce these artifacts. In any case, I would not expect that the main conclusions change after these small changes.

RESPONSE: We thank the reviewer for the helpful suggestion. The unexpected clustering in the presented volcano plot was caused by the relatively low stringency of our filtering strategy. We have revised the analysis of the mass spectrometry data to exclude all proteins that exhibit ≥ 1 invalid value (with the exception of proteins that showed invalid values in all three replicates of one experimental condition). This has improved the clustering without affecting the main conclusions of the manuscript. We have also repeated the Reactome pathway analysis. We have updated Supplementary Fig. 1 and the corresponding figure legend. Furthermore, we have edited the pertinent parts of the results and methods section of the manuscript (pages 30-31, lines 716-720; page 5, lines 41-48; changes marked in yellow) as follows:

• *Methods:*

[...] LFIQ intensity values were imported to Perseus software (version 1.6.14.0)⁸⁰. Filters were set to exclude proteins identified by site, matching to the reverse database, or contaminants. For the identification of injury-induced changes in protein abundance (lesioned vs. unlesioned), proteins that exhibited ≥ 1 invalid value were excluded from further analysis. However, we included proteins that showed invalid values in all three replicates of one experimental condition (either lesioned or unlesioned). This was necessary to include proteins undetectable in unlesioned animals but with enrichment after SCI. For identification of changes in protein abundance caused by cell-type specific induction of SLRPs, all proteins that exhibited ≥ 1 invalid value were excluded. [...]

• *Results:*

[...] Proteomic profiling of trunk tissue containing the lesion site at 1 dpl and 2 dpl as well as corresponding age-matched unlesioned control tissue identified 4782 unique proteins after quality control (Supplementary Fig. 1a,b). Differential abundance analysis (FDR < 0.1, |FC| \geq 1.3, s_0 = 0.1) revealed 910 proteins whose abundance was altered in 1 dpl compared to unlesioned control samples (573 up- and 337 down-regulated; Supplementary Fig. 1c). The abundance of 656 proteins differed between 2 dpl and unlesioned control samples (443 up- and 213 down-regulated; Supplementary Fig. 1d). [...]

2. In section “SLRPs are abundant in human CNS lesions”, the authors do not analyze anymore the Dcn and Ogn proteins. The authors should justify why they decided to focus on the other four proteins, and not to continue the Dcn and Ogn analysis.

RESPONSE: We thank the reviewer for the comment. We have added the following statement to the revised manuscript, which explains our decision to concentrate on the SLRPs Chad, Fmod, Lum, and Prelp (page 8, lines 121-126, changes marked in yellow):

[...] To obtain further evidence that the capacity for axon regeneration scales with SLRP protein abundance, we analyzed their expression after SCI in adult zebrafish and adult mice, which possess high and low regenerative capacity for the CNS, respectively. We decided to concentrate on Fmod, Lum, and Prelp due to their structural similarities as class II SLRPs. Additionally, we focused on Chad which showed the highest abundance among SLRPs in the rat spinal lesion site (Fig. 2a). [...]

3. I can find some problems in the interpretation of the section “SLRPs do not directly act on neurons to inhibit neurite extension”. In the first part (Fig 6A and 6B), the authors show that expressing SLRPs in neurons has no effect on axons. This experiment does not rule out a direct action of SLRPs in neurons, it could just be that neurons are not able to secrete SLRPs, and SLRPs may only have an effect if they are secreted, so this experiment is not very conclusive.

RESPONSE: We thank the reviewer for the comment. We agree with the reviewer that SLRPs need to be secreted into the extracellular space in order to fulfill their function. This is consistent with a recent paper by the research group of Jeffrey D. Macklis, showing that *lumican* is expressed by neurons to control non-cell-autonomously axon collateralization⁶¹. To verify secretion of SLRP-mCherry fusion protein by zebrafish neurons, we have performed the following additional experiments:

i) We compared the fluorescence pattern of SLRP-mCherry fusion proteins in *Xla.Tubb:SLRP* transgenics with that of cytoplasmic DsRed proteins in *Xla.Tubb:DsRed* transgenics animals on transversal spinal cord sections. This clearly showed mCherry signal in DsRed-negative regions of the spinal cord, including the central canal and its ependymal lining. We conclude that SLRP-mCherry fusion proteins are secreted since the central canal is surrounded by ependymo-radial glia cells in which the *Xla.Tubb* promoter is inactive. We have added these data as Supplementary Fig. 5d.

ii) We prepared primary neuronal culture from dissociated *Xla.Tubb:SLRP-mCherry* transgenic animals in which all neurons were additionally labeled with membrane-tethered GFP (*elavl3:GFP-F*). This revealed perineuronal mCherry fluorescence. We have added these data as Supplementary Fig. 5e.

Together, these data support that neurons are capable of secreting SLRP-mCherry fusion protein. We describe these new data in the revised manuscript (page 12, lines 218-228, changes marked in yellow) as follows:

[...] Induction of *chad-mCherry*, *fmoda-mCherry*, *lum-mCherry*, or *prelp-mCherry* expression in *Xla.Tubb:TetA;TetRE:SLRP-mCherry* transgenic animals (henceforth abbreviated as *Xla.Tubb:SLRP*) led to robust and largely confined fluorescent labeling of the brain and spinal cord (Fig. 6b,c; Supplementary Fig. 5c). Comparing the fluorescence pattern of SLRP-mCherry fusion proteins in *Xla.Tubb:SLRP* transgenics to that of cytoplasmic DsRed proteins in *Xla.Tubb:DsRed* transgenics animals revealed prominent mCherry signal also in DsRed-negative regions of the spinal cord, including the central canal and its ependymal lining (Supplementary Fig. 5d). Moreover, primary cell culture from dissociated *Xla.Tubb:SLRP;elavl3:GFP-F* transgenic animals showed perineuronal mCherry fluorescence (Supplementary Fig. 5e). These data support that neuronal SLRP-mCherry fusion proteins are secreted into the extracellular space. [...]

I do not think that the second part of the section (Fig. 6C) corroborates the same findings. It is just a different experiment, which is more conclusive, showing that extracellular SLRPs do not affect axon growth. However, the interpretation of this experiment also generates some doubts. I would expect that adding these proteins would affect the mechanical properties of the plate, and according to the main claim of the paper, changes in these mechanical properties should affect axon regeneration. How do the authors explain the absence of effect?

RESPONSE: We apologize for any lack of clarity that may have caused confusion. The proposed role of SLRPs in modulating the mechanical properties of the lesion environment is considered to be indirect and is related to their known function in regulating (collagen) fibril growth, fibril formation, and ECM assembly⁶⁸. Our experimental *in vivo* data suggest that the presence of SLRPs in CNS lesions influences the structural properties of injury-induced ECM deposits, which in turn, confers mechanical properties to the ECM that are unfavorable to axon growth. In this context, mechanical properties refer to how the tissue responds to mechanical load, including compression, bending, or shearing.

The aim of the *in vitro* experiment shown in Fig. 6C (now Fig. 6e') was to determine whether SLRPs can inhibit neurite growth by directly interacting with neurons through biochemical signaling (that is receptor-ligand interactions), which has been reported for several other CNS scar components, including CSPGs⁶⁴, myelin-associated glycoprotein (MAG)⁶⁵, and NogoA⁶⁶. To isolate the direct effects of SLRPs on axon regeneration from potential indirect effects through modulating the immune response, ECM composition, or ECM structure/mechanical ECM properties, we used a defined two-dimensional *in vitro* system^{47,68}. We seeded adult primary dorsal root ganglion (DRG) neurons on glass coverslips and assessed neurite outgrowth in the presence of substrate-bound SLRP proteins. Before seeding the DRGs, we adhered laminin proteins, a mixture of laminin and SLRP proteins, or a mixture of laminin and CSPG proteins to the glass coverslip using a standard protocol that is predicted to apply a monomolecular layer of 3-5 nm thickness⁶⁹. Two main reasons support the exclusion of any relevant impact of SLRP proteins on the mechanical properties (viscoelasticity) of the substrate that are sensed by the cells:

i) Laminin-coated glass coverslips do not recapitulate the complexity of the injury ECM composition, nor its three-dimensional structure. Therefore, the known roles of SLRPs in controlling mechanical ECM properties through regulating ECM structure can be disregarded in this context⁶⁸.

ii) Using glass as a substrate provides a defined and constant stiffness, which is important due to the well-established role of substrate stiffness in influencing neurite extension^{e.g., 70-76}. The elastic modulus (E modulus; describes the material's resistance to deformation as mechanical forces are applied to it) of glass is in the GPa range, which is several orders of magnitude higher than that of CNS tissue (which ranges from 0.1-10 kPa)^{8,73,77}. When a thin layer of proteins is adhered to the glass surface, it may, at best alter the E modulus at the sub-nanoscale. While cells are capable of sensing and interpreting very small changes in the E modulus, they do sense the stiffness of the underlying substrate up until a depth of multiple tens of μm depending on the site of the cell (approximate radius of the cell)⁷⁸. Furthermore, the relative differences between the stiffness of glass and the stiffness of glass with protein coating is too small to be detected by the neurons due to the very high absolute value of the E modulus of the glass substrate. Several studies have shown that neurite extension scales with substrate stiffness, even when the substrate is coated with (laminin) proteins^{e.g., 73,74,79}. In other words, neurons will sense and interpret the E modulus of the glass substrate underneath an adhered thin layer of proteins. Hence, the mechanical properties of the substrate (i.e., the glass) remain unaltered in the presence of adhered SLRP proteins.

To further verify that adhering SLRPs to glass coverslips does not lead to major changes in the E modulus, we performed atomic force microscopy-based nanoindentation. This confirmed the absence of a measurable difference in the E modulus between glass coverslips coated with laminin and a mixture of laminin and SLRPs (see Fig. R5).

Fig. R5: Atomic force microscopy-based nanoindentation measurements confirm the absence of a difference in E modulus between glass coverslips coated with laminin proteins or a mixture of laminin and SLRP proteins.

Furthermore, we have expanded on the two-dimensional (2D) *in vitro* assay. We have added new data showing that the growth medium supplemented with SLRP proteins does neither affect neurite growth negatively. We have added these data to Fig. 6e”.

Finally, we have edited the pertinent parts in the results section (pages 12-13, lines 237-248, changes marked in yellow) and methods section (page 36, lines 859-863, changes marked in yellow) in order to provide more detail and make the manuscript easier to follow.

• **Results:**

[...] To corroborate these findings, we seeded adult primary murine dorsal root ganglion (DRGs) neurons on laminin-coated glass coverslips and assessed neurite outgrowth in the presence of soluble or substrate-bound SLRPs. These 2D *in vitro* assays allow to test primarily for direct biochemical (ligand-receptor) interactions of SLRPs with the axonal growth cone. Potential indirect effects of SLRPs on axon regeneration through modulating inflammation, ECM composition, ECM assembly, or mechanical ECM properties, can be excluded. Although substrate-bound high molecular weight CSPGs potentially reduced the average neurite length by 66%, a mixture of human CHAD, FMOD, LUM, and PRELP proteins had no effect (Fig. 6e’). Similarly, average neurite length was not inhibited when SLRP proteins were supplemented to the growth medium but decreased by 29% in the presence of soluble CSPGs (Fig. 6e”). Taken together, *in vivo* and *in vitro* data support that SLRPs inhibit axon growth indirectly. [...]

• **Methods:**

[...] For coating of glass coverslips, a standard protocol was employed that is predicted to apply a monomolecular layer of 3-5 nm thickness⁶⁹. Alternatively, DRG neurons were seeded on glass coverslips pre-coated with PDL and laminin and cultured in the presence of either 1 µg/ml CSPGs, a mix of 1 µg/ml of each SLRP protein, or solvent control. [...]

The authors should explain better the implications of each of the experiments, and interpret them more carefully following the indications above.

RESPONSE: We hope that our explanations, new data, and amendments to the manuscript have sufficiently addressed the reviewer’s concerns.

4. After a first reading, it is difficult to understand why the authors do the MTZ experiment (S6A), as it is not explained in the main text. After reading the S6A figure legend, it becomes

clear that it is just a positive control. The authors could make easier the understanding of this experiment by including a sentence in the main text. Also, this TUNEL experiment is not really connected with the title and conclusions of its section: “SLRPs do not alter the fibroblast response”. TUNEL signal is not detected in fibroblasts, but it is also not detected in any other tissue.

RESPONSE: We thank the reviewer for the helpful comment and apologize for any lack of clarity. As the reviewer correctly pointed out, the rationale for the experiment presented in Fig. S6A (now Supplementary Fig. 4f) was to confirm that our TUNEL protocol provides sufficient sensitivity to efficiently detect apoptotic *pdgfrb*⁺ cells in whole-mount preparations of uninjured animals. Hence, this experiment serves as a positive control for the experiment presented in Fig. S6B (now Supplementary Fig. 4e), which demonstrates that TetON system-driven expression of the SLRPs *chad*, *fmoda*, *lum*, or *prelp* in *pdgfrb*⁺ cells does not lead to enhanced cell death of these cells. We consider this experiment important since *pdgfrb*⁺ cells are recruited to the zebrafish spinal lesion site and deposit a growth-promoting injury ECM that is required for axon regeneration^{3,4}. We agree with the reviewer that the description of both datasets should be improved to enhance understanding of the manuscript. Hence, we have made the following amendments:

1) we have moved the data to Supplementary Fig. 4f, and describe them in the revised manuscript in more detail already in the results section “SLRPs are inhibitory to CNS regeneration” (page 10, lines 172-186, changes marked in yellow):

[...] Induction of *aspn-mCherry*, *chad-mCherry*, *fmoda-mCherry*, *lum-mCherry*, or *prelp-mCherry* expression in unlesioned *pdgfrb*:SLRP transgenic animals resulted in fluorescent labeling of mainly the myosepta and vasculature, thus resembling the *pdgfrb*⁺ cell niche (Supplementary Fig. 4b,c)³. 3D reconstruction of high-resolution images of *pdgfrb*:SLRP;*pdgfrb*:GFP transgenics revealed accumulation of mCherry signal in the pericellular environment of *pdgfrb*⁺ cells (Supplementary Fig. 4d). Furthermore, we observed variations in the mCherry fluorescence pattern among different SLRP proteins, including distinct mesh and fiber-like structures in the case of *Aspn*, *Fmoda*, and *Prelp* (Supplementary Fig. 4b,d). Together, this supports the secretion and binding of the SLRP-mCherry fusion proteins to fibrillar ECM⁶⁰. Importantly, using the TUNEL assay showed that TetON system-driven expression of *chad*, *fmoda*, *lum*, or *prelp* did not lead to an increase in the number of apoptotic *pdgfrb*⁺ cells over baseline levels (Supplementary Fig. 4e). Of note, our protocol was sufficiently sensitive to detect *pdgfrb*⁺/TUNEL⁺ cells in whole-mount preparations following nitroreductase-mediated targeted ablation in unlesioned *pdgfrb*:GFP;*pdgfrb*:NTR-mCherry transgenic animals (Supplementary Fig. 4f). [...]

2) we have edited the figure legend title of the revised Fig. S6 (now Supplementary Fig. 7) to indicate that induction of *chad*, *fmoda*, *lum*, or *prelp* expression in *pdgfrb*⁺ cells does not prevent their recruitment to the lesion site.

5. In S7D, initially, it is also difficult to understand why the authors choose 5 different genes for each experiment, instead of analyzing the 25 genes in the 4 different experiments. It would be helpful to add one sentence in the main text explaining this. Also, I count 20 genes, not 25.

RESPONSE: We apologize for any lack of clarity and this oversight. As the reviewer correctly pointed out, in Fig. S7D (now Supplementary Fig. 8d), we analyzed five different genes for each of the four presented mass spectrometry (MS) analyses. Each experiment corresponds to one condition, that is induction of one particular SLRP (*chad/fmoda/lum/prelp*) in *pdgfrb*⁺ fibroblasts in comparison to their respective controls. Since the MS analysis did not yield any significant changes in matrixome protein abundance, we decided to further verify the results using *in situ* hybridization (ISH). To that aim, we chose five proteins that showed the highest (yet non-significant) fold change in each of the four experimental conditions. We did not analyze the same set of genes for all experimental conditions because the proteins exhibiting the highest (yet non-significant) fold change differed among experimental conditions. We have edited the results section “SLRPs do not alter the fibroblast response” (pages 13-14, lines 264-282, changes marked in yellow; also see our response to your point #4) and the legend of Supplementary Fig. 8d to point this out more clearly:

• **Results:**

[...] Accumulation of *pdgfrb*⁺ fibroblasts in the lesion site and secretion of a growth-conductive ECM is critical for axon regeneration after SCI in the zebrafish^{3,4}. Hence, we sought to determine whether SLRPs inhibit axon regeneration by altering the fibroblast response in *pdgfrb*:SLRP transgenic animals. We did not detect significant changes in the area coverage of *pdgfrb*⁺ fibroblasts in the lesion site at 1 dpl (analyzed in *pdgfrb*:GFP transgenics), indicating that their recruitment was largely unperturbed (Supplementary Fig. 7). This is consistent with our findings that *pdgfrb*⁺ cell-specific induction of *slrp-mCherry* fusions does not trigger apoptosis of *pdgfrb*⁺ cells (Supplementary Fig. 4e). We next analyzed the composition of the injury ECM in *pdgfrb*:SLRP transgenic animals at 1 dpl, using MS-based quantitative proteomics. We found that induction of either *chad*, *fmoda*, *lum*, or *prelp* in *pdgfrb*⁺ cells did not lead to significant changes (FDR < 0.1, |FC| ≥ 1.3) in the abundance of detected matrisome proteins as compared to their respective controls (Supplementary Fig. 8a,b,c). To further validate the MS results, we used ISH to evaluate the expression pattern and transcript levels of five genes, coding for matrisome proteins that showed the highest (yet non-significant) fold change in each of the four experimental conditions. Consistent with the MS analysis, ISH signals in the lesion site were comparable between *pdgfrb*:SLRP transgenics and their respective controls for all genes analyzed (Supplementary Fig. 8d). We thus conclude that targeting SLRPs to *pdgfrb*⁺ cells does not lead to major changes in the biochemical composition of the injury ECM. [...]

• **Supplementary Fig. 8d legend:**

Induction of *chad*, *fmoda*, *lum*, or *prelp* in *pdgfrb*:SLRP transgenic zebrafish does not lead to major changes in the expression (blue) of indicated genes in the lesion site at 1 dpl, as determined by ISH. Transcript levels of five genes coding for matrisome proteins that showed the highest (yet non-significant) fold change in each of the four experimental conditions in (c) were evaluated. The number of specimens displaying the phenotype and the total number of experimental specimens is given. Black arrowheads indicate ISH signal in the center of the lesion site, white arrowheads indicate absence of ISH signal. Images shown are brightfield recordings of the lesion site (lateral view; rostral is left). Scale bars: 100 μm.

6. Brillouin microscopy experiment are correctly performed. However, not all the required information is displayed. In Fig 7C, the authors should show a representative image of each of the conditions that they analyze, as they do in Fig 7B. The same applies to Fig 8. Also, the overlay of brightfield, Brillouin frequency shift and fluorescence in 7C is almost impossible to see. The authors should show this information in separated channels.

RESPONSE: We thank the reviewer for the helpful comment. We have edited Fig. 7c and Fig. S8 (now Supplementary Fig. 9) to implement the suggested changes. Furthermore, we have updated the figure legends accordingly.

7. In Fig. 7C and Fig S8, the authors should avoid the use of the “jet” colorscale, and use instead viridis, inferno, magma or plasma, which are perceptually uniform and robust to colorblindness.

RESPONSE: We apologize for this oversight. In the revised manuscript we have used lookup tables that are robust to color blindness (Fig. 7c, viridis; Supplementary Fig. S8 [now Supplementary Fig. 9], plasma).

8. Although the authors provide information about how the plasmids were generated, it would not be possible to exactly reproduce these experiments unless the authors provide the full sequence of the plasmids they used. The authors should attach the plasmids maps as supplementary data.

RESPONSE: We thank the reviewer for the helpful suggestion. In the revised manuscript, we have added the plasmid maps and sequences of the constructs used to generate the transgenic zebrafish lines reported in this study as Supplementary Data 2 to 6. Furthermore, we refer to the data in the respective section of the methods (page 22, lines 482-483, changes marked in yellow):

[...] All primer sequences for molecular cloning are given in Supplementary Data 1, plasmid maps and sequences are given in Supplementary Data 2 to 6. [...]

References

1. Tsata, V., and Wehner, D. (2021). Know How to Regrow-Axon Regeneration in the Zebrafish Spinal Cord. *Cells* *10*. 10.3390/cells10061404.
2. Becker, T., Lieberoth, B.C., Becker, C.G., and Schachner, M. (2005). Differences in the regenerative response of neuronal cell populations and indications for plasticity in intraspinal neurons after spinal cord transection in adult zebrafish. *Molecular and cellular neurosciences* *30*, 265-278. 10.1016/j.mcn.2005.07.008.
3. Tsata, V., Möllmert, S., Schweitzer, C., Kolb, J., Möckel, C., Böhm, B., Rosso, G., Lange, C., Lesche, M., Hammer, J., et al. (2021). A switch in pdgfrb+ cell-derived ECM composition prevents inhibitory scarring and promotes axon regeneration in the zebrafish spinal cord. *Developmental cell* *56*, 509-524.e509. 10.1016/j.devcel.2020.12.009.
4. Wehner, D., Tsarouchas, T.M., Michael, A., Haase, C., Weidinger, G., Reimer, M.M., Becker, T., and Becker, C.G. (2017). Wnt signaling controls pro-regenerative Collagen XII in functional spinal cord regeneration in zebrafish. *Nature communications* *8*, 126. 10.1038/s41467-017-00143-0.
5. Didangelos, A., Bartus, K., Tica, J., Puglia, M., Roschitzki, B., and Bradbury, E.J. (2017). Rats and axolotls share a common molecular signature after spinal cord injury enriched in collagen-1. *bioRxiv*. 10.1101/184713
6. Möllmert, S., Kharlamova, M.A., Hoche, T., Taubenberger, A.V., Abuhattum, S., Kuscha, V., Kurth, T., Brand, M., and Guck, J. (2020). Zebrafish Spinal Cord Repair Is Accompanied by Transient Tissue Stiffening. *Biophysical journal* *118*, 448-463. 10.1016/j.bpj.2019.10.044.
7. Baumann, H.J., Mahajan, G., Ham, T.R., Betonio, P., Kothapalli, C.R., Shriver, L.P., and Leipzig, N.D. (2020). Softening of the chronic hemi-section spinal cord injury scar parallels dysregulation of cellular and extracellular matrix content. *J Mech Behav Biomed Mater* *110*, 103953. 10.1016/j.jmbm.2020.103953.
8. Moendarbary, E., Weber, I.P., Sheridan, G.K., Koser, D.E., Soleman, S., Haenzi, B., Bradbury, E.J., Fawcett, J., and Franze, K. (2017). The soft mechanical signature of glial scars in the central nervous system. *Nature communications* *8*, 14787. 10.1038/ncomms14787.
9. Becker, T., Wullmann, M.F., Becker, C.G., Bernhardt, R.R., and Schachner, M. (1997). Axonal regrowth after spinal cord transection in adult zebrafish. *The Journal of comparative neurology* *377*, 577-595.
10. Bernstein, J.J. (1964). Relation of Spinal Cord Regeneration to Age in Adult Goldfish. *Experimental neurology* *9*, 161-174. 10.1016/0014-4886(64)90014-7.
11. Hanslik, K.L., Allen, S.R., Harkenrider, T.L., Fogerson, S.M., Guadarrama, E., and Morgan, J.R. (2019). Regenerative capacity in the lamprey spinal cord is not altered after a repeated transection. *PloS one* *14*, e0204193. 10.1371/journal.pone.0204193.
12. Doyle, L.M., Stafford, P.P., and Roberts, B.L. (2001). Recovery of locomotion correlated with axonal regeneration after a complete spinal transection in the eel. *Neuroscience* *107*, 169-179. 10.1016/s0306-4522(01)00402-x.
13. Yamada, H., Miyake, T., and Kitamura, T. (1995). Regeneration of axons in transection of the carp spinal cord. *Zoolog Sci* *12*, 325-332. 10.2108/zsj.12.325.
14. Vanhunsel, S., Bergmans, S., and Moons, L. (2022). Killifish switch towards mammalian-like regeneration upon aging. *Aging (Albany NY)* *14*, 2924-2925. 10.18632/aging.203995.
15. Shimizu, Y., and Kawasaki, T. (2021). Differential Regenerative Capacity of the Optic Tectum of Adult Medaka and Zebrafish. *Front Cell Dev Biol* *9*, 686755. 10.3389/fcell.2021.686755.
16. Lust, K., and Wittbrodt, J. (2018). Activating the regenerative potential of Muller glia cells in a regeneration-deficient retina. *eLife* *7*. 10.7554/eLife.32319.
17. Edwards-Faret, G., Gonzalez-Pinto, K., Cebrian-Silla, A., Penailillo, J., Garcia-Verdugo, J.M., and Larrain, J. (2021). Cellular response to spinal cord injury in regenerative and non-regenerative stages in *Xenopus laevis*. *Neural development* *16*, 2. 10.1186/s13064-021-00152-2.
18. Lee-Liu, D., Mendez-Olivos, E.E., Munoz, R., and Larrain, J. (2017). The African clawed frog *Xenopus laevis*: A model organism to study regeneration of the central nervous system. *Neuroscience letters* *652*, 82-93. 10.1016/j.neulet.2016.09.054.
19. Endo, T., Yoshino, J., Kado, K., and Tochinal, S. (2007). Brain regeneration in anuran amphibians. *Dev Growth Differ* *49*, 121-129. 10.1111/j.1440-169X.2007.00914.x.
20. Duffy, M.T., Simpson, S.B., Jr., Liebich, D.R., and Davis, B.M. (1990). Origin of spinal cord axons in the lizard regenerated tail: supernormal projections from local spinal neurons. *The Journal of comparative neurology* *293*, 208-222. 10.1002/cne.902930205.
21. Lozito, T.P., Londono, R., Sun, A.X., and Hudnall, M.L. (2021). Introducing dorsoventral patterning in adult regenerating lizard tails with gene-edited embryonic neural stem cells. *Nature communications* *12*, 6010. 10.1038/s41467-021-26321-9.
22. Gilbert, E.A.B., and Vickaryous, M.K. (2018). Neural stem/progenitor cells are activated during tail regeneration in the leopard gecko (*Eublepharis macularius*). *The Journal of comparative neurology* *526*, 285-309. 10.1002/cne.24335.
23. Shen, T., Wang, Y., Zhang, Q., Bai, X., Wei, S., Zhang, X., Wang, W., Yuan, Y., Liu, Y., Liu, M., et al. (2017). Potential Involvement of Snail Members in Neuronal Survival and Astrocytic Migration during the Gecko Spinal Cord Regeneration. *Frontiers in cellular neuroscience* *11*, 113. 10.3389/fncel.2017.00113.
24. Nogueira-Rodrigues, J., Leite, S.C., Pinto-Costa, R., Sousa, S.C., Luz, L.L., Sintra, M.A., Oliveira, R., Monteiro, A.C., Pinheiro, G.G., Vitorino, M., et al. (2022). Rewired glycosylation activity promotes scarless regeneration and functional recovery in spiny mice after complete spinal cord transection. *Developmental cell* *57*, 440-450 e447. 10.1016/j.devcel.2021.12.008.
25. Streeter, K.A., Sunshine, M.D., Brant, J.O., Sandoval, A.G.W., Maden, M., and Fuller, D.D. (2020). Molecular and histologic outcomes following spinal cord injury in spiny mice, *Acomys cahirinus*. *The Journal of comparative neurology* *528*, 1535-1547. 10.1002/cne.24836.
26. Zukor, K.A., Kent, D.T., and Odelberg, S.J. (2011). Meningeal cells and glia establish a permissive environment for axon regeneration after spinal cord injury in newts. *Neural development* *6*, 1. 10.1186/1749-8104-6-1.
27. Zaky, A.Z., and Mofteh, M.Z. (2014). Neurogenesis and growth factors expression after complete spinal cord transection in *Pleurodeles waltlii*. *Frontiers in cellular neuroscience* *8*, 458. 10.3389/fncel.2014.00458.
28. McHedlishvili, L., Epperlein, H.H., Telzerow, A., and Tanaka, E.M. (2007). A clonal analysis of neural progenitors during axolotl spinal cord regeneration reveals evidence for both spatially restricted and multipotent

- progenitors. *Development* 134, 2083-2093. 10.1242/dev.02852.
29. Clarke, J.D., Alexander, R., and Holder, N. (1988). Regeneration of descending axons in the spinal cord of the axolotl. *Neuroscience letters* 89, 1-6. 10.1016/0304-3940(88)90471-5.
 30. Cura Costa, E., Otsuki, L., Rodrigo Albers, A., Tanaka, E.M., and Chara, O. (2021). Spatiotemporal control of cell cycle acceleration during axolotl spinal cord regeneration. *eLife* 10. 10.7554/eLife.55665.
 31. Tachibana, M., and Nishizuka, Y. (1978). [Dr. Osamu Hayaishi; his work and opinions (2)]. *Seikagaku* 50, 1257-1265.
 32. Jha, R.M., Kochanek, P.M., and Simard, J.M. (2019). Pathophysiology and treatment of cerebral edema in traumatic brain injury. *Neuropharmacology* 145, 230-246. 10.1016/j.neuropharm.2018.08.004.
 33. Stokum, J.A., Gerzanich, V., and Simard, J.M. (2016). Molecular pathophysiology of cerebral edema. *J Cereb Blood Flow Metab* 36, 513-538. 10.1177/0271678X15617172.
 34. Dalby, T., Wohl, E., Dinsmore, M., Unger, Z., Chowdhury, T., and Venkatraghavan, L. (2020). Pathophysiology of Cerebral Edema—A Comprehensive Review. *J Neuroanaesth Crit Care* 08, 163-172. 10.1055/s-0040-1721165.
 35. Bradbury, E.J., and Burnside, E.R. (2019). Moving beyond the glial scar for spinal cord repair. *Nature communications* 10, 3879. 10.1038/s41467-019-11707-7.
 36. Dias, D.O., and Göritz, C. (2018). Fibrotic scarring following lesions to the central nervous system. *Matrix biology : journal of the International Society for Matrix Biology* 68-69, 561-570. 10.1016/j.matbio.2018.02.009.
 37. Tran, A.P., Warren, P.M., and Silver, J. (2022). New insights into glial scar formation after spinal cord injury. *Cell and tissue research* 387, 319-336. 10.1007/s00441-021-03477-w.
 38. Fawcett, J.W., Schwab, M.E., Montani, L., Brazda, N., and Muller, H.W. (2012). Defeating inhibition of regeneration by scar and myelin components. *Handbook of clinical neurology* 109, 503-522. 10.1016/B978-0-444-52137-8.00031-0.
 39. O'Shea, T.M., Burda, J.E., and Sofroniew, M.V. (2017). Cell biology of spinal cord injury and repair. *The Journal of clinical investigation* 127, 3259-3270. 10.1172/JCI90608.
 40. Tica, J., and Didangelos, A. (2018). Comparative Transcriptomics of Rat and Axolotl After Spinal Cord Injury Dissects Differences and Similarities in Inflammatory and Matrix Remodeling Gene Expression Patterns. *Front Neurosci* 12, 808. 10.3389/fnins.2018.00808.
 41. Wehner, D., and Becker, C.G. (2022). An exception to the rule? Regeneration of the injured spinal cord in the spiny mouse. *Developmental cell* 57, 415-416. 10.1016/j.devcel.2022.02.002.
 42. Dias, D.O., Kim, H., Holl, D., Werne Solnestam, B., Lundeberg, J., Carlen, M., Goritz, C., and Frisen, J. (2018). Reducing Pericyte-Derived Scarring Promotes Recovery after Spinal Cord Injury. *Cell* 173, 153-165 e122. 10.1016/j.cell.2018.02.004.
 43. Bradbury, E.J., Moon, L.D., Popat, R.J., King, V.R., Bennett, G.S., Patel, P.N., Fawcett, J.W., and McMahon, S.B. (2002). Chondroitinase ABC promotes functional recovery after spinal cord injury. *Nature* 416, 636-640. 10.1038/416636a.
 44. Klapka, N., Hermanns, S., Straten, G., Masannek, C., Duis, S., Hamers, F.P., Muller, D., Zuschratter, W., and Muller, H.W. (2005). Suppression of fibrous scarring in spinal cord injury of rat promotes long-distance regeneration of corticospinal tract axons, rescue of primary motoneurons in somatosensory cortex and significant functional recovery. *The European journal of neuroscience* 22, 3047-3058. 10.1111/j.1460-9568.2005.04495.x.
 45. Ruschel, J., Hellal, F., Flynn, K.C., Dupraz, S., Elliott, D.A., Tedeschi, A., Bates, M., Sliwinski, C., Brook, G., Dobrindt, K., et al. (2015). Axonal regeneration. Systemic administration of epothilone B promotes axon regeneration after spinal cord injury. *Science* 348, 347-352. 10.1126/science.aaa2958.
 46. Stichel, C.C., Hermanns, S., Luhmann, H.J., Lausberg, F., Niermann, H., D'Urso, D., Servos, G., Hartwig, H.G., and Muller, H.W. (1999). Inhibition of collagen IV deposition promotes regeneration of injured CNS axons. *The European journal of neuroscience* 11, 632-646.
 47. Francos-Quijorna, I., Sanchez-Petidier, M., Burnside, E.R., Badea, S.R., Torres-Espin, A., Marshall, L., de Winter, F., Verhaagen, J., Moreno-Manzano, V., and Bradbury, E.J. (2022). Chondroitin sulfate proteoglycans prevent immune cell phenotypic conversion and inflammation resolution via TLR4 in rodent models of spinal cord injury. *Nature communications* 13, 2933. 10.1038/s41467-022-30467-5.
 48. Liebscher, T., Schnell, L., Schnell, D., Scholl, J., Schneider, R., Gullo, M., Fouad, K., Mir, A., Rausch, M., Kindler, D., et al. (2005). Nogo-A antibody improves regeneration and locomotion of spinal cord-injured rats. *Annals of neurology* 58, 706-719. 10.1002/ana.20627.
 49. Wei, Y., and Andrews, M.R. (2022). Advances in chondroitinase delivery for spinal cord repair. *J Integ Neurosci* 21, 118. 10.31083/j.jin2104118.
 50. Goldshmit, Y., Sztal, T.E., Jusuf, P.R., Hall, T.E., Nguyen-Chi, M., and Currie, P.D. (2012). Fgf-dependent glial cell bridges facilitate spinal cord regeneration in zebrafish. *The Journal of neuroscience : the official journal of the Society for Neuroscience* 32, 7477-7492. 10.1523/JNEUROSCI.0758-12.2012.
 51. Dervan, A.G., and Roberts, B.L. (2003). Reaction of spinal cord central canal cells to cord transection and their contribution to cord regeneration. *The Journal of comparative neurology* 458, 293-306. 10.1002/cne.10594.
 52. Nona, S.N., and Stafford, C.A. (1995). Glial repair at the lesion site in regenerating goldfish spinal cord: an immunohistochemical study using species-specific antibodies. *Journal of neuroscience research* 42, 350-356. 10.1002/jnr.490420309.
 53. Tsarouchas, T.M., Wehner, D., Cavone, L., Munir, T., Keatinge, M., Lambertus, M., Underhill, A., Barrett, T., Kassapis, E., Ogryzko, N., et al. (2018). Dynamic control of proinflammatory cytokines Il-1beta and Tnf-alpha by macrophages in zebrafish spinal cord regeneration. *Nature communications* 9, 4670. 10.1038/s41467-018-07036-w.
 54. Matsuoka, R.L., Marass, M., Avdesh, A., Helker, C.S., Maischein, H.M., Grosse, A.S., Kaur, H., Lawson, N.D., Herzog, W., and Stainier, D.Y. (2016). Radial glia regulate vascular patterning around the developing spinal cord. *eLife* 5, e20253. 10.7554/eLife.20253.
 55. John, N., Kolb, J., and Wehner, D. (2022). Mechanical spinal cord transection in larval zebrafish and subsequent whole-mount histological processing. *STAR Protocols* 3, 101093. <https://doi.org/10.1016/j.xpro.2021.101093>.
 56. Ma, R.C., Kocha, K.M., Mendez-Olivos, E.E., Ruel, T.D., and Huang, P. (2023). Origin and diversification of fibroblasts from the sclerotome in zebrafish. *Developmental biology* 498, 35-48. 10.1016/j.ydbio.2023.03.004.
 57. Freeman, J., Vladimirov, N., Kawashima, T., Mu, Y., Sofroniew, N.J., Bennett, D.V., Rosen, J., Yang, C.T., Looger, L.L., and Ahrens, M.B. (2014). Mapping brain

- activity at scale with cluster computing. *Nature methods* 11, 941-950. 10.1038/nmeth.3041.
58. Zeng, C.W., Kamei, Y., Shigenobu, S., Sheu, J.C., and Tsai, H.J. (2021). Injury-induced Cav1-expressing cells at lesion rostral side play major roles in spinal cord regeneration. *Open Biol* 11, 200304. 10.1098/rsob.200304.
 59. Ohnmacht, J., Yang, Y.J., Maurer, G.W., Barreiro-Iglesias, A., Tsarouchas, T.M., Wehner, D., Sieger, D., Becker, C.G., and Becker, T. (2016). Spinal motor neurons are regenerated after mechanical lesion and genetic ablation in larval zebrafish. *Development* 143, 1464-1474. 10.1242/dev.129155.
 60. Chen, S., and Birk, D.E. (2013). The regulatory roles of small leucine-rich proteoglycans in extracellular matrix assembly. *Febs J* 280, 2120-2137. 10.1111/febs.12136.
 61. Itoh, Y., Sahni, V., Shnyder, S.J., McKee, H., and Macklis, J.D. (2023). Inter-axonal molecular crosstalk via Lumican proteoglycan sculpts murine cervical corticospinal innervation by distinct subpopulations. *Cell reports* 42, 112182. 10.1016/j.celrep.2023.112182.
 62. Milich, L.M., Choi, J.S., Ryan, C., Cerqueira, S.R., Benavides, S., Yahn, S.L., Tsoulfas, P., and Lee, J.K. (2021). Single-cell analysis of the cellular heterogeneity and interactions in the injured mouse spinal cord. *The Journal of experimental medicine* 218. 10.1084/jem.20210040.
 63. Eze, U.C., Bhaduri, A., Haeussler, M., Nowakowski, T.J., and Kriegstein, A.R. (2021). Single-cell atlas of early human brain development highlights heterogeneity of human neuroepithelial cells and early radial glia. *Nature neuroscience* 24, 584-594. 10.1038/s41593-020-00794-1.
 64. Shen, Y., Tenney, A.P., Busch, S.A., Horn, K.P., Cuascut, F.X., Liu, K., He, Z., Silver, J., and Flanagan, J.G. (2009). PTPsigma is a receptor for chondroitin sulfate proteoglycan, an inhibitor of neural regeneration. *Science* 326, 592-596. 10.1126/science.1178310.
 65. Domeniconi, M., Cao, Z., Spencer, T., Sivasankaran, R., Wang, K., Nikulina, E., Kimura, N., Cai, H., Deng, K., Gao, Y., et al. (2002). Myelin-associated glycoprotein interacts with the Nogo66 receptor to inhibit neurite outgrowth. *Neuron* 35, 283-290. 10.1016/s0896-6273(02)00770-5.
 66. Fournier, A.E., GrandPre, T., and Strittmatter, S.M. (2001). Identification of a receptor mediating Nogo-66 inhibition of axonal regeneration. *Nature* 409, 341-346. 10.1038/35053072.
 67. Peri, F., and Nusslein-Volhard, C. (2008). Live imaging of neuronal degradation by microglia reveals a role for v0-ATPase a1 in phagosomal fusion in vivo. *Cell* 133, 916-927. 10.1016/j.cell.2008.04.037.
 68. Lopez, S.G., and Bonassar, L.J. (2022). The role of SLRPs and large aggregating proteoglycans in collagen fibrillogenesis, extracellular matrix assembly, and mechanical function of fibrocartilage. *Connect Tissue Res* 63, 269-286. 10.1080/03008207.2021.1903887.
 69. Bougas, K., Stenport, V.F., Currie, F., and Wennerberg, A. (2012). Laminin Coating Promotes Calcium Phosphate Precipitation on Titanium Discs in vitro. *J Oral Maxillofac Res* 2, e5. 10.5037/jomr.2011.2405.
 70. Rosso, G., Wehner, D., Schweitzer, C., Mollmert, S., Sock, E., Guck, J., and Shahin, V. (2022). Matrix stiffness mechanosensing modulates the expression and distribution of transcription factors in Schwann cells. *Bioeng Transl Med* 7, e10257. 10.1002/btm2.10257.
 71. Flanagan, L.A., Ju, Y.E., Marg, B., Osterfield, M., and Janmey, P.A. (2002). Neurite branching on deformable substrates. *Neuroreport* 13, 2411-2415. 10.1097/00001756-200212200-00007.
 72. Koser, D.E., Thompson, A.J., Foster, S.K., Dwivedy, A., Pillai, E.K., Sheridan, G.K., Svoboda, H., Viana, M., Costa, L.D., Guck, J., et al. (2016). Mechanosensing is critical for axon growth in the developing brain. *Nature neuroscience* 19, 1592-1598. 10.1038/nn.4394.
 73. Abe, K., Baba, K., Huang, L., Wei, K.T., Okano, K., Hosokawa, Y., and Inagaki, N. (2021). Mechanosensitive axon outgrowth mediated by L1-laminin clutch interface. *Biophysical journal* 120, 3566-3576. 10.1016/j.bpj.2021.08.009.
 74. Rosso, G., Young, P., and Shahin, V. (2017). Mechanosensitivity of Embryonic Neurites Promotes Their Directional Extension and Schwann Cells Progenitors Migration. *Cell Physiol Biochem* 44, 1263-1270. 10.1159/000485485.
 75. Georges, P.C., Miller, W.J., Meaney, D.F., Sawyer, E.S., and Janmey, P.A. (2006). Matrices with compliance comparable to that of brain tissue select neuronal over glial growth in mixed cortical cultures. *Biophysical journal* 90, 3012-3018. 10.1529/biophysj.105.073114.
 76. Franze, K. (2020). Integrating Chemistry and Mechanics: The Forces Driving Axon Growth. *Annu Rev Cell Dev Biol* 36, 61-83. 10.1146/annurev-cellbio-100818-125157.
 77. Chorfa, A., Madjoubi, A., Mohamed, H., Bouras, N., Rubio, J., and Rubio, F. (2010). Glass hardness and elastic modulus determination by nanoindentation using displacement and energy methods. *Ceramics - Silikaty* 54.
 78. He, S., Su, Y., Ji, B., and Gao, H. (2014). Some basic questions on mechanosensing in cell-substrate interaction. *Journal of the Mechanics and Physics of Solids* 70, 116-135. 10.1016/j.jmps.2014.05.016.
 79. Koch, D., Rosoff, W.J., Jiang, J., Geller, H.M., and Urbach, J.S. (2012). Strength in the periphery: growth cone biomechanics and substrate rigidity response in peripheral and central nervous system neurons. *Biophysical journal* 102, 452-460. 10.1016/j.bpj.2011.12.025.
 80. Tyanova, S., and Cox, J. (2018). Perseus: A Bioinformatics Platform for Integrative Analysis of Proteomics Data in Cancer Research. *Methods in molecular biology* 1711, 133-148. 10.1007/978-1-4939-7493-1_7.

REVIEWERS' COMMENTS

Reviewer #1 (Remarks to the Author):

The authors have thoroughly and comprehensively addressed all concerns raised regarding their manuscript. Although I may have some differing interpretations of the results, I strongly believe that embracing diverse perspectives is crucial for healthy academic discourse. As such, I wholeheartedly support and endorse the publication of this manuscript.

Reviewer #2 (Remarks to the Author):

The authors have done great work in respond to my concerns. I have no further questions.

Reviewer #3 (Remarks to the Author):

The authors addressed all the points I raised. Taking also into consideration the comments by the others reviewers, the authors performed additional experiments that strengthen their conclusions.

Reviewer #4 (Remarks to the Author):

The results in the first version of the paper were already very relevant. However, I had some concerns about the interpretation and display of some of the results. I am now impressed by how well the authors have successfully addressed all my concerns, specifically:

- Analysis of proteomic data has now been revised and improved.
- Addition of new data and clarification of the interpretation of the in vitro and TUNEL data.
- Brillouin data is now correctly displayed.
- Experimental details on the generation of transgenic lines and plasmid sequences are now included.

In my opinion, this is a very exciting paper that is ready for publication.

REVIEWERS' COMMENTS

Reviewer #1 (Remarks to the Author):

The authors have thoroughly and comprehensively addressed all concerns raised regarding their manuscript. Although I may have some differing interpretations of the results, I strongly believe that embracing diverse perspectives is crucial for healthy academic discourse. As such, I wholeheartedly support and endorse the publication of this manuscript.

RESPONSE: We thank this reviewer for their previously helpful suggestions and insightful comments, which have improved the manuscript. We also thank this reviewer for their open academic discourse. No further changes are required.

Reviewer #2 (Remarks to the Author):

The authors have done great work in respond to my concerns. I have no further questions.

RESPONSE: We thank this reviewer for their previously helpful suggestions and insightful comments, which have improved the manuscript. No further changes are necessary.

Reviewer #3 (Remarks to the Author):

The authors addressed all the points I raised. Taking also into consideration the comments by the others reviewers, the authors performed additional experiments that strengthen their conclusions.

RESPONSE: We thank this reviewer for their previously helpful suggestions and insightful comments, which have improved the manuscript. No further changes are necessary.

Reviewer #4 (Remarks to the Author):

The results in the first version of the paper were already very relevant. However, I had some concerns about the interpretation and display of some of the results. I am now impressed by how well the authors have successfully addressed all my concerns, specifically:

- Analysis of proteomic data has now been revised and improved.
- Addition of new data and clarification of the interpretation of the in vitro and TUNEL data.
- Brillouin data is now correctly displayed.

- Experimental details on the generation of transgenic lines and plasmid sequences are now included.

In my opinion, this is a very exciting paper that is ready for publication.

RESPONSE: We thank this reviewer for their previously helpful suggestions and insightful comments, which have improved the manuscript. No further changes are necessary.